§

# CRI-HOM: A novel chemical mechanism for simulating Highly Oxygenated Organic Molecules (HOMs) in global chemistry-aerosol-climate models.

James Weber[1], Scott Archer-Nicholls[1], Paul Griffiths[1,2], Torsten Berndt[3], Michael Jenkin[4], Hamish Gordon[5], Christoph Knote[6], Alexander T. Archibald[1,2]

[1]Centre for Atmospheric Science, Department of Chemistry, University of Cambridge, Cambridge, CB2 1EW, UK
[2]National Centre for Atmospheric Science, Department of Chemistry, University of Cambridge, CB2 1EW, UK
[3]Atmospheric Chemistry Department (ACD), Leibniz Institute for Tropospheric Research (TROPOS), Leipzig, 04318, Germany
[4]Atmospheric Chemistry Services, Okehampton, Devon, UK
[5]Engineering Research Accelerator and Center for Atmospheric Particle Studies, Carnegie Mellon University, Pittsburgh, PA 15213, USA
[6]Meteorologisches Institut, Ludwig-Maximilians-Universität München, Munich, 80333, Germany

*Correspondence to*: James Weber (jmw240@cam.ac.uk)

**Abstract.**

*We present here results from a new mechanism, CRI-HOM, which we have developed to simulate the formation of highly oxygenated organic molecules (HOMs) from the gas phase oxidation of α-pinene, one of the most widely emitted BVOCs by mass. This concise scheme adds 12 species and 66 reactions to the Common Representative Intermediates (CRI) mechanism v2.2 Reduction 5 and enables the representation of semi-explicit HOM treatment suitable for long term global chemistry-aerosol-climate modelling, within a comprehensive tropospheric chemical mechanism. The key features of the new mechanism are (i) representation of the autoxidation of peroxy radicals from the hydroxyl radical and ozone initiated reactions of α-pinene, (ii) formation of multiple generations of peroxy radicals, (iii) formation of accretion products (dimers) and (iv) isoprene-driven suppression of accretion product formation, as observed in experiments. The mechanism has been constructed through optimisation against a series of flow tube laboratory experiments. The mechanism predicts a HOM yield of 2-4.5% under conditions of low to moderate NO$_x$, in line with experimental observations, and reproduces qualitatively the decline in HOM yield and concentration at higher NO$_x$. The mechanism gives a HOM yield that also increases with temperature, in line with observations, and our mechanism compares favourably to some of the limited observations of [HOM] observed in the boreal forest in Finland and in the south east USA.*

*The reproduction of isoprene-driven suppression of HOMs is a key step forward as it enables global climate models to capture the interaction between the major BVOC species, along with the potential climatic feedbacks. This suppression is demonstrated when the mechanism is used to simulate atmospheric profiles over the boreal forest and rainforest; different isoprene concentrations result in different [HOM] distributions, illustrating the importance of BVOC interactions in atmospheric composition and climate. Finally particle nucleation rates calculated from [HOM] in present day and pre-industrial atmospheres suggest that "sulphuric acid free" nucleation can compete effectively with other nucleation pathways in the boreal forest, particularly in the pre-industrial, with important implications for the aerosol budget and radiative forcing.*

## 1 Introduction

Aerosols play an important role in the Earth system by affecting the Earth's radiative balance as well as local air quality and thus human health (Carslaw et al., 2010). Aerosols can interact directly with solar radiation through scattering or absorption

§

and indirectly by influencing cloud properties by seeding cloud droplets as well as increasing cloud albedo (Forster et al., 2007, Twomey., 1974). Thus, aerosols change the balance between the energy received from the Sun and the energy emitted from the planet at the top of the atmosphere. However, a major uncertainty in climate change predictions arises from aerosols and aerosol-cloud interactions (Stocker et al., 2014). This arises in part from a lack of understanding of pre-industrial (PI) aerosol and it is the change in aerosol burden from the PI to the present day (PD) which determines the effective radiative forcing (ERF) of aerosols. As PI aerosols sources are almost exclusively natural, an understanding of natural sources and the associated aerosol formation processes is essential if better predictions for climate change are to be made.

An important formation route for aerosols is oxidation of volatile organic compounds which form less volatile species that can partition into the aerosol phase or nucleate new particles (Kirkby et al., 2016, Shrivastava et al., 2017). Recently it has been established that the oxidation of organic compounds can lead to the formation of "highly oxygenated organic molecules" (HOMs) (Mentel et al., 2014, Ehn et al., 2014, Kurtén et al., 2016, Bianchi et al., 2019) (also referred to as "highly oxidised multifunctional organic compounds (Ehn et al., 2012)) which are formed by multiple intramolecular oxidation steps, termed autoxidation (Crounse et al., 2013, Bianchi et al., 2019). Autoxidation typically involves the abstraction by a peroxy radical of a hydrogen atom bonded to a carbon elsewhere on the molecule resulting in an alkyl radical and hydroperoxide group. The alkyl radical reacts rapidly with atmospheric oxygen to form a new peroxy radical, ultimately reducing the species' volatility and enabling particle formation/condensation. HOMs are defined as closed-shell species with at least 6 oxygens formed by initial atmospheric oxidation and subsequent autoxidation steps (Bianchi et al., 2019). HOM formation has been observed from anthropogenic species (Berndt et al., 2018a) and biogenic species such as α-pinene (Molteni et al., 2019, Berndt et al., 2018b). The semi-explicit mechanism described for the first time in this paper describes the formation of HOMs from α-pinene in a form suitable for global modelling studies. This provides a framework for incorporating a comprehensive description of pure biogenic nucleation into a global model and, ultimately, allowing for a more rigorous description of aerosol formation and the climatic consequences. α-pinene is considered as it is the most widely studied and widely emitted monoterpene (~32 Tg yr$^{-1}$, Sindelarova et al., 2014) and, with measured HOM yields around 3-10 % (Ehn et al., 2014, Jokinen et al. 2015), has the potential to produce 2-7 Tg HOM yr$^{-1}$ with the range arising from uncertainties in emissions, HOM yield and difference in mass between the precursor BVOC and the HOMs which will have at least 6 additional oxygen atoms but in some cases, considerably more. HOM yields from β-pinene (the second most widely emitted monoterpene, Sindelarova et al., 2014) and isoprene (the most widely emitted BVOC, Sindelarova et al., 2014) are negligible (Ehn et al., 2014). Limonene has emissions around 25% of α-pinene (Sindelarova et al., 2014) and is likely to have a higher HOM yield (Ehn et al., 2014, Jokinen et al., 2015) although a much wider range of values have been reported than for α-pinene. Limonene thus may have the potential to produce a similar mass of HOM as α-pinene and its consideration may be an area of future work. Emissions of anthropogenic VOCs account for ~10% of total VOC emissions (Guenther et al., 1995), roughly the same quantity as monoterpene emissions, and, as no species have HOM yields above 2.5% (Ehn et al., 2014, Jokinen et al., 2015, Kirkby et al., 2016, Bianchi et al., 2019), the contribution of anthropogenic VOCs to HOM is likely to be significantly smaller. Nevertheless, the speciation of anthropogenic VOCs in the mechanism means that addition of HOMs from these sources will be possible and, from an urban air quality perspective very important.

Nucleation of new particles from sulphuric acid is an important means of new particle formation (NPF) in the atmosphere (Kulmala et al., 1998). Sulphuric acid can also form new particles with oxidised organic species (Riccobono et al., 2015). However, extremely involatile HOMs can participate in NPF, without necessarily needing a sulphuric acid seed in a process

§

termed pure biogenic nucleation (PBN) (Kirkby et al., 2016, Gordon et al., 2016). Despite playing important roles in aerosol formation and growth, the relatively recent discovery of HOMs and the complexity of their formation means that their role in particle formation and contribution to aerosol has been assessed in only very few global model studies (Gordon et al., 2016, Zhu et al., 2019). The ability of PBN to change atmospheric aerosol loading by providing a route to particle formation without sulphuric acid has been illustrated (Gordon et al., 2016) with this effect particularly important in the pre-industrial (PI) atmosphere, where lower $SO_2$ emissions resulted in greater sensitivity of aerosol loading to alternative formation routes (i.e. including PBN) and a higher simulated aerosol burden than previous studies. As a result, Gordon et al (2016) calculated that the radiative forcing change from the PI to PD caused by cloud cover change was 27% lower than previous estimates. Meanwhile, Zhu et al (2019), highlighting the fact that many chemistry schemes fail to reproduce nucleation rates in low sulphuric acid concentrations, showed the complex effect PBN has in the PI and PD with a more complicated mechanism but one which also omitted autoxidation and accretion product formation. Including PBN in a global chemistry-aerosol scheme resulted in a much larger increase in the magnitude of the (negative) aerosol indirect effect (AIE) in the PI than the PD. This has potentially important consequences as it means that the effective radiative forcing (ERF) of aerosols from the PI to PD may be smaller than previously expected. This in turn would mean that climate sensitivity is lower than previously thought as aerosols are offsetting a smaller amount of warming arising from the enhanced concentrations of greenhouse gases than previously thought, with implications for predictions of future climate change as well.

The peroxy radicals produced from α-pinene oxidation by OH or $O_3$ have been observed to undergo autoxidation under typical atmospheric conditions (Ehn et al., 2014, Jokinen et al., 2014, Berndt et al., 2016, Berndt et al., 2018b). The autoxidation competes with the bimolecular reaction of peroxy radicals with NO, $NO_3$, $HO_2$ and other peroxy radicals and its yield is thus dependent on background atmospheric composition. Therefore, an accurate description of HOMs requires consideration of $NO_x$ and oxidant concentrations as well as autoxidation; indeed, elevated $NO_x$ has been observed to suppress HOM formation (Lehtipalo et al., 2018). The first order rate constants for autoxidation can vary over several orders of magnitude (~$10^{-6}$ – $10^{2}$ $s^{-1}$) depending on nearby functional groups (Otkjær et al., 2018, Bianchi et al., 2019, Crounse et al., 2013, Xu et al., 2018, Kurten et al., 2015). Autoxidation rates also exhibit a significant positive temperature dependence (Jenkin et al., 2019a, Bianchi et al., 2019) and HOM yield has been observed to be highly temperature dependent (Quéléver et al., 2019).  Thus, the overall competitiveness of autoxidation is dependent on the background atmospheric composition and ambient temperature as well as the molecule undergoing oxidation.

In addition to autoxidation, the formation of HOM accretion products (also called dimers (Kurtén et al., 2016, Bianchi et al., 2019)) by reactions between two peroxy radicals has been observed to be significant with large peroxy radicals (Kirkby et al., 2016, Berndt et al., 2018a, Berndt et al., 2018b, Jenkin et al., 2019a, Molteni et al., 2019). These species are predicted to be more involatile than 10-carbon HOMs (also termed monomers (Kurtén et al., 2016)) with important implications for new particle formation and contribution to SOA. In established schemes such as the Master Chemical Mechanism (MCM) (Jenkin et al., 1997, Saunders et al., 2003), the Common Representative Intermediates (CRI) (Utembe et al., 2010, Watson et al., 2010, Jenkin et al., 2010, Jenkin et al., 2019) and the Chemistry of the Stratosphere and Troposphere (Strat-Trop) used in the climate model UKCA (United Kingdom Chemistry and Aerosol) (Archibald et al., 2019), formation of accretion products is not included as it was previously considered negligible or too complex to include. However, experimental work suggests that accretion product formation is a competitive pathway for larger peroxy radicals, such as those formed from  α-pinene (Berndt et al., 2018b, Molteni et al., 2019, Simon et al., 2020) and indeed becomes more favourable with increasing functionality and size; the rate coefficient for the accretion reaction between two α-pinene derived peroxy radicals was observed to be 16-80 times greater (depending on the extent of oxidation) than for the analogous reaction between two isoprene derived peroxy radicals (Berndt et al., 2018a, Berndt et al., 2018b).

§

The mechanistic treatment of HOMs in numerical models has varied considerably from simple steady state approximations (Gordon et al., 2016) through basic mechanisms linked to a volatility basis set (Schervish et al., 2019) to more explicit descriptions based on the MCM featuring either a limited subset of HOMs without accretion product formation or autoxidation

(Zhu et al., 2019) or a near explicit description involving over 1700 reactions (Roldin et al., 2019). Whilst the addition of PBN represents an important process level improvement in models, the approaches discussed above all have some degree of limitation. As well as omitting accretion product formation, simpler models (Gordon et al., 2016, Schervish et al., 2019) do not fully capture the influence of oxidant levels, $NO_x$ or temperature while the more explicit schemes (Zhu et al., 2019, Roldin et al., 2019) are too computationally expensive for long term climate studies. Further, none of the schemes include the relatively

novel observation of suppression via reactive $RO_2$ cross reactions (McFiggans et al., 2019), although this has been identified as an important area for future research (Zhu et al., 2019, Roldin et al., 2019) and is addressed in this work.

New experimental evidence suggests that isoprene may suppress the formation of the most involatile accretion products and thus the smallest aerosol particles (Berndt et al 2018b, McFiggans et al., 2019, Heinritzi et al., 2020) and reproducing the

effect of isoprene has been identified as an important requirement for future mechanisms (Roldin et al., 2019). This inhibition is driven firstly by isoprene scavenging OH radicals (Lee et al., 2016, Kiendler-Scharr et al., 2009, Berndt et al, 2018b) thus reducing the formation of large peroxy radicals by reaction of α-pinene with OH. The second driver is the scavenging of the 10-carbon α-pinene peroxy radicals ("C10RO2) by isoprene peroxy radicals ("C5RO2") and the other smaller peroxy radicals from species such as CO and $CH_4$ (McFiggans et al., 2019). These C10RO2 could otherwise form

20-carbon accretion products (Eq. 1) which are predicted to be highly involatile (Kurtén et al., 2016). The reaction of isoprene peroxy radicals with the α-pinene radicals produces the 15-carbon accretion product ("C15d") (Eq. 2) as well as closed shell species ("C10" and "C5") (Eq. 3) and alkoxy radicals ("C10RO" and "C5RO") (Eq. 4) which isomerise or fragment (Jenkin et al, 2019a).

$C10RO_2 + C10RO_2 \rightarrow C20d + O_2$ (1)

$C10RO_2 + C5RO_2 \rightarrow C15d + O_2$ (2)

$C10RO_2 + C5RO_2 \rightarrow C10 + C5 + O_2$ (3)


$C10RO_2 + C5RO_2 \rightarrow C10RO + C5RO + O_2$ (4)

This inhibition affects the aerosol size distribution (an important parameter for the radiative forcing of aerosol (Zhu et al., 2019)) by favouring the growth of larger existing aerosol particles by promoting the production of smaller, more volatile

species which can partition to pre-existing aerosol rather than the nucleation of new particles from larger, less volatile species. This has the potential to have important consequences for future predictions of SOA and the negative feedback proposed to exist between biogenic VOC emissions and atmospheric temperature (Kulmala et al., 2004, Carslaw et al., 2010, Sporre et al., 2018). Such predictions, based on the modelled increases in isoprene and monoterpene emissions in a warmer climate (Kulmala et al., 2013, Sporre et al., 2018) have not considered the potential perturbation to NPF and atmospheric aerosol loading by

isoprene and the changes in radiative forcing which may result. Reassessing the sign and size of this feedback by coupling the chemistry scheme described in this work to an aerosol scheme within a global climate models is a key long-term aim of this work.

§

In this study we describe our work developing a new mechanism sufficiently concise for global chemistry climate models that
can simulate the process of autoxidation, the formation of HOMs from α-pinene and the influence of isoprene. In Section 2 the
development of the mechanism from the principles of gas phase chemistry is described and in Section 3 we discuss mechanism
optimisation and validation against experimental data and the parent mechanism, CRI v2.2. In Section 4, the mechanism is
used to simulate atmospheric HOM profiles and explore implications for new particle formation. Finally, in Section 5,
conclusions for further work are drawn.

**2 Mechanism Development**

Our new mechanism we have developed builds on the Common Representative Intermediates (CRI) scheme version 2.2
Reduction 5 (Jenkin et al., 2019b) (hereafter the "base mechanism"), developed from the fully explicit Master Chemical
Mechanism (MCM) version 3.3.1 (Jenkin et al., 2015) which describes the degradation of organic compounds in the
troposphere. In the CRI framework, species are lumped together into surrogate molecules whose behaviour is optimised against
the fully explicit MCM. The CRI v2.2 R5 mechanism describes the degradation of α-pinene, β-pinene, isoprene and 19 other
emitted VOC species.

The updates we have made to the base mechanism to produce the new HOM-forming functionality include the addition of
autoxidation of α-pinene oxidation products and a more detailed peroxy radical pool scheme. These changes enable the
formation of 10-carbon, 15-carbon and 20-carbon HOMs and add 12 species and 66 reactions to the base mechanism. The new
chemical mechanisms for ozonolysis and OH oxidation are shown in Figure 1 and Figure 2 respectively. We now describe the
changes made to the base mechanism to incorporate HOM chemistry in more detail.

**2.1 Ozonolysis**


To simulate the autoxidation reactions formed from ozonolysis 5 new peroxy radical species were added to the base
mechanism. The peroxy radicals formed from ozonolysis of α-pinene and subsequent autoxidation steps are collectively termed
"O3RO2". In the base mechanism, α-pinene reacts with ozone to produce the single lumped peroxy radical RN18AO2 and
acetone. This single mechanistic step represents multiple chemical steps, with RN18AO2 representing a 6 or 7 carbon species.
In addition, TNCARB26 (closed shell carbonyl species) and RCOOH25 (pinonic acid) arise from the reaction of Criegee
intermediates with water. The yields of these species, 17.5% and 2.5% respectively, remain unchanged from the CRI v2.2
mechanism and are well supported in the literature (IUPAC Task Group on Atmospheric Chemical Kinetic Data Evaluation,
Atkinson and Arey (2003), Johnson, Marston et al (2008)). RN18AO2 goes on to react with standard peroxy radical reaction
partners; HO₂ forming hydroperoxides; NO and NO₃ forming alkoxy radicals; and the peroxy radical pool forming alkoxy
radicals, carbonyls and alcohols, as described in (Jenkin et al., 2019a).

In our new mechanism, RN18AO2 is replaced with the tracers RN26BO2 and RTN24O2. RN26BO2 represents the 10-carbon
peroxy radicals formed directly from the cleavage of the ozonide and subsequent addition of atmospheric oxygen, which can
then undergo autoxidation. RTN24O2, a species already in the CRI mechanism, represents the 9 carbon peroxy radical species
(MCM species C96O2) which is also formed from ozonolysis but does not undergo autoxidation in this mechanism. RN26BO2
is termed "1st generation" as it has undergone one oxidation step and it can undergo autoxidation to form the 2nd, 3rd, 4th and
lumped higher generation species, termed RN25BO2O2, RN24BO4O2, RN23BO4O2 and RNxBOyO2 species respectively
(RNxBOyO2 does not undergo further autoxidation but does undergo all the other reactions). In the base mechanism, the first

§

number featured in a species' name is an index which refers to the number of NO-to-NO$_2$ conversions possible, which depends on the number of C-C and C-H bonds. During the H-shift step of autoxidation, a C-H bond is usually broken to produce the alkyl radical which then forms the peroxy radical from atmospheric oxygen and so each autoxidation step reduces the index by one while the number of oxygens is increased by 2. For example, the autoxidation of the 2$^{nd}$ generation O3RO2 to 3$^{rd}$ generation is expressed by Eq.5.

$$RN25BO2O2 \rightarrow RN24BO4O2 \text{ (5)}$$

Each generation of peroxy radical also undergoes bimolecular reactions. Reaction with HO$_2$ produces a hydroperoxide species; for the 2$^{nd}$ and later generations, the product is classified as a HOM (C10z) as they fulfil the criteria discussed by Bianchi et al (2019), while for the 1$^{st}$ generation species the resulting hydroperoxide is RTN26OOH, a species already present in the CRI (Eq. 6).

$$RNaBObO2 + HO_2 \rightarrow RTN26OOH \text{ or } C10z \text{ (6)}$$

Reaction with NO and NO$_3$ yields nitrates or alkoxy radicals and accurately representing the behaviour of these products is crucial to reproducing the effect of NO$_x$ on HOM formation. Alkoxy radicals are not represented explicitly due to their rapid reactions which, typically for alkoxy radicals formed from larger peroxy radicals, are decomposition or isomerisation. Decomposition produces two smaller species, one closed shell and one a peroxy radical while isomerisation produces a more functionalised peroxy radical via an alkyl radical intermediate with one fewer oxygen than would have been added via autoxidation. Faced with very limited data and the fact that the precise fate of an alkoxy radical will depend considerably on molecular structure and neighbouring groups, a branching ratio 50:50 for decomposition and isomerisation was adopted. "Sensitivity tests perturbing the branching ratio between 75:25 and 25:75 were performed to probe the consequences of this uncertainty. These tests suggested the precise value of this branching ratio within this range did not affect the fitting of rate coefficients for autoxidation and accretion product formation (Section 3.1). These branching ratio perturbations did lead to changes in HOM yield (Fig. S4(b)) and are discussed in more detail in Section 3.2." The decomposition products are existing CRI species CARB16 and RN10O2 or RN9O2 and the isomerisation product the next generation peroxy radical as shown in the example reaction (Eq. 7).

$$RN25BO2O2 + NO \rightarrow 0.5RN24BO4O2 + 0.5CARB16 + 0.5RN9O2 \text{ (7)}$$

A schematic of the additions made to the CRI for the ozonolysis scheme is shown in Figure 1.

## 2.2 OH oxidation

Autoxidation through the OH initiated oxidation pathway resulted in the addition of 6 new species, including 5 new peroxy radicals. The peroxy radicals formed from OH oxidation of α-pinene and subsequent autoxidation steps are termed "OHRO2". The single peroxy radical RTN28O2 in the base mechanism is replaced by RTN28AO2, representing the two species which do not undergo autoxidation (APINAO2 and APINBO2 in the MCM), and RTN28BO2 (MCM APINCO2) which can undergo autoxidation to form higher generation peroxy radicals (Xu et al., 2018). The 2$^{nd}$ and 3$^{rd}$ generation OHRO2 are represented explicitly (RTN27BO2O2 and RTN26BO4O2) and all 4$^{th}$ generation and higher species are lumped together as RTNxBOyO2

§

for mechanistic simplicity. The chemical treatment of RTN28AO2 is the same as the original CRI species RTN28O2 while all other OHRO2 (except RTNxBOyO2) can undergo autoxidation (Eq. 8).

$$RTNaBObO2 \rightarrow RTN(1-a)BO(b+2)O2 \ (8)$$


Reaction of the 1$^{st}$ and 2$^{nd}$ generation OHRO2 (RTN28BO2 and RTN27BO2O2) with HO$_2$ yields the hydroperoxide RTN28OOH which is not classified as a HOM due to insufficient oxygens.   The HOM produced by all later generation OHRO2 is termed C10x (Eq. 9).

$\quad RTNaBObO2 + HO_2 \rightarrow RTN28OOH \ or \ C10x \ (9)$

Reaction with NO and NO$_3$ is treated in the same manner as O3RO2 except for RTN28BO2 which follows the reaction of the analogous species APINCO2 in the MCM. A schematic of the additions made to the base mechanism for the OH oxidation scheme is shown in Fig. 2.


The pathway initiated by reaction of α-pinene with NO$_3$ was not considered in this work but is identified as an area for future work. A summary of the peroxy radical species in the two pathways is given in Table 1 and full mechanistic description provided in the SI.

## 2.3 Peroxy Radical + Peroxy Radical Interactions

Reactions between peroxy radicals can result in the formation of two alkoxy radicals (Eq. 10), a carbonyl and an alcohol (Eq. 11) or accretion product (Eq. 12) (Jenkin et al., 2019a).

$$RO_2 + R'O_2 \rightarrow RO + R'O + O_2 \ (10)$$

$$RO_2 + R'O_2 \rightarrow R_{-H}O + R'OH + O_2 \ or \ R'_{-H}O + ROH + O_2 \ (11)$$


$$RO_2 + R'O_2 \rightarrow ROOR' + O_2 \ (12)$$

Rather than represent every possible RO$_2$-RO$_2$ reaction combination, the base mechanism uses a peroxy radical pool. Each peroxy radical undergoes a unimolecular reaction with a first order rate coefficient determined by the total peroxy radical 275 concentration and the geometric mean of the self-reaction rates of the methyl peroxy radical and radical of interest (Jenkin et al., 2019a).

While computationally efficient, such a mechanism fails to represent the effect of peroxy radical size on the distribution of products. While negligible for small peroxy radicals, accretion product formation is more favourable when larger, more 280 functionalised peroxy radicals react (Berndt et al., 2018b, Schervish et al., 2019). To describe the reactions between the differently sized peroxy radicals, we have split the single peroxy radical pool into three pools for small (<4 carbons), medium (4-7 carbons) and big (>7 carbons) peroxy radicals. Each big radical reacts separately with each peroxy radical pool while, to minimise the total number of reactions, all small peroxy radicals react with the total pool as accretion product formation is much less favourable (Jenkin et al., 2019a). Medium peroxy radicals are discussed below. The use of large and medium peroxy

§

radical pools allows for improved representation of the competition between peroxy radicals with different reactivity in our new mechanism and is a substantial improvement over the base mechanism.

Table 2 summarises the products for a specific peroxy radical reacting with the different peroxy radical pools. As discussed previously, alkoxy radials were not simulated explicitly, rather they decomposed into closed shell products and peroxy radicals. 290 A full list of the contents of each peroxy radical pool is given in the SI.

In practice, the alkoxy radicals formed from the reaction of isoprene-derived peroxy radicals (which are likely to dominate the medium size pool) with other peroxy radicals decompose rapidly into the major products: closed shell carbonyls methyl vinyl ketone and methacrolein (UCARB10) and the minor product hydroxy vinyl carbonyl (UCARB12) (Jenkin et al., 2015). The 295 accretion of isoprene-derived peroxy radicals has been measured to be over an order of magnitude slower than the accretion of peroxy radicals derived from α-pinene (Berndt et al., 2018b), supporting the theory that accretion product formation becomes more favourable with increasing peroxy radical size. Therefore, to limit complexity, all medium peroxy radicals in the mechanism simply react with the overall peroxy radical pool and their accretion product formation was ignored.

**2.3.1 Large Peroxy Radical Pool**

The reaction of a large peroxy radical with the large peroxy radical pool ($RO2_b$) can produce an accretion product (Eq. 13), closed species (Eq. 14) or an alkoxy radical (Eq. 15) (Jenkin et al., 2019a) which then reacts as discussed in sections 2.1 and 2.2. Note that a single C10RO2 species will produce half a C20d accretion product for the purposes of mass conservation.

$$C10RO_2 + RO2_b \rightarrow 0.5C20d + O_2 : k_{13} \text{ (13)}$$

$$C10RO_2 + RO2_b \rightarrow C10z/C10x/TNCARB26 : k_{14} \text{ (14)}$$

$$C10RO_2 + RO2_b \rightarrow C10RO : k_{15} \text{ (15)}$$


The rate coefficient for C20d formation, $k_{13}$, increased with the extent of oxidation of the reacting peroxy radical. This was done to simulate the observed behaviour that accretion product formation becomes faster as the reacting peroxy radicals become more functionalised (Berndt et al., 2018a, Berndt et al., 2018b). Thus, the reaction forming C20d from the 1st generation peroxy radicals had a lower rate coefficient than the analogous reactions involving higher generation RO2 species 315 (see reactions 21, 23, 25, 27, 29, 46, 63, 64, 65 and 66 in the SI reaction list for full breakdown). The rate coefficients used, 0.4-3.6×10⁻¹¹ cm³ molecule⁻¹ s⁻¹ were derived from fitting against experimental data (Berndt et al., 2018b) and, as discussed in Section 3.1.3, and were in line with the range measured by Berndt et al (2018b) (0.97-7.9×10⁻¹¹ cm³ molecule⁻¹ s⁻¹). This resulted in R13 being more important that R14 and R15 which had rate coefficients based on literature (Molteni et al., 2019, Roldin et al., 2019, ) up to an order of magnitude lower.


The rate coefficient for the formation of the closed shell species from O3RO2, $k_{14}$, was taken as the mean of the rate coefficients measured by Molteni et al (2018) (1.68×10⁻¹² cm³ molecule⁻¹ s⁻¹). The rate coefficient for alkoxy radical formation, $k_{15}$, was assumed to have the same value as $k_{14}$ (i.e. a 50:50 branching ratio between these pathways).This value is close to the value of closed shell : alkoxy radical of 40:60 ratio suggested for primary and secondary peroxy radicals by Jenkin et al (2019a) but 325 further from the 20:80 suggested for tertiary peroxy radicals. However, sensitivity tests where the mechanism was run with branching ratios of 40:60 and 20:80 revealed that the precise values of this branching ratio within this range did not affect the fitting of rate coefficients for autoxidation and accretion product formation (Section 3.1). These branching ratio perturbations

§

led to negligible changes in HOM yield (Section 3.2, Simulation C) which were much smaller than the range in the HOM yield simulated to arise from the uncertainty in the autoxidation temperature dependence and are therefore considered to be of minor
importance (Table S6).

The rate coefficients for formation of closed shell species and alkoxy radicals from OHRO2 with $RO2_b$ were taken from Roldin et al (2019).

**2.3.2 Medium and Small Peroxy Radical Pools**
Reaction of a large peroxy radical with the medium peroxy radical pool ($RO2_m$) can produce a 15-carbon accretion product (Eq. 16), closed shell species (Eq. 17) or an alkoxy radical (Eq. 18) which is not modelled explicitly but rather decomposes rapidly into another closed shell product and peroxy radical.

$$C10RO_2 \; + \; RO2_m \; \rightarrow \; 0.667 C15d \; + \; O_2 : k_{16} \; (16)$$


$$C10RO_2 \; + \; RO2_m \; \rightarrow \; C10z/C10x/TNCARB26 : k_{17} \; (17)$$

$$C10RO_2 \; + \; RO2_m \; \rightarrow \; C10RO : k_{18} \; (18)$$

In a manner similar to C20d formation, the rate coefficient for C15d formation, $k_{16}$, in the mechanism increases with the extent of oxidation of the reacting large peroxy radical; the reaction forming C15d from the 1st gen peroxy radicals had a lower rate coefficient than the analogous reactions involving higher generation $RO_2$ species (see reactions 36, 37, 38, 39, 40, 47, 71, 72, 73 and 74 in the SI reaction list for full breakdown). The fitting of $k_{16}$ rate coefficients to experimental data is discussed in Section 3.1.3. The chosen fitted values of $1.8$-$7.5 \times 10^{-12}$ cm$^3$ molecule$^{-1}$ s$^{-1}$ were lower than the range measured by Berndt et al
(2018b) ($1.2$-$3.6 \times 10^{-11}$ cm$^3$ molecule$^{-1}$ s$^{-1}$).

Reaction of a large peroxy radical with the small peroxy radical pool ($RO2_s$) produces a closed shell species or an alkoxy radical, in a manner analogous to Eq. 17 and Eq. 18.

**2.4 HOM Loss Mechanisms**

The number of different molecules falling under the C10x, C10z, C15d and C20d umbrellas is huge, making the treatment of loss processes complex. Losses will occur via chemical or photolytic degradation as well as to condensation to aerosol, to the nucleation sink and dry and wet deposition. Physical loss is believed to be the major sink for HOM (Maso et al., 2002, Petäjä et al., 2009, Tan et al., 2018, Wu et al., 2018, Bianchi et al., 2019). For simplicity, in the simulations below, we ignore wet
and dry deposition but do include loss to the condensation sink for modelling simulations C and D (Table 3).

Chemical losses of HOMs are highly uncertain. It is suggested that HOMs will react with OH (Bianchi et al., 2019). In this model, OH reacts with C10x, C10z, C15d and C20d with the rate coefficient of the large hydroperoxide, RTN28OOH to produce the smaller closed shell CRI species CARB10 and CARB15 as well as UCARB10 (lumped methacrolein and methyl
vinyl ketone) for 15-carbon dimers. This rate coefficient was $2.38 \times 10^{-11}$ molecules$^{-1}$ cm$^3$ s$^{-1}$ and, in light of suggestion that the rate coefficient of OH with HOM could be higher (Bianchi et al., 2019), sensitivity tests increasing the rate coefficient to $1 \times 10^{-10}$ molecules$^{-1}$ cm$^3$ s$^{-1}$ were performed but no material effect was observed. Photolysis of peroxide and carbonyl linkages produce alkoxy radicals which behave as previously described. Photolysis frequencies are taken from the MCM (Jenkin et al.,

§

1997, Saunders et al., 2003). Given the small concentrations of HOMs we predict that uncertainty in these gas phase loss
processes are likely to have small impacts on the general features of tropospheric chemistry (i.e. OH reactivity or ozone
production).

Physical loss is believed to be the major sink for HOM (Dal Maso et al., 2002, Petäjä et al., 2009, Tan et al., 2018,
Wu et al., 2018, Bianchi et al., 2019). Loss to the condensation sink presents a complex challenge. The saturation vapour
pressure will vary considerably (Kurtén et al., 2016) for HOMs, even within the C10 umbrella, affecting the fraction which
partition to the aerosol phase. Furthermore, some HOMs are likely to have aldehyde and alcohol moieties which will enhance
their removal via reactive uptake into the aerosol phase, particularly if it is aqueous. When using our new mechanism in
different simulations (Table 3), condensation sinks have been set to fixed values or values taken from literature.

Having described the additions and changes made to the base mechanism to develop our new mechanism, we now discuss the
optimisation                and                validation                of                the                mechanism.

### 3 Mechanism Optimisation and Validation

Here we discuss the optimisation of the new mechanism and its validation. In total 4 simulations were performed with the
mechanism as detailed in Table 3.

### 3.1 Comparison to Experimental Data

There exists a limited amount of experimental data which provides an insight into the behaviour of the multiple generations of
peroxy radicals produced from ozonolysis and OH-oxidation of α-pinene (Berndt et al., 2018b). Using a flow cell, Berndt
measured the concentration of α-pinene-derived peroxy radicals produced by ozonolysis (O3RO2) and OH oxidation (OHRO2)
and 20-carbon (C20d) and 15-carbon (C15d) accretion products at the end of the flow tube using a chemical ionisation-
atmospheric pressure interface-time of flight (CI-APi-TOF) mass spectrometer and a chemical ionisation-time of flight (CI3-
TOF) mass spectrometer. The observed peroxy radicals spanned several generations of autoxidation, namely the 1st-4th
generation for O3RO2 species and 1st-3rd generation for OHRO2. Berndt et al (2018b) also calculated rate coefficients for
accretion product formation using the observation that accretion product concentration increased linearly with time with an
assumed uncertainty no greater than a factor of 3. Reagent ions used in the CI-APi-TOF were $C_3H_7NH_3^+$, $CH_3COO^-$ and $NO_3^-$
and in the CI3-TOF $NH_4^+$. The flow tube experiments lasted for 7.9 s, at which point the flow was sampled by the mass
spectrometers. Reactions proceeded under dark conditions at 1 atm and 297 K under low [$NO_x$] (<$10^8$ cm$^{-3}$). The low
concentrations of bimolecular reaction partners $HO_2$ and NO meant that multiple autoxidation steps could occur in the reaction
time.

Flow tubes operating under laminar flow are easily modelled using box models as there are very few complications to consider
in terms of mixing and wall loss and no new particle formation was observed. A box model version of the mechanism was
compiled in the BOXMOX framework (Knote et al., 2015). The experimental data allowed the autoxidation coefficients of the
peroxy radical species and the rate coefficients of accretion product formation to be constrained. The concentrations of peroxy
radicals and accretion products in the box model were evaluated at the end of the 7.9 s reaction period and compared with
experiments. A process of iterative adjustment to autoxidation and accretion product formation rate coefficients in the
mechanism was performed to produce the best reproduction of the experimental data by the mechanism.

§

Two experiments from Berndt et al (2018b) were considered. In the first experiment, flow tube runs were performed with varying initial concentrations of α-pinene (3-50 ppb) with initial $O_3$ at 28 ppb (Simulation A). In the second experiment, runs were performed with fixed initial α-pinene (15.6 ppb) and $O_3$ (80 ppb) but with initial isoprene concentrations varying from 0-60 ppb (Simulation B). Comparison to this experimental data facilitated examination of the model's ability to reproduce the concentration of HOM-precursors and accretion products with and without isoprene as well as at moderate and high $O_3$ mixing

ratios.

An important parameter in the mechanism was the yield of RN26BO2 from α-pinene ozonolysis. This yield is uncertain and it was found to affect the autoxidation coefficients required to reproduce the experimental data for the O3RO2 species. To constrain it, a first-order autoxidation rate coefficient of 0.206 $s^{-1}$ for RN26BO2 was imposed, based on theoretical analysis of

the α-pinene ozonolysis system (Kurtén et al., 2015), and the yield adjusted until the mechanism was able to achieve the best possible fit to the data. This resulted in a yield of 50% for RN26BO2 and 30% for RTN24O2. However, this remains a source of uncertainty and warrants further investigation. The low $NO_x$ conditions meant that the autoxidation coefficients dominated the concentration of later generation O3RO2 and OHRO2 and, from this starting point, the autoxidation rate coefficients for later generations were fitted against experimental data over multiple rounds of optimisation (Table 4). The autoxidation

coefficient for the 1st generation OHRO2, RTN28BO2, was taken as 2.1 $s^{-1}$ based on Xu et al (2018). An estimation of the uncertainty in the autoxidation is also provided in Table 4. These values were calculated by adjusting the autoxidation rate coefficients one at a time to determine the maximum and minimum values of an autoxidation rate coefficient for which the corresponding peroxy radical would fall within the experimental uncertainty region. This approach neglects any cross-sensitivities through the joint uncertainty in several rate coefficients. A full Monte Carlo uncertainty analysis addressing this

issue is beyond the scope of this manuscript but would make a valuable follow up for future work in this field. Therefore, the autoxidation rate coefficient uncertainties are large as the experimental error uncertainties are large.

The autoxidation coefficients in Table 4 are higher than those considered in the theoretical study of Scherivish et al (2019) but closer to the values measured by Zhao et al (2018) and the values suggested by Roldin et al (2019). The mechanism using the autoxidation coefficients from Table 4 and predicted the lumped higher generation species (5th generation for O3RO2, 4th

generation for OHRO2) at concentrations higher than observed in the work by Berndt et al (2018b). This suggests there may be additional, as yet unknown loss processes for the more highly oxidised peroxy radical species which are not incorporated in this work. Additional loss process would likely reduce the fitted autoxidation coefficients because they would provide an additional sink for the $RO_2$ species which does not lead to the production of the next generation $RO_2$. Therefore, the autoxidation coefficients determined in this work are likely to be upper limits but further insight into this is not possible with

the data currently available. This is a key area for further study.

Unfortunately, the flow tube studies of Berndt et al (2018b) lack observations to constrain the full chemical space simulated by the box model. In particular there were no measurements of OH, $HO_2$ or NO. The effect of the uncertainty in the initial experimental concentrations of NO, $HO_2$ and OH on the modelled concentrations of O3RO2, OHRO2 and accretion products

and thus fitted the autoxidation coefficients and accretion production formation rates coefficients was investigated with a series of sensitivity tests. Initial conditions of $10^6$ $cm^{-3}$ for OH and 4 ppt for $HO_2$ were used. NO and $NO_2$ were initialised at 4 ppt, based on the purity of the flow gas (personal communication with T. Berndt). For $HO_2$, NO and $NO_2$ sensitivity simulations indicated that increases of 10 ppt (250% increase) and decreases of 3 ppt (75% decrease) did not lead to deviations in the concentrations of $RO_2$ or accretion products sufficient to warrant a change in the rate coefficients for autoxidation or accretion

product formation. Initial OH concentration had negligible effect (<5% change) on O3RO2, OHRO2 and C20d when varied over $10^5$ – $2\times10^6$ $cm^{-3}$ (90 % decrease, 100 % increase).

§

### 3.1.1 Varying α-pinene Experiment

In Simulation A, the mechanism was used to simulate various experiments with increasing initial mixing ratios of α-pinene with a fixed mixing ratio of 26 ppb of $O_3$. The modelled 1st-4th generation O3RO2 species (Fig. 3(b)) agreed well with the observed concentrations, with all of the model results falling within experimental uncertainty bounds (although we note these are large).

The model was also able to reproduce the 2nd and 3rd gen OHRO2 species (Fig. 3(a)) but struggled with the 1st generation OHRO2 species, overestimating it by a factor of 10, despite reproducing the general trend of variation with α-pinene. The experimental measurements of 1st generation OHRO2 concentration from Berndt et al (2018b) were believed to be underestimated by about a factor of 5 (Fig. S3). This suggests the model overprediction of the concentration of OHRO2 may be about a factor of 2. The cause of the discrepancy between modelled and measured 1st generation OHRO2 remains unclear. The rate coefficient for the production of the 1st generation OHRO2 has undergone extensive evaluation and the same coefficient is used in the CRI v2.2 parent mechanism which has been optimised against the Master Chemical Jenkin (Jenkin et al., 1997, Saunders et al., 2003, Jenkin et al., 2015, Jenkin et al., 2019a). Sensitivity tests perturbing the branching ratio between RTN28AO2 and RTN28BO2 revealed that even doubling the fraction of RTN28BO2, a significant deviation from literature (Berndt et al., 2016, Pye et al., 2018), had negligible effect as did changing initial [OH] by +100 % / -90%. Another explanation is the presence of additional, as yet unknown loss processes not currently included in the model, but in the absence of additional data, no further insights can be made at this time. More importantly, the 1st generation OHRO2 does not form HOM itself and so it is unlikely to have a significant impact on HOM concentration. Furthermore, the modelled 1st generation OHRO2 was dominated by RTN28AO2, the species which does not autoxidise to form later generation $RO_2$. Nevertheless, this remains an important area for future work but one where more data is needed for additional constraints to be put in place.

### 3.1.2 Varying Isoprene Experiment

In Simulation B, varying initial concentrations of isoprene were employed under conditions of fixed initial mixing ratios of α-pinene (15.6 ppb) and $O_3$ (80 ppb). The model was able to reproduce the observed decline in the 2nd and 3rd generation OHRO2 species (Fig. 4(a)) with increasing isoprene while also reproducing the minor decrease in the O3RO2 species (Fig. 4(b)). The decline in OHRO2 mirrored the modelled decrease in OH, providing a strong indication that OH scavenging by isoprene is playing a major role in decrease in OHRO2, as suggested by McFiggans et al (2019), also highlighting the need for better understanding of $HO_x$-recycling during isoprene oxidation.

### 3.1.3 Simulation of Accretion Products

The 20-carbon accretion products were measured in both the isoprene-free, varying α-pinene experiment (as in Fig. 3) and, separately, under conditions of constant α-pinene and varying isoprene (as in Fig. 4). The rate coefficients for 20-carbon accretion product formation were fitted against experimental data (Berndt et al., 2018b) and incorporated the increase in propensity to form accretion products with $RO_2$ oxidation.

The species with the lowest functionality, the 1st generation OHRO2 (RTN28AO2 and RTN28BO2), which contain only 3 oxygens, had the lowest value of $k_{13}$ ($0.4\times10^{-11}$ $cm^3$ $molecule^{-1}$ $s^{-1}$) while the 1st generation O3RO2 (RN26BO2) with 4 oxygens had $k_{13}=0.97\times10^{-11}$ $cm^3$ $molecule^{-1}$ $s^{-1}$, its self-reaction rate coefficient determined by Berndt et al (2018b). The most functionalised species for O3RO2 (RNxBOyO2) and OHRO2 (RTNxBOyO2) were assigned values of $k_{13}$ of $3.6\times10^{-11}$ $cm^3$

molecule$^{-1}$ s$^{-1}$ and $3.5\times10^{-11}$ cm$^3$ molecule$^{-1}$ s$^{-1}$ respectively. The fitted rate coefficients used were in line with the range 0.97-7.9$\times10^{-11}$ cm$^3$ molecule$^{-1}$ s$^{-1}$ (with an uncertainty no greater than a factor of 3) measured by Berndt et al (2018b) and the full list of values is given in the reaction list in the SI. This reproduced, within experimental error, the total observed C20d concentrations for both experiments (Fig. 5 and Fig. S1) as well as the RO$_2$ in Simulations A and B. Sensitivity studies which scaled all k13 values by the same factor before rerunning Simulation B and comparing the output to experimental data suggested that variations in the C20d formation rate coefficients of +100 % / -35 % spanned the experimental uncertainty (Table S6).

Using values of $k_{13} = 0.1$-$3 \times 10^{-12}$ cm$^3$ molecule$^{-1}$ s$^{-1}$, as suggested by Roldin et al (2019) for C20d formation, produced C20d concentrations lower than those observed (Fig S1) while values of $1$-$8 \times 10^{-10}$ cm$^3$ molecule$^{-1}$ s$^{-1}$ from Molteni et al (2019) produced values which were higher than observation.

The rate coefficients for 15-carbon accretion product formation, fitted against experimental data, also increased with the extent of oxidation of the reacting peroxy radical. Values of $k_{16}$ ranging from $1.2 \times 10^{-12}$ cm$^3$ molecule$^{-1}$ s$^{-1}$ for the least oxidised RO$_2$ to $5 \times 10^{-12}$ cm$^3$ molecule$^{-1}$ s$^{-1}$ for the most oxidised species  reproduced observed levels of the C15d accretion product (Fig. 5 and Fig. S2) from the constant α-pinene and variable isoprene experiments (as in Fig. 4) and were lower than the values measured by Berndt ($1.3$-$2.3 \times 10^{-11}$ cm$^3$ molecule$^{-1}$ s$^{-1}$ with an uncertainty no greater than a factor of 3). Sensitivity studies which scaled all k13 values by the same factor before rerunning Simulation B and comparing the output to experimental data suggested that variations in the C15d formation rate coefficients of ±50 % spanned the experimental uncertainty (Table S6).

Figure 5 shows that the decrease in 20-carbon accretion products with increasing isoprene far outweighs the increase in 15-carbon accretion products. The mechanism reproduces the general trend of suppression of total accretion product concentration with increasing initial isoprene concentration. This finding is in good agreement with McFiggans et al (2019) and highlights a key component of the new mechanism which simple mechanisms (e.g. Gordon et al., 2016) will miss. In the model this net decrease in accretion products concentrations is driven in part by OH scavenging (and the subsequent reduction in OHRO2 (Fig. 4)). In this work this was the major driver of C20d decrease. However, suppression was also observed due to scavenging of C10RO2 by isoprene-derived RO2 as observed by McFiggans et al (2019). The influence of smaller peroxy radicals such as that from methane on accretion product formation (McFiggans et al., 2019) will be an area of future investigation.

## 3.2 HOM yield variation with temperature and NO$_x$

Autoxidation reactions have significant positive temperature dependencies (Praske et al., 2018, Bianchi et al., 2019, Jenkin et al., 2019a). Accordingly, HOM yields are expected to be highly temperature sensitive. Quéléver et al (2019) recorded a 50-fold increase in HOM yield at 293 K relative to 273 K. This temperature variation cannot be attributed to the temperature dependence of the initial oxidation of α-pinene as the rate coefficient of ozonolysis increases only 17% between 273 K and 293 K while the reactions with OH and NO$_3$ exhibit negative temperature dependencies. Frege et al (2018) measured a decrease in O:C ratio values in HOMs with reducing temperatures, attributing this to a reduction in autoxidation.

Variation in peroxy radical structure and functionality will result in different generation peroxy radicals having different barriers to autoxidation (Bianchi et al., 2019). A few modelling studies have considered the temperature dependencies of the autoxidation rate coefficient in peroxy radical from α-pinene derivatives. Schervish et al (2019) considered a simple approach where all generations of peroxy radicals from α-pinene ozonolysis had a fixed activation energy of 62.4 or 66.5 KJ mol$^{-1}$ (θ =

§

7500-8000 K when rate coefficient is expressed as k=Ae$^{-\theta/T}$), while noting that a reduction in barriers to autoxidation with increasing functionality is plausible but so far unproven. By contrast, Roldin et al (2019) considered a higher activation energy of 100.4 KJ mol$^{-1}$ ($\theta$ = 12077 K) based on the theoretical work of Kurtén et al (2015) which identified activation energies of 90-120            KJ            mol$^{-1}$            for            $\alpha$-pinene.

Given the lack of additional literature in this area, 3 versions of the new mechanism were created to probe the effects of temperature and activation energy on HOM yield and subsequent evolution. In each mechanism all autoxidation reactions (for O3RO2 and OHRO2) had the same activation energy while all other rate coefficients were the same across mechanisms (Table S2). Activation barriers of $\theta$ = 6000 K, $\theta$ = 9000 K and $\theta$ = 12077 K were chosen as they included the range suggested by Roldin et al (2019) and Schervish et al (2019) and the mechanism versions termed HOM$_{6000}$, HOM$_{9000}$ and HOM$_{12077}$

respectively. For the temperature dependent versions, the pre-exponential factor of the autoxidation coefficient (Table S5) was adjusted so that the autoxidation coefficients were the same at 297 K as those derived from the comparison to experimental data from Berndt et al (2018b). It is recognised that the autoxidation steps are likely to have different activation energies, but this analysis provides a first approximation of the influence of activation energy on HOM formation.

In a simulation modelling an instantaneous injection of $\alpha$-pinene (Simulation C), the HOM yield for the 10-carbon species, individually and in total (defined in the SI), was calculated with the three different temperature dependencies (HOM$_{6000}$, HOM$_{9000}$ and HOM$_{12077}$) at temperatures of 270 K, 290 K and 310 K for initials conditions of $\alpha$-pinene 15 ppb, O$_3$ 40 ppb and OH at $10^6$ cm$^{-3}$ and a temperature independent condensation sink of $2\times10^{-3}$ s$^{-1}$.

Figure 6(a) shows the results from the simulations performed with initial concentrations of NO and NO$_2$ of 0.01-10 ppb. These simulations showed that the 10-carbon HOM yield tends to increase slightly from very low (0.02 ppb) to moderate (2 ppb) NO$_x$ and before starting to decrease with increasing NO$_x$ thereafter. This behaviour is likely to be due to the inclusion in the mechanism of the isomerisation pathway via reaction with NO which yields the next generation peroxy radical. This pathway has been suggested as an important route for forming more highly oxidised derivatives of $\alpha$-pinene due to the potential rapid

ring-opening mechanism involving alkoxy radicals and the cyclobutyl ring found in $\alpha$-pinene ozonolysis products (Rissanen et al., 2015). Figure 6(a) indicates that the absolute yield is also sensitive to temperature, with the highest yields simulated at the highest temperatures. At low temperatures (blue data), the uncertainty in autoxidation temperature dependence has the greatest effect while at high temperature this feature is muted. The increase in yield with temperature is in qualitative agreement with observation (Quéléver et al., 2019, Simon et al., 2020).


The model predicted total HOM yields at 290 K of 1.9±0.2 % (0.01 ppb NO) to 3.9±0.5% (1 ppb NO), with the quoted range resulting from the range of temperature dependencies considered. This is within the ranges previously suggested by Jokinen et al (2015) (1.7-6.8 %) and close to the values from Ehn et al (2014) (3.5-10.5 %) and Sarnela et al (2018) (3.5-6.5%), while lower than Roldin et al (2019) (~7%).. In addition, the HOM yield at 270 K of ~0.6-1.9 % compared favourably with the yield

of ~2% determined by Roldin et al (2019). This suggests that the mechanism is doing a good job at simulating HOM yield. The slight low bias may be in part due the values of k14 and k15 which were shown to influence the HOM yield relatively strongly. Sensitivity tests involving doubling and halving of the rate coefficients produced HOM yield changes of around +65% and -40% respectively while preserving the general dependencies on NO$_x$ and temperature (Fig. S4(a)) (Table S6). This area of uncertainty will be the focus of future work.


§

**The HOM yield showed negligible sensitivity to the alkoxy radical decomposition-isomerisation branching ratio below 200 ppt of NO$_x$ and around ±0.7 percentage points** (~20 %) **at 2 ppb NO$_x$.** However, **this range was encompassed by the range arising from autoxidation temperature dependence uncertainty. Above 2 ppb NO$_x$, this ratio had greater influence as NO reactions with RO$_2$ started to compete more efficiently with autoxidation but this coincided with the sharp drop in HOM yield** (Fig. S4(b)). (Therefore, **while further work is needed to develop the isomerisation-decomposition branching ratio description, it is unlikely to have a significant influence in the low-NO$_x$ conditions where HOM are predicted to** be **most prevalent and in these conditions the uncertainty in temperature dependence of autoxidation is predicted to have a larger effect**

**3.3 Comparison to CRI v2.2**

The ability of the new mechanism to reproduce the concentrations of key atmospheric species from the CRI v2.2 under different emissions of NO$_x$ and α-pinene was assessed using an 8 day box modelling run (Simulation D). Over the majority of emissions space, O$_3$ differed by less than 0.05 ppb (0.1%), OH by less than 0.4% and NO by less than 2.5 ppt (0.4%) (Fig. S6-S8) with similar (or better) agreement for other important species (Fig. S9-S17). Acetone was routinely underpredicted (Fig. S18) by between a factor of ~14 at 50 ppt NO$_x$ and by ~20 % at 2-10 ppb of NO$_x$ but this did not result in significant deviation between the base mechanism and new mechanism for O$_3$ or OH. This indicates that the basic features of atmospheric chemistry, such as HO$_x$ recycling processes added in the CRI v2.2, which have been shown to have important consequences for atmospheric composition (Jenkin et al., 2019b), are preserved in the new mechanism.

**3.3.1 Peroxy Radicals and HOMs**

As our model simulations indicate, and as has previously been observed (Lehtipalo et al., 2018), at higher concentrations of NO$_x$ there is inhibition of HOM formation with the principle driver being the reaction of NO with peroxy radicals occurring at a rate outcompeting autoxidation. While reaction with NO can in part aid HOM formation by increasing the isomerisation pathway, the fragmentation pathway (forming smaller species) and formation of non-HOM nitrates leads to a reduction in total HOM. Accordingly, all these mechanisms predict a decrease in HOM concentration with increasing NO$_x$ (Fig. 6(b)). Furthermore, very little difference is observed between the different HOM mechanisms, suggesting that uncertainty in the activation energy may not be too great an impediment to understanding general HOM behaviour, at least at the temperatures considered.

However, it should be remembered that, at present, the HOM tracers in the mechanism represent a range of species with varying levels of oxidation. For example, C10z corresponds to HOMs formed from 2$^{nd}$ to 5$^{th}$ O3RO2 generations of O3RO2 and C10x to HOMs from all generations of OHRO2. The predicted concentrations of different generations of O3RO2 (Fig. S19) and OHRO2 (Fig. S20) were also observed to decrease with NO$_x$ with little difference between different HOM mechanisms. Furthermore, the most abundant peroxy radicals for both pathways were the lumped highest generation species, highlighting the potential issue of lack of loss processes for the most highly oxidised RO$_2$ species as was encountered when fitting parameters to flow cell data.

**3.3.2 Closed Shell and non-HOM Species**

In the base mechanism, α-pinene oxidation predominantly leads to the formation of the closed shell species CARB16 and TNCARB26 (carbonyls) and RN18NO3 and RTN28NO3 (nitrates). Whilst the base mechanism has been optimised against

§

the MCM, these pathways have not themselves been tuned extensively to reproduce concentrations observed in experimental systems. These species are still present in the new mechanism but the concentrations of RTN28NO3 and TNCARB26 are predicted to be slightly lower than in the base mechanism (Fig. S21) while CARB16 is simulated as being much lower in concentration than in the base mechanism at low $NO_x$ with the difference attributed to the added competition from the autoxidation pathways.

The nitrate species formed from RN18AO2 in the base mechanism, RN18NO3, is significantly lower in the HOM mechanism. Nitrate yield is a complicated topic, not least because the MCM predicts that the peroxy radicals C107O2 and C109O2, which are used to represent the 1st generation O3RO2 in part (RN26BO2), do not form nitrate upon reaction with NO or $NO_3$. Importantly this does not affect the $O_3$ and OH concentrations, but this should be an area that future work addresses.

## 4 Simulation of HOM Vertical Profiles

Given the success of the mechanism in simulating the laboratory flow tube experiments and its ability to capture the sensitivity of HOMs to changes in the reactivity of the peroxy radical pool, we focus now on using the scheme to simulate tropospheric conditions to investigate, to first order, how the scheme would predict ambient HOM concentrations. Here we focus on using our new mechanism to simulate the surface [HOMs] and the vertical profile of [HOMs]. We also look at the effect of the simulated HOMs on nucleation rates in the lower troposphere.

The vertical profiles of HOMs over the boreal forest near Hyttalia in Finland (61° 9' N, 23° 4' E) and near Manaus in the Amazon rainforest (-2° 35' N, 60° 12' W) were investigated along with surface concentrations representative of Brent, Alabama (32.903°N, 87.250°W). For each altitude level, the sensitivity simulations were performed with three different activation energies for the autoxidation steps ($HOM_{6000}$, $HOM_{9000}$ and $HOM_{12077}$). Vertical transport was neglected - a different box model was run at each vertical level with the inputs being the output of a simulation with the UKCA model (Archibald et al., 2019) sampled at 14:00 LT (photolysis frequencies were adjusted to account for the solar zenith angle but not for altitude variation). Hourly concentration data from UKCA were provided for $O_3$, OH, isoprene, $\alpha$-pinene, $HO_2$, NO, $NO_2$, $NO_3$, $N_2O_5$, temperature and pressure for 16th June averaged over 2010-2014, from the grid boxes in UKCA corresponding to the respective locations discussed above, and supplemented with monthly mean concentrations of a further 23 species (Table S3) with the concentration of certain species adjusted to observations (Kuhn et al., 2007 and Table 4.1) and scaled vertically to account for biases in the UKCA output. The scaled values of isoprene and $\alpha$-pinene showed reasonable agreement with observations taken up to 80 m in altitude at the ATTO tower (Yáñez-Serrano et al., 2015). Modelled isoprene fell within 0.5 ppb of observation taken at 2 pm in June while modelled monoterpene were within 0.1 ppb of observation, well within the observational standard deviation in both cases (Fig. S22). HOM condensation sinks (CS) (equal for all HOM species) discussed in Lee et al (2016) (Table S4) were used at the surface and scaled using the modelled vertical profile of aerosol surface area density. Sensitivity studies revealed that the output of the 1D box modelling shows significant sensitivity of [HOMs] to the magnitude and profile of the CS (Fig. S22). Therefore, we can suggest that our simulated vertical profiles be regarded as illustrativeses more work is required to identify if the condensation sink should be species dependent.

Table 5 summarises the comparison of our model simulations of near surface [HOMs] compared to observations. In the boreal forest in Hyytiala, the range of predicted 10-carbon [HOM] falls at the higher end of the mean observational range and well below the maximum observed concentrations (1-1.5 ppt) (Roldin et al., 2019). The predicted 20-carbon accretion product concentration is around double the the mean observational range and well below the maximum observed values (0.6-0.7 ppt).

§

In Alabama, the model produces a reasonable value given that the observation dataset includes 9-carbon species not considered in the model at present. The model results in Table 5 provide strong support that when implemented in a global chemistry climate model, our new scheme should perform well if the underlying emissions of BVOCs and NO$_x$ and the CS are well simulated.

Figure 7 shows the concentrations of the different HOMs from the box model version as a function of altitude above Hyytiala and over the Amazon rainforest near Manaus. In both locations, the 10-carbon HOM profiles roughly mirror α-pinene with roughly equal abundance of the species from ozonolysis and OH oxidation. HOM from OH showed a significantly greater sensitivity to temperature, diverging from the HOM from O$_3$ at around 5 km in Hyytiala and 8 km in the Amazon due to the elevated temperature profile. This was attributed to the requirement for 1$^{st}$ generation OHRO2 to undergo two autoxidation steps before HOMs can be formed (Section 2.2) while 1$^{st}$ generation O3RO2 only need to undergo one autoxidation step and thus have a weaker temperature dependence. This effect only becomes noticeable at temperatures below ~250 K when autoxidation ceases to compete effectively with bimolecular reactions. In spite of higher [BVOCs], the considerably higher CS in the Amazon region (Lee et al., 2016) resulted in lower [HOM] within the boundary layer than at Hyytiala, while the warmer temperatures also resulted in a negligible dependence on the activation energy in the lowest 4 km (i.e. the shaded areas are smaller in Fig. 7(b) than in Fig. 7(a)).

Hyytiala and the Amazon represent very different chemical environments with the isoprene/α-pinene ratio (I/AP) playing an important role in the accretion product distribution; 15-carbon accretion products are simulated as being more abundant than 20-carbon accretion products in the Amazon with the biggest difference predicted at low altitude where I/AP is greatest. By contrast, in Hyytiala where I/AP is smaller, 20-carbon accretion products are more abundant.

Figure 7 highlights strong vertical profiles for the simulated [HOMs]. The simulations over the Amazon suggest a significant secondary peak in [HOMs] at around 4-5 km in altitude; in part linked to an increase in the [BVOCs] at this height. In our simulations each vertical level is represented by a different box model simulation so there is no simulation of the advection of HOMs. It will be interesting to see how future fully coupled model simulations simulate the vertical profile of [HOMs] and how this affects processes like aerosol formation and climate.

**4.1 Nucleation Rates**

Given the important role Gordon et al (2016) identified for HOMs in NPF we extend our 1-D calculations to investigate the implications of the predicted HOM profiles on nucleation rates using monthly mean climate model data from the PD and PI. Nucleation rates from two different nucleation mechanisms were studied: (i) neutral and ion-induced pure biogenic nucleation (PBN) (Kirkby et al., 2016); and (ii) activation of sulphuric acid (SA$_{act}$) (Kulmala et al., 2006; Sihto et al., 2006) suitable for the boundary layer. All HOMs were treated as being equally proficient at nucleating new particles, in agreement with approach and nucleation rates used by Kirkby et al (2016) and Gordon et al (2017). Recent work by Heinritzi et al (2020) suggests that 20-carbon accretion products may be better at nucleating new particles and therefore the results presented are likely to be an upper bound although nevertheless informative. Representing the different nucleation efficiencies of different HOM species will be investigated in future work. The results of the calculations of nucleation rates using these schemes are summarised in Figure 8. (The nucleation rate expressions are given in the SI.)

There exists little observational data on nucleation solely from PBN mechanisms, making model validation hard. Modelled

§

surface sulphuric acid concentrations at Hyytiala ($2\text{-}3\times10^6$ cm$^{-3}$) fall within the range of observations ($3\times10^5\text{-}2\times10^7$ cm$^{-3}$ (Boy et al., 2005, Petäjä et al., 2009)). Modelled concentrations in the Amazon ($3\times10^4$ cm$^{-3}$) were lower than observation ($10^5\text{-}10^6$ cm$^{-3}$ (Wimmer et al., 2018)) although the observations were taken in a pasture site downwind of Manaus surrounded by the rainforest not in the rainforest itself and are therefore likely to be higher than in-situ rainforest values. Thus, the nucleation rates we have calculated for SA$_{act}$ are likely to be a reasonable estimate in Hyytiala and low biased in the Amazon.

Figure 8 shows predicted nucleation rates in the PI and PD in the Amazon and Hyytiala derived from our simulated [HOMs] vertical profile in the boundary layer and low free troposphere using June monthly mean data from a UKESM historical run taken from the PI (June average 1851-1856) and PD (June average 2009-2014). In all cases, the PBN nucleation rates decline rapidly with height above the boundary layer. In the Boreal forest, the nucleation rate from PBN at very low altitudes is calculated to be around 20-25% of that from SA$_{act}$ in the PD. However, in the PI it is comparable to the SA$_{act}$ rate, contributing 40-80 % of the total nucleation rate in the lowest 500 m (Fig. S23). The greater relative importance of PBN in the PI, despite lower predicted [HOM], was attributed to two factors. Firstly, predicted steady state ion concentrations were higher in the PI in Finland than the PD due to the PI's lower ion CS. This increased the rate of the ion-induced pathway PBN pathway. Secondly, the considerably lower modelled concentrations of sulphuric acid in PI (around 10x lower than in the PD) reduced the importance of SA$_{act}$. By contrast, the lower concentrations of predicted [HOM] in the Amazon led to PBN having a much smaller contribution to the total nucleation rate (< 5% in the PD and a negligible impact in the PI). This is in agreement with multiple sources (Andreae et al., 2015, Wimmer et al., 2018, Rizzo et al., 2018). The importance of PBN in the PI atmosphere in certain locations, qualitatively in agreement with Gordon et al (2016), illustrates the potential importance including PBN in climate models could have on aerosol burden and the associated radiative effects.

**5 Summary and Conclusions**

We present a novel chemical mechanism, CRI-HOM, for simulating HOM formation based on the latest version of the Common Representative Intermediate scheme (CRI v2.2). Focusing on the most important natural source of HOMs, α-pinene, the CRI-HOM mechanism is one of the first HOM mechanisms ready for incorporation into existing chemistry-aerosol climate models. The scheme is much more complex than previous steady state approximations (Gordon et al., 2016) and so enables non-linear interactions and feedbacks with the chemical environment to be represented, but far more concise than other mechanisms that have been developed which treat the complex structural characteristics of the formation of HOMs (Roldin et al., 2019). The addition of 12 new species and 66 reactions means that this scheme can be used for long term global chemistry-aerosol-climate studies.

Firstly, the mechanism was optimised against flow cell data and validated by comparison to observed HOM yields. A key result was the ability of the mechanism to reproduce observations of isoprene driving a decline in HOM peroxy radical precursors and 20-carbon accretion products (and total accretion product concentration) (McFiggans et al., 2019). The need for further research into the loss processes of the highly oxidised peroxy radicals was identified to reconcile the disparity between modelled and observed concentrations. The effect of other peroxy radicals, such as those from smaller more abundant organic species, on accretion product formation is also an area for future investigation.

After optimisation, the CRI-HOM was compared to the base mechanism (CRI v2.2) and very good agreement was observed for a wide range of atmospheric gases including O$_3$, OH and its precursors. This indicated that the important features of HO$_x$ recycling and accurate O$_3$ representation, developed in the CRI v2.2, had been preserved in CRI-HOM.

§

In further tests of CRI-HOM, HOM yields and concentrations were predicted to decrease with increasing $NO_x$ and increase with temperature, in agreement with previous theoretical and observational studies. The temperature dependence of autoxidation was investigated using activation energies spanning the range of values suggested in literature (Roldin et al., 2019, Schervish et al., 2019). Temperature dependence was predicted to have a significant effect on HOM yield at 270 K but a much smaller effect at 290 K and 310 K.

In a final experiment, vertical profiles of HOM were simulated using a pseudo 1D box model for Finland, Alabama and the Amazon using chemistry climate model data as inputs. The model performed well relative to observations in Finland and Alabama. The influence of the condensation sink (CS) value on [HOM] was shown to be significant with [HOM] predicted to be significantly lower in the Amazon, despite higher [BVOC], due to the higher CS. Improving the description of the CS within the mechanism has been identified as an important area of future work. The profiles also illustrated the dependence of HOM on the chemical environment, driven chiefly by the interaction of isoprene and monoterpenes. The higher concentrations of isoprene in the Amazon resulted in lower concentrations of the most involatile species, the 20-carbon accretion product. Simulated [HOM] were also used to probe the importance of various particle nucleation mechanisms. In Finland, pure biogenic nucleation mechanism (i.e. nucleation without sulphuric acid) at low altitude was predicted to be responsible for around 60% of new particle formation in the lowest 500 m in the pre-industrial atmosphere and around 20% in the present day, indicating the importance of HOMs, particularly in the pre-industrial, with implications for aerosol burden and climate. PBN was predicted to be less important in the Amazon given the lower simulated [HOM].

CRI-HOM can provide a framework for simulating HOMs in global chemistry-aerosol-climate studies and simulating the effects of isoprene-driven suppression of involatile biogenic-derived species and the consequences on SOA and NPF while also providing a state-of-the-art description of atmospheric chemistry. Such an effect, and the influence on the proposed "BVOC negative feedback", is likely to be important in a warming climate with enhanced emissions of BVOCs and determining the size and sign of the feedback. Once incorporated into a global chemistry-aerosol-climate model, assessing the effect of HOMs on pre-industrial, present day and future climate will be key area of future work.

*Supplement.* The supplement related to the article is available online.

*Financial Support.* JW is funded by a Vice Chancellor's Award from the Cambridge Trust. We would like to thank the Cambridge-LMU Strategic Partnership for supporting collaborations with the BOXMOX model. We would like to thank NERC, through NCAS, and the Met Office for the support of the JWCRP UKCA project. SA and ATA would like to thank NERC PROMOTE (NE/P016383/1). HG is supported by the NASA ROSES Atmospheric Composition Modeling and Analysis Program under grant number 80NSSC19K0949.

*Data Availability.* All modelled data is available upon request from James Weber and all experimental data from Torsten Berndt. The KPP files for the CRI-HOM mechanism have been deposited in the University of Cambridge data repository and can be viewed at doi.org/10.17863/CAM.54546."

*Author Contributions.* Mechanism development was done by JW, ATA, ME, SA, modelling experiments were designed and executed by JW, ATA, PG, HG, CK and flow cell data was compiled and interpreted by TB. JW and ATA wrote the paper. All co-authors discussed the results and commented on the paper.

*Competing Interests.* All authors declare that they have no competing interests.

§

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

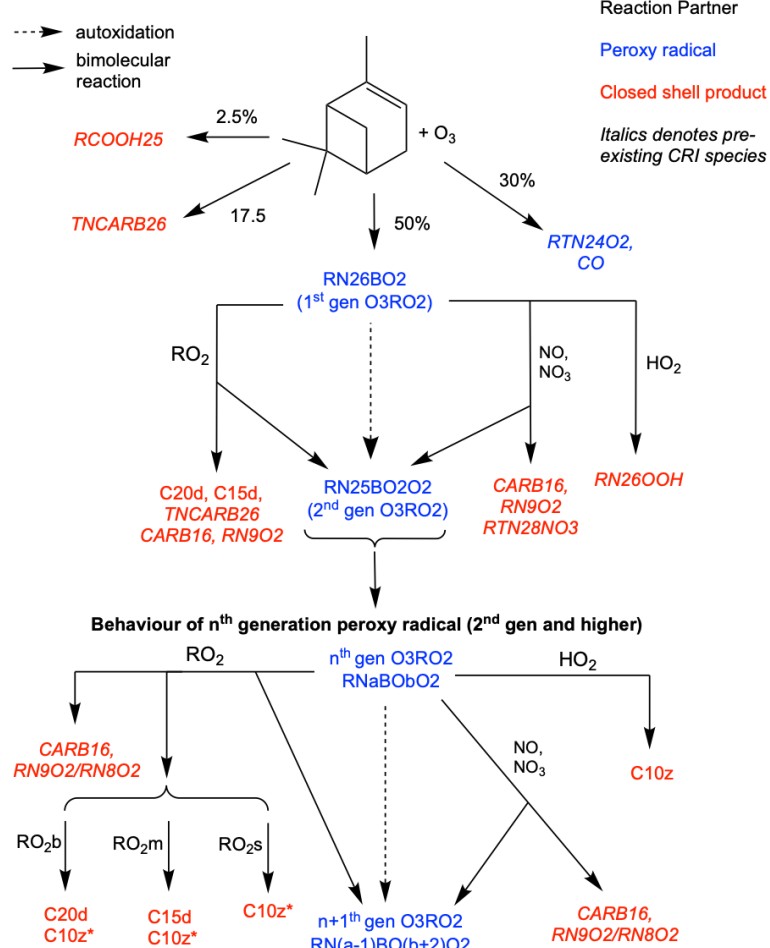

Figure 1: **Schematic of additions to CRI v2.2 to represent autoxidation and HOM formation via ozonolysis of α-pinene. HOMs (C10z, C15d, C20d) can be produced via reaction of the O3RO2 with HO₂ and RO₂ while reaction with NO, NO₃ and RO₂ can produce alkoxy radicals which can fragment or isomerise. New species introduced in this scheme are denoted by normal font, existing species by italics. The RO₂ pool is split into subsections covering big (RO₂b), medium (RO₂m) and small (RO₂s) peroxy radicals to facilitate addition of accretion product formation.**


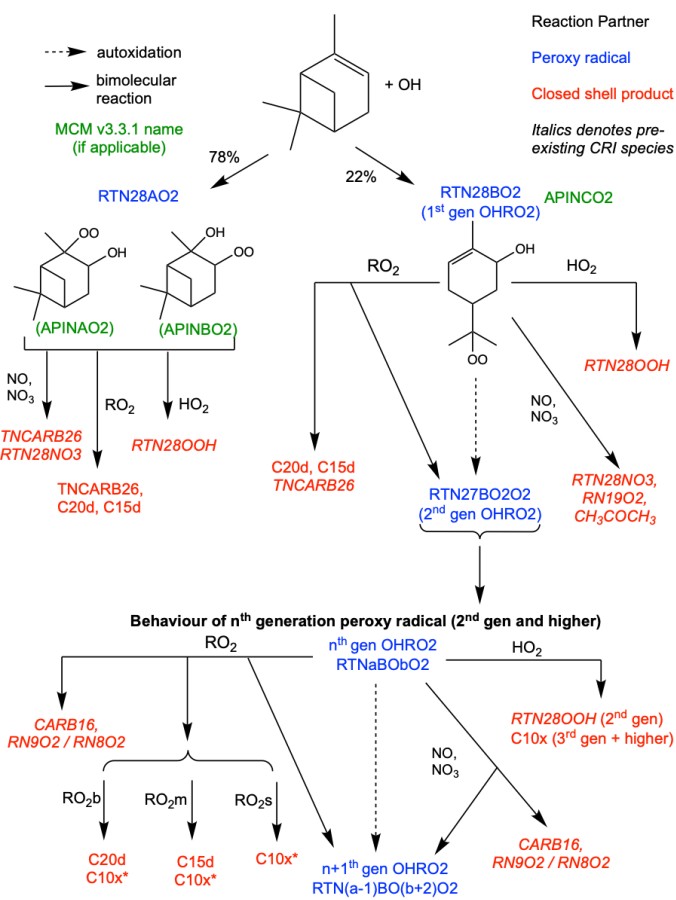

**Figure 2 - Schematic of additions to CRI v2.2 to represent autoxidation and HOM formation via OH oxidation. HOMs can be produced via reaction with HO₂ and RO₂ while reaction with NO, NO₃ and RO₂ can produce alkoxy radicals which can fragment or isomerise. The RO₂ pool is split into subsections covering big (RO₂b), medium (RO₂m) and small (RO₂s) peroxy radicals to facilitate addition of accretion product formation.**

**Table 1 - Summary of new species added in HOM mechanism. The removal of species RN18AO2 and RTN28O2 results in a net increase of 12 species.**

| Species | Classification | Origin | MCM v3.3.1 equivalent |
|---|---|---|---|
| RN26BO2 | 1st generation peroxy radical | Ozonolysis | C107O2, C109O2 |
| RN25BO2O2 | 2nd generation peroxy radical | Ozonolysis | Not in MCM |
| RN24BO4O2 | 3rd generation peroxy radical | Ozonolysis | Not in MCM |
| RN23BO6O2 | 4th generation peroxy radical | Ozonolysis | Not in MCM |
| RNxBOyO2 | Lumped 5th and higher generation peroxy radical | Ozonolysis | Not in MCM |
| RTN28AO2 | 1st generation peroxy radical (no autoxidation) | Ozonolysis | APINAO2 and APINBO2 |

| RTN28BO2 | 1st generation peroxy radical (autoxidation possible) | OH oxidation | APINCO2 |
|---|---|---|---|
| RTN27BO2O2 | 2nd generation peroxy radical | OH oxidation | Not in MCM |
| RTN26BO4O2 | 3rd generation peroxy radical | OH oxidation | Not in MCM |
| RTNxBOyO2 | Lumped 4th and higher generation peroxy radical | OH oxidation | Not in MCM |
| C10z | 10-carbon HOM | Ozonolysis | Not in MCM |
| C10x | 10-carbon HOM | OH oxidation | Not in MCM |
| C15d | 15-carbon HOM | Ozonolysis and OH oxidation | Not in MCM |
| C20d | 20-carbon HOM | Ozonolysis and OH oxidation | Not in MCM |

**Table 2 - Summary of possible products formed for a particular peroxy radical reacting with the big, medium and small peroxy**
**radical pools.**

| Size of reacting peroxy radical | Product of reaction with big pool | Product of reaction with medium pool | Product of reaction with small pool |
|---|---|---|---|
| Big (>7 C, e.g. RN26BO2) | C20 HOM accretion product C10 HOM / Existing CRI species* Peroxy radical | C15 HOM accretion product C10 HOM / Existing CRI species* Peroxy radical | C10 HOM / Existing CRI species* Peroxy radical |
| Medium (4-7 C, e.g. RU14O2) | Closed shell, alkoxy radical | | |
| Small (<4 C, e.g. CH3O2) | Closed shell species, alkoxy radical (no change from CRI v2.2 treatment) | | |

* The result depends on the extent to which the reacting peroxy radical has been oxidised prior to the $RO_2$-$RO_2$ reaction. A HOM is classified as a species which has undergone at least one autoxidation step at atmospherically relevant temperatures and contains at least 6 oxygen atoms (Bianchi et al., 2019). Thus, some of the less oxidised peroxy radicals may not qualify as
HOMs and are assigned to the most relevant non-HOM species already in the CRI.


§

**Table 3 - Simulations used for developing and testing new mechanism**

| Simulation | Purpose | Mechanism Version(s) Used | Conditions |
|---|---|---|---|
| A: Flow cell experiment | Optimise mechanism by fitting autoxidation coefficients and rate coefficients for accretion product formation | Temperature independent mechanism | 297 K, $NO_x < 10^8$ $cm^{-3}$, dark 26 ppb $O_3$, initial α-pinene concentration varied |
| B: Flow cell experiment | Along with Simulation A, optimise mechanism by fitting autoxidation coefficients and rate coefficients for accretion product formation | Temperature independent mechanism | 297 K, $NO_x < 10^8$ $cm^{-3}$, dark 80 ppb $O_3$, 15.6 ppb α-pinene concentration, initial isoprene concentration varied |
| C: Chamber Experiment | HOM yield calculation | Temperature dependent mechanisms with autoxidation activation energies of 6000K, 9000K and 12077K | 270 K, 290 K or 310 K α-pinene 15 ppb, $O_3$ 40 ppb NO, $NO_2$ varied from 0.01-10 ppb |
| D: Tropical Boundary Layer Experiment | Compare new mechanism with concentrations predicted by CRI v2.2 | Temperature independent and all temperature dependent mechanisms | 8 day run with diurnally varying photolysis, temperature (298±4 K), α-pinene and isoprene emissions Multiple runs performed with scaled NO and α-pinene emissions (Full details in SI) |


**Table 4 - Autoxidation coefficients for peroxy radicals after fitting to experimental data (at 297 K) with estimated uncertainty.**

| Generation | O3RO2 / $s^{-1}$ | OHRO2 / $s^{-1}$ |
|---|---|---|
| 1st | 0.206 (+0.025/ -0.04) | 2.1[a] |
| 2nd | 1.7 (+1.1/-0.4) | 2.1 (+1.6 / -0.2) |
| 3rd | 1.7 (+1.1/-0.4) | 0.25 (+0.3 /-0.1) |
| 4th | 1.6 (+0.8/ -0.5) | N/A |

[a]Taken directly from Xu et al (2018)


§

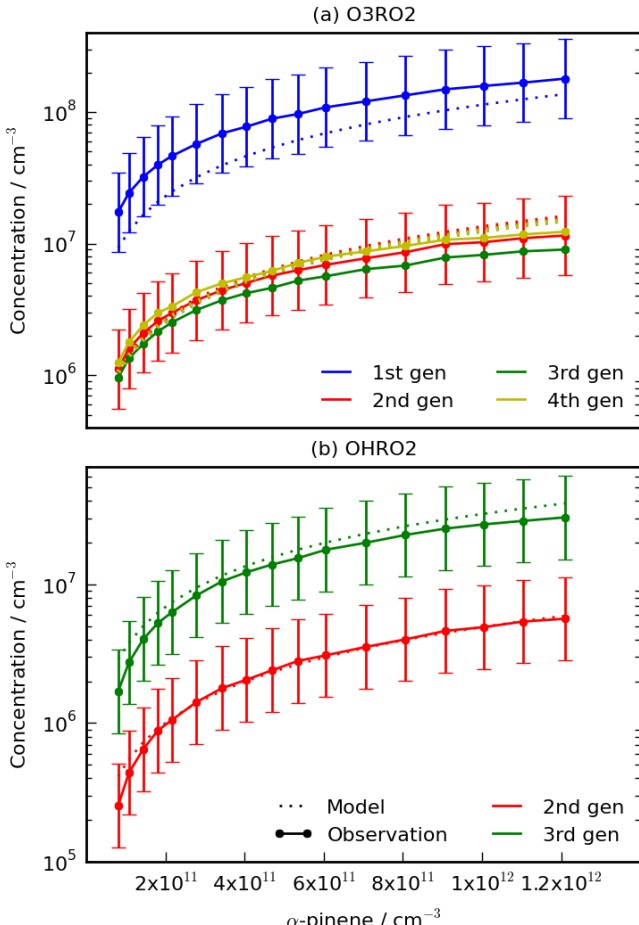

**Figure 3 - Comparison of the HOM-precursors (a) O3RO2 and (b) OHRO2 produced by the model and from Berndt et al (2018b) for experiments performed with different initial concentrations of α-pinene (Simulation A). The model reproduces the increase in O3RO2 and 2nd and 3rd generation OHRO2 with initial α-pinene well.  The model struggled to reproduce concentrations of the 1st generation OHRO2 (not shown).  Note that the error shown is the experimental error from Berndt et al (2018b) and the error bars for the 3rd and 4th generation O3RO2 species are of very similar size to the error bars of the 2nd generation species but have been omitted for clarity.**


§

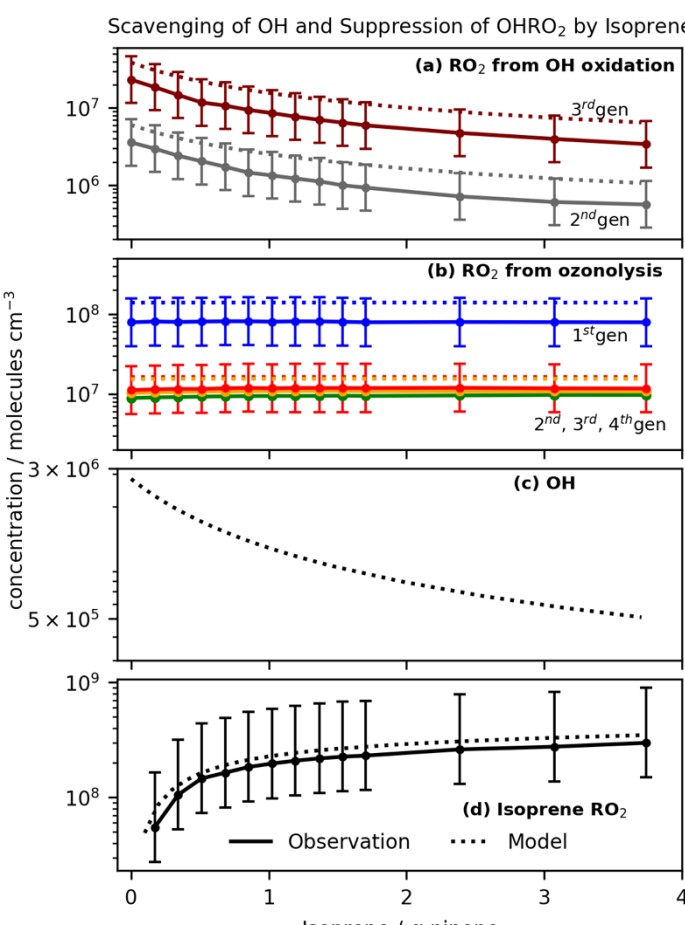

**Figure 4 –** **Observed and modelled variation for Simulation B of OHRO2 (a), O3RO2 (b), OH (c) and the peroxy radical formed from isoprene oxidation (d) with increasing isoprene (observed data from Berndt et al (2018b)). The model is able to reproduce the decrease in OHRO2 as well as their concentrations. The fractional decline of OHRO2 mirrors that observed in the OH concentration, suggesting the major driver is OH scavenging. The error shown is the experimental error from Berndt et al (2018b).**


§

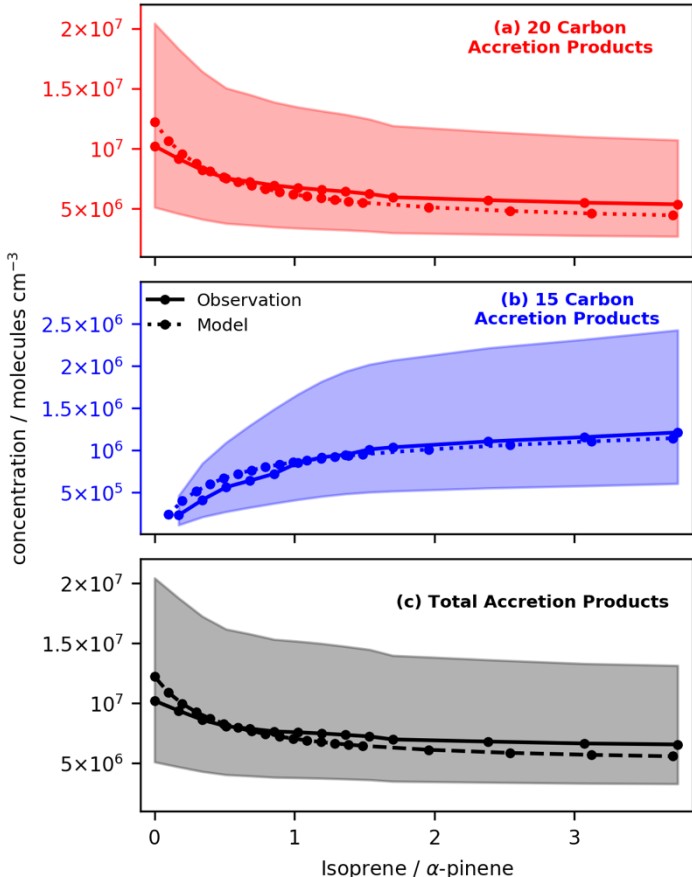


**Figure 5 – Variation in Simulation B of observed and modelled concentrations of C20d (a), C15d (b) and total accretion products (c) with isoprene at fixed initial concentrations of α-pinene (observed data from Berndt et al (2018b)). The modelled data falls within the experimental uncertainty shown by the pale red, blue and grey regions. The model reproduces the observed decrease in C20 accretion products and increase in C15 accretion products well. Furthermore, the model reproduces qualitatively the result observed**

**by McFiggans et al (2019) that addition of isoprene reduces the total accretion products concentration with potentially important implications for total aerosol burden and particle size distribution.**

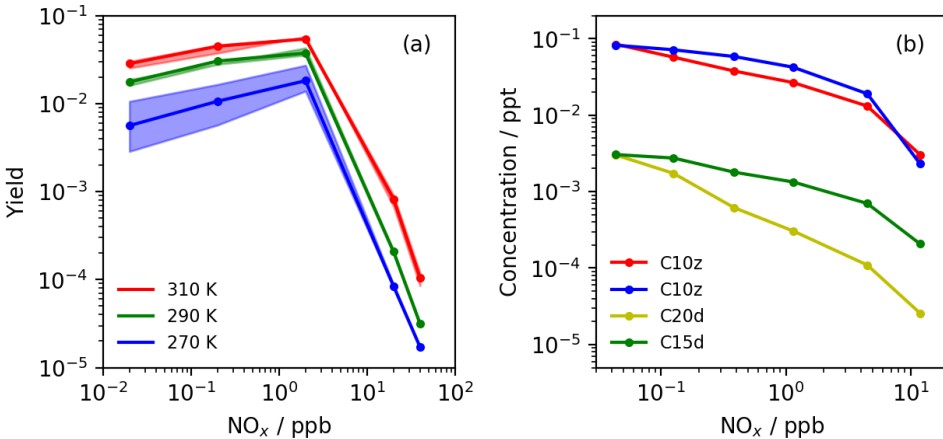


**Figure 6 – (a) Maximum modelled HOM yields (C10x and C10z) exhibiting significant decline under high NOₓ conditions (Simulation C). The spread in the modelled yield between HOM mechanisms (HOM$_{6000}$, HOM$_{9000}$ and HOM$_{12077}$), shown by the shaded regions, indicates the lower sensitivity to autoxidation activation energy at temperatures above ~290 K. (b) Observed HOM concentrations from 8 day tropical PBL run (Simulation D) showing decrease in concentration for all HOM species with NOₓ. Under tropical PBL**

**conditions, negligible difference was observed between HOM mechanisms due to daytime temperatures exceeding 300 K.**

**Table 5 - Observed and modelled concentrations after adjustment of model input. The model performs well in comparison to observed [HOM] at Hyytiala.**

§

| Location | Species Adjusted to Observations | Mean observed concentrations at relevant time of day / ppt[c] | Model concentrations with input adjusted to observations / ppt[e] |
|---|---|---|---|
| Hyytiala[a] | Monoterpene | 10-carbon HOM: 0.2-0.8 (mean 0.4 ) 20-carbon accretion product: 0.04-0.16 (mean 0.08 ) | 10-carbon HOM: 0.75-0.85 20-carbon accretion product: 0.28-0.30 |
| Alabama[b] | Monoterpene, Isoprene, OH, O3 | C9 & C10[d]: 30 | C10[f]: 4.5-13.3 |

[a]Roldin et al., 2019, [b]Lee et al., 2016, [c]Ranges given accounts for factor of 2 uncertainty in observed concentrations, [d]Includes concentrations from C9 species (C9H14-20O4-10) and C10 species ($C_{10}H_{16-22}O_{4-10}$), [e]Model was run with surface conditions in May for comparison to data from Roldin et al (2019). [f]Range arises from model runs using range of CS values suggested in Lee et al (2016)

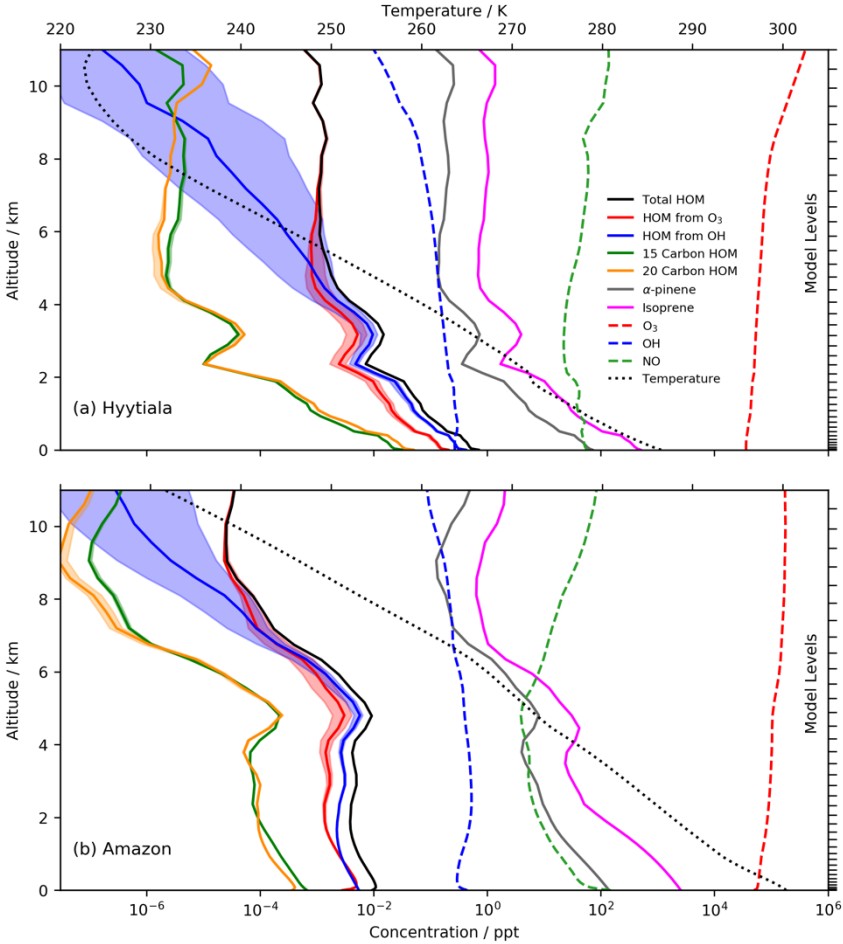

**Figure 7 - HOM profiles in June (2pm LT) above (a) Hyytiala and (b) the Amazon near Manaus. Despite higher [α-pinene], the significantly higher CS in the Amazon results in lower [HOM]. The temperature dependence (shown by the shading) is more significant at low altitude with Hyytiala's cooler temperatures. The Amazon's higher isoprene/ α-pinene ratio (~20) resulted in greater suppression of the 20-carbon accretion product than at Hyytiala.**

§

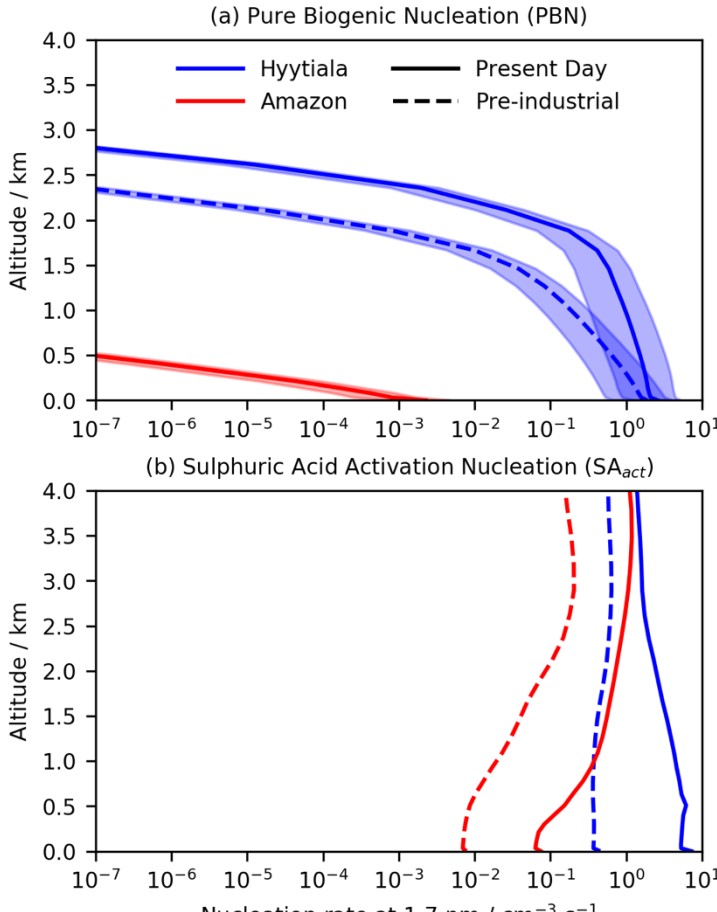

**Figure 8 - PD and PI modelled nucleation rates averaged over June for (a) summed (neutral + ion-induced) pure biogenic nucleation (assumes altitude-independent ion production rate (IPR) of 2 cm$^{-3}$ s$^{-1}$ (Hirsiko et al., 2011), shading shows IPR variation 0.5-5 cm$^{-3}$ s$^{-1}$) and (b) nucleation from sulphuric acid only (SA$_{act}$). Both mechanisms are predicted to produce greater nucleation rates in the PD due to greater concentrations of precursor species. Importantly however, PBN at low altitude at Hyytiala is predicted to be comparable to SA nucleation in the PI due to the greater modelled ion concentration arising from a lower condensation sink and reduction in rates from SA$_{act}$ due to lower sulphuric acid concentrations. This leads to a larger increase in the total nucleation rate in the PI than the PD with potential implications for PI aerosol burden and climate.**