# Peer review of "CRI-HOM: A novel chemical mechanism for simulating Highly Oxygenated Organic Molecules (HOMs) in global chemistry-aerosol-climate models."

_Atmospheric Chemistry and Physics, 2020_

## Referee Comment (RC1) · Anonymous Referee #1 · 1 Apr 2020

The article demonstrates the implementation of a reduced HOM mechanism into a reduced mechanism, CRI, which was derived from MCM3.3.1. HOM were discovered a few years ago. They are supposedly formed by a fast process called autoxidation. Since from their structure HOM and accretion products have low to extreme low vapor pressure, they are important candidates for SOA formation and persistence. This links HOM to relevant issues related to aerosol effects, e.g. for climate warming. The purpose of the presented work was to provide a simple enough version of HOM formation mechanism within a CRI for implementation in larger models.

[Figure]

The authors describe quite clearly the steps of the implementation with respect to the changes in the CRI. Their attention turned to the implementation of autoxidation in competition to bimolecular termination reactions. In addition they put efforts also in the implementation of HOM accretion products, as those may play a role in atmospheric nucleation. HOM accretion products are supposedly formed by recombination reactions of (HOM) peroxy radicals. By splitting the sum of peroxy radicals in three classes and considering class interactions they enabled a treatment of the recently described interaction of HOM peroxy radicals with low molecular weight peroxy radicals, specifically the interaction of a-pinene and isoprene. They further implemented the temperature dependence of HOM formation as recently observed. Aspects of their improvement of the CRI – HOM mechanism were tested against flow tube experiments by Berndt et al., with satisfying results. These parts of the work are very interesting and important.

The new CRI-HOM was then used to calculate vertical profiles over two stations in Hyytiälä and Manaus. Here the descriptions of the results are not fully congruent with the graphical representations. Overall, the vertical profiles of HOM and HOM accretion products are however in a reasonable range. That means within general experience it could be possible. Any verification of the height profiles by observation is missing. Things become even more speculative when the authors try to predict PI and PD nucleation and the role of biogenics and HOM. I am not sure if that part is really helpful. They authors admit that any validation is currently impossible (in the first sentence of the according paragraph). Depite of the latter, overall the article is timely, well written and can be published in ACP after addressing the comments below.

Comments:

line 53: Bianchi et al., is a review; here you reference to the original papers in order to give the authors the credits

line 199f: I think it must be RTN24O2, instead of RTN24BO2. Or it must be "RTN26BO2 - RTN23BO2". Or something else in the nomenclature is not consistent in this paragraph.

line 204 and line 236: If you want to make the mechanism efficient, why do you start lumping O3-HOM from the 5th generation, while you starting for OH HOM with the 4th generation?

line 219f: "Alkoxy radicals are not represented explicitly due to their rapid reactions which, typically for larger peroxy radicals, are decomposition or isomerisation. " The statement doesn't make sense to me; alkoxy instead of peroxy?

line 339: How much effort would it be to implement and to test a range lifetimes typical for sticky molecules applied to your HOM species. Isn't deposition always a week point in atmospheric models? If deposition is faster than upward transport, your vertical profiles would be obsolete.

line 403-409 and Table 4: If I understand the autoxidation process correctly, the autoxidation rate must slow down at the end as suited H atoms are already consumed in previous autoxidation steps. This is the case for OH, but not for ozone? The overprediction of the highest generations compared to Berndt et al. is not necessarily only due to missing loss processes, it can be also due to overestimated source strength, by your last step autoxidation rate coefficients.

line 426-429: Overpredicting the first generation of OHRO2 has nothing to do with the HOM-mechanism, correct? From this point of view it may be a severe principal failure of your CRI scheme. What could be reasons for that? This should be discussed a little more extensively.

line 467: In section 4, I do not understand the selections of sites for comparison. Why compare Alabama with Southern Finland at the ground, but calculating vertical profiles over Southern Finland and Amazonia. You should compare with the Manus ground data, too. Moreover, there were big campaigns over Amazonia and Finland, also with airplanes. Can't you use data to validate at least parts of your vertical profiles, e.g. OH,

O3, NOX, a-pinene, isoprene, selected OVOC?

line 589: Do really mean "semi-qualitative". That would implicate not even qualitatitve...? The profiles look quite reasonable, overall. And you highlight features of the profiles...

line 592-594: What is the sense of the comparison then (see comment above)? Can't you split off from the observations the compounds which are in your model?

line 596-602 and Figure 7: In contrast to your statement, the OH data decrease with height and O3-HOM and OH-HOM do not have the same share there anymore. Why do the OH-HOM decrease in upper troposphere? It would be helpful to show vertical profiles for OH, O3 and possibly NO, too. Moreover, in legend of Figure 7 you describe features (arrows) which I cannot see. And the color code for O3-HOM is brown, while the line is orange.

line 672: There were no vertical profiles over Alabama, right?

Captions, Figure 1 and 2: I suggest to introduce the meaning of RO2s, m, b also in the captions.

Typos, errors:

line 70: reference Sindelarova et al., 2014, is missing in the reference list

line 141: reference Kiendler-Scharr, instead of Kiendler-Scherr

line 150-160: in the reaction equation 3 and 4: it should be C5RO2 instead of C10RO2

line 245: in equ. (9) 10x instead of 10z

line 274: Jenkin et al. 2019, a or b missing

line 287: m instead of me

line 306: than instead of that

line 625: you call the underestimation of H2SO4 slightly, I see 1-2 orders of magnitude

There are frequently passages in the text using a different font size. e.g. lines 57/58, line 378/379, line 421, line 422, line 458/459, line 464/465, line 561

Table 1: "gen."

Table 3: needs reformatting of the text fields. . .

In general, some page formatting issues with Figures and Tables.

Figures S17, S18: The subscripts are too small.

---

## Referee Comment (RC2) · Anonymous Referee #2 · 6 Apr 2020

General comments:

This manuscript describes the development of a new mechanism module for the simulation of highly oxygenated molecules (HOMs) from the oxidation of alpha-pinene by ozone and OH in the gas phase. Such HOMs have recently been demonstrated to contribute substantially to new particle formation in certain environments, and (highly uncertain) measurements are beginning to constrain the dependence of HOM formation on temperature, NOx, the presence of other peroxy radicals, and other parameters. By developing a mechanism that fits easily within a larger mechanism already in use

in models (the Common Representative Intermediates or CRI mechanism), the study described herein will hopefully provide a precursor from which further research efforts can both fine-tune the mechanism and apply it to broader investigations of HOM impacts on atmospheric chemistry and climate. The mechanism adds 12 species and 66 reactions to CRI, which is both reduced from the full complexity of the HOM system and retains the most salient aspects of HOM formation and its chemical dependencies. The authors first describe the construction of the mechanism in detail, including the sources of and assumptions made about certain parameters (e.g. branching ratios and rate constants), after which they discuss the tuning of the various unknown mechanism parameters for optimization against laboratory experiments. Finally, they demonstrate the applicability of their mechanism by using it to model the role of HOMs in the ambient environment, simulating alpha-pinene chemistry in vertical profiles over the Finnish boreal forest and the Southeast United States

The manuscript represents a valuable addition to the recent surge in literature reports regarding HOM formation from monoterpenes, as it synthesizes inputs from various sources into a modular chemical mechanism that can be applied to further research into the role of HOMs in the atmosphere. However, with an eye toward those future projects to which this mechanism will surely make an important contribution, it is imperative that the authors make clear the remaining uncertainties, sensitivities, and assumptions inherent to the key parameters and output of this CRI-HOM mechanism. The mechanism is an important first step, but in order for future fine-tuning to be conducted, it will be necessary to not just acknowledge but to actively advertise the aspects of the mechanism that remain most uncertain. This will give the critical dialog between models, observations, and laboratory studies room to improve this mechanism as new constraints become available. To that end, I think a number of efforts could be made in this manuscript to clarify the sources and magnitudes of uncertainty, the origins of certain specific assumptions, the sensitivity of mechanism parameters (e.g. branching ratios, rate coefficients) to assumptions made, and the ranges of parameter values that would be consistent with the limited and/or highly uncertain HOM and RO2 observa-

tions.

As has already been alluded to, the mechanism relies on a number of assumptions and extrapolations between species for many branching ratios and reactions rates. That in itself isn't bad and doesn't invalidate any of this, but requires careful attention to the sensitivity to those assumptions and the resulting uncertainty in the mechanism's parameterizations and output. I would hate to suggest running more models; instead I think (a) some of this sensitivity analysis already exists in the SI (e.g. L 364-366) and should be given a more prominent billing; (b) some parameters and model output could benefit from bootstrap back-of-envelope calculations (or comparisons to previous literature, as I suspect exists for e.g. the first-generation pinene ozonolysis yields) as to their sensitivities to certain inputs or ranges that would be consistent with observations, and (c) descriptions of the mechanism would benefit from more careful attention to what can be stated with certainty and what results of the simulations are so sensitive to highly uncertain numbers that they can't be considered conclusive. Along these lines, see especially the comments to L222-224, 304-314, 341-347, 507-508 below.

The large uncertainty bounds on measurements to which the mechanism was compared (as noted in L 428-429) suggests a need for reporting a range of mechanistic parameterizations consistent with the measurements, rather than single values. The kind of analysis you do on L 364-366 of the SI is hugely useful for these purposes, and should be incorporated into the main text (and, as mentioned below, the ranges and sources reported in Table S6 would also be useful in the main text). However, these sensitivity studies could benefit from more detailed descriptions. It sounds as though the uncertainties were only estimated by changing one rate at a time and comparing the resulting changes in the concentration of the peroxy radical in question with the upper and lower bounds of the experimental uncertainty. This would neglect any compounding effects from simultaneous changes in multiple autoxidation coefficients, or in both autoxidation coefficients and alkoxy decomposition : isomerization branching ratios, correct? This should be acknowledged (or, if possible without too much additional

work, the cumulative uncertanties could be estimated and reported). Also, if similar sensitivities exist to show the range of HOM yields from RO2-RO2 chemistry that is consistent with the large uncertainty bounds shown in Figure 5, those would be useful to see as well.

Finally, it's somewhat unclear without reading the whole paper and SI very carefully what branching ratios / rates are fit, which are plugged in from measured values or extrapolated from similar species, and which are educated guesses. Someone wanting to use this mechanism or adapt it for their own uses might want to know which coefficients are flexible and which are most tightly constrained (or measured). Can a quick representation of that be provided? Either as an expanded Table 4 / Table S6 (with, say, a superscript character on each rate or branching ratio to denote which come from what sources) or additional annotations to figures 1 and 2 that make it clear (e.g. colour-coded arrows corresponding to which rates and branching ratios come from which sources).

Specific comments:

L 154-155: For mass conservation and to fit with the explanation in L 145-147, I assumer reactions 3-4 are supposed to have C10RO2 + C5RO2 as the reactants?

L 194-195: Figure 1 implies that these TNCARB26 and RCOOH25 co-products, along with the major products of RN26BO2 and RTN24O2, are formed in fixed yields from pinene ozonolysis. However, this sentence (and my understanding of Criegee intermediates) would imply that the branching ratios to these products depends on the relative abundance of the Criegee intermediates' reaction partners, such as water. Could you clarify here and/or in the caption to Figure 1 whether/how this Criegee chemistry is represented, and whether it matters?

L 199: This sentence refers to "RTN24BO2", but the co-product in Figure 1 is "RTN24O2".

L 222-224: Are there any constraints or uncertainty bounds on this 50:50 ratio of decomposition and isomerization pathways used here? It seems that in environments where reaction with NO is competitive with the other reaction pathways (and even in very low-NO conditions, since the alkoxy radicals are formed in RO2-RO2 reactions as well), the resulting HOM yield could be highly sensitive to this branching ratio. In the absence of concrete evidence for these specific alkoxy intermediates' branching ratios, can you provide some estimate of the sensitivity of your mechanism's output to the chosen ratio? This would be useful either here or later, when you describe the important of this NO-derived HOM in the context of the model output (∼L 505-511)

L 244-246: Should equation 9 (and the line preceding it) read C10x or C10z? I realize the difference is described in Table 1, but it would be helpful to spell it out in the text as well so the reader doesn't get confused on this point.

L 283-294: Either this semicolon is meant to be a colon, or something's missing in the description of the UCARB10/UCARB12 products that would turn it into a standalone clause.

L 287: Does this "me" mean medium? If so, doesn't this contradict the statement (previous page, L272-3) that medium peroxy radicals react individually with each peroxy radical pool too?

L 304-314: The citation of MCM implies that k14 and k15 were derived from Jenkin et al. 2019a, but the citation two lines later implies they are from Molteni et al. 2018. Which were they? Also, this presents another fixed input in the mechanism to which the model output and parameter fitting might be highly sensitive. Are there uncertainty bounds on the rate coefficients from Molteni et al. or Roldin et al. 2019 that can be used to estimate this sensitivity? Are there constraints on the 50:50 branching between R14 and R15, or any particular reason to have chosen that branching? Can any estimates be made of the mechanism's sensitivity to this branching?

L 325 & L309: You report a range of chosen fitted values for k13 and k16. Is this range

of different values for the different peroxy radicals within each group, or does it represent some sort of uncertainty? Referring forward to section 3.1.3 as suggested here does not clarify the fitting procedure. The repeated use of "chosen" and "assigned" sounds more like the values were user-selected out of a pre-defined range rather than fit to data, and the model output then compared favourably to the highly uncertain aggregate HOM measurements (which, as far as I can tell, don't distinguish between the generation in which the HOM was formed). How was the initial range over which to "fit" chosen?

L 321: Why doesn't R16 conserve mass the same way R13 does?

L 341-347: What is the rate coefficient for the reaction of RTN28OOH + OH (and was it measured?), and why is this the one to stand in for the HOMs? Where did the MCM photolysis frequencies come from – were they measured or also extrapolated from other species? Again, are there any estimates of the uncertainty in these photolysis/reactions rate coefficients or the sensitivity of the mechanism to them? While your reasoning that their gas phases losses are unlikely to affect OH or O3 seems reasonable, the loss rates should be very important to new particle formation in some circumstances.

L 392-397: As mentioned in the comment above to L 194-195, couldn't these yields be variable instead of static, and dependent on the environmental conditions that affect the branching pathways of the Criegee intermediates? Numerous past experimental efforts have quantified product yields from a-pinene ozonolysis; are these branching ratios consistent with those past efforts? With what certainty is the 0.206 s-1 autoxidation rate coefficient known, and how sensitive are all the subsequent steps in this mechanism to it?

L 404-406: This sentence seems grammatically incomplete. Perhaps the "and" isn't needed?

L 406-408: This is a crucial point for the present study, and one that I don't think should

be so quickly discarded. How (i.e. to what extent) would this change the autoxidation coefficients?

L 416-418: This type of sensitivity study is very helpful for understanding the strength of the constraints on your mechanism parameters, but I am confused by the wording of "realistic deviations". Are these the maximum deviations consistent with the autoxidation coefficients, or the maximum deviations consistent with the experiments? If the former, do we have reason to believe that the experiments fell within this range? And if the latter, how were they deemed "realistic"?

L 426: Should this refer to Fig. 3b instead of a?

L 428-429: Why are these measurement believed to be so drastically underestimated? Whether or not they are, the results suggest that some element of the first-generation OH-derived peroxy radical chemistry is substantially biased in the mechanism; do you have any indication of what this might be?

L 490: This sentence mentions 4 mechanism versions, but the rest of the paragraph only seems to describe 3.

L 507-508: See comment above on lines 222-224: this NOx-dependent behaviour derives from a highly uncertain 50:50 alkoxy radical decomposition : isomerization branching ratio, but is described here as an important consequence of the mechanism. Would this behaviour hold true for a range of reasonable estimates of the branching ratio? How sensitive is it to the chosen branching?

L 516: Are the model uncertainty ranges reported here for the different temperature dependences? It's very important to distinguish this from some sort of total uncertainty estimate.

L 518: Missing a word in "comparing favourably to yield measured by" ?

L 527: What explains the drastic decrease in the simulated acetone production?

L 571: "We also at the effect" ... something's wrong here.

L 588: What does it mean for a result to be semi-qualitative? That even some non-quantitative qualities of the simulated vertical profiles are expected to be erroneous? What ones?

Table 4: The uncertainties and sources of these rates are important enough that I think Table S6 should be combined into Table 4

Figure 3: Why are there no error bars on the 3rd and 4th generation O3RO2 observations? Where do the error bars come from (e.g. are they instrument uncertainty on the measurements or something else)?

L 979-980: While the note that first-generation OHRO2 are poorly reproduced is appreciated, it seems misleading not to put them on the graph, and deprives the reader of a visual representation of this important element of the mechanism. Are the concentrations too high to fit on this graph?

Figure 5: Can the y-axes be adjusted to show the reader the extent of the measurement uncertainty? The large uncertanties suggest that a wide range of HOM yields would be consistent with the observations, including some yield parameterizations that wouldn't display the much-heralded decrease in C20 (and total) accretion products and increase in C15 products. Were any sensitivity estimates made regarding HOM yields in the mechanism that could be shown here?

---

## Author Response (AR1)

**Response to Reviews of "CRI-HOM: A novel chemical mechanism for simulating Highly Oxygenated Organic Molecules (HOMs) in global chemistry-aerosol-climate models" by Weber et al.**

We are very grateful to both reviewers for their comments and efforts which have helped us improve this manuscript. Following the structure recommended by ACP, we have responded to each reviewers' comments sequentially below with italicised text showing the reviewer's comments and plain text showing our response. Text which has been added to the manuscript is coloured red. Original manuscript text is in blue and any text which has been removed from the manuscript is blue and has been struck through. The locations of changes are stated. We hope these revisions address the concerns of the reviewers.

While responding to the reviewers' comments, three errors were discovered. These have been corrected and are detailed below. While the corrections are important, we do not believe any of the errors diminish the validity of the conclusions drawn.

**Correction to yield calculation**

While working on the response to the reviewers' comments, a minor error in the calculation of the yield was discovered. In brief, the correction reduces the HOM yield but it still remains within the range of experimentally-measured values. Importantly, the sensitivity to $NO_x$ and temperature remain unchanged. This error has been corrected and resulted in changes to Figure 6 and the corresponding text. While the correction of this error is important, it does not change the conclusions of the paper. The results of the other simulations performed with the CRI-HOM mechanism were not affected and so the strong performances of the model against observations in Alabama and Hyytiala remain valid. The response to the reviewer comments have been written factoring in the correction to the HOM yield. To address this correction in the manuscript, the following changes have been made.

Figure 6(a) has been updated.

The sentence beginning on line 24 has been amended to read:

The mechanism predicts a HOM yield of  2-4.5% under conditions of low to moderate $NO_x$, in line with experimental observations, and reproduces qualitatively the decline in HOM yield and concentration at higher $NO_x$.

The paragraph starting line 516 has been amended as follows.

The model predicted total HOM yields at 290 K of 1.9±0.2% (0.01 ppb NO) to 3.9±0.5% (1 ppb NO) with the quoted range resulting from the range of temperature dependencies considered. This is within the ranges previously suggested by Jokinen et al (2014) (1.7-6.8%) and close to the values from Ehn et al (2014) (3.5-10.5 %)  and Sarnela et al (2018) (3.5-6.5%) while lower than Roldin et al (2019) (7%)  In addition, the HOM yield at 270 K of ~ 0.6-1.9% compared favourably with the yield

of ~2% determined by Roldin et al (2019). This suggests that the mechanism is doing a good job at simulating HOM yield. The slight low bias may be in part due the values of k14 and k15 which were shown to influence the HOM yield relatively strongly. Sensitivity tests involving universal doubling and halving of the rate coefficients produced HOM yield changes of around +65% and -40% respectively (Fig. S4(a)) while preserving the general dependencies on $NO_x$ and temperature (Table S6). This area of uncertainty will be the focus of future work.

**Correction to Model Surface Concentrations**

An error in the code used to calculate photolysis frequencies for May as used in the calculation of surface concentrations at Alabama and Hyytiala (Table 5) was discovered. This error did not affect the calculation of the code for June which was used for the calculation of vertical profiles and nucleation rates (Figs. 7, 8, S23). Correcting this error resulted in changes to the modelled surface concentrations in both locations.

In Alabama the modelled surface concentrations changed to 4.5 - 13.3 ppt (compared to 8.1-12.1 ppt before correction). While this is a large relative change, the only conclusion drawn in the text from the comparison to the observational data from Alabama (30 ppt) is that the model returned a "reasonable value" with more detailed comparison not possible due to the wider range of species included in the observational data. We believe this conclusion is still valid.

In Hyytiala, the modelled concentrations changed to 0.75-0.85 ppt for 10-carbon HOM (0.33-0.37 ppt previously) and 0.28-0.30 ppt for 20-carbon HOM (0.20 ppt previously). The 10-carbon HOM concentration remains within the mean observational range (0.2-0.8 ppt) and the 20-carbon HOM is now around double the mean observed maximum (0.16 ppt) but still lower than the maximum concentrations observed of 0.7 ppt. Therefore, we do not believe these new results invalidate our conclusions that the model "*mechanism compares favourably to some of the limited observations of [HOM] observed in the boreal forest in Finland and in the south east USA*". Table 5 has been updated to include the new values and the the following amendments has been made to line 590:

"In the boreal forest in Hyytiala, the range of predicted 10-carbon [HOM] falls at the higher end of the mean observational range is close to the mean observational valueand well below the maximum observed concentrations (1-1.5 ppt) (Roldin et al., 2019). and Tthe predicted 20-carbon accretion product concentration is around double slightly above the mean observational range and well below the maximum observed values (0.6-0.7 ppt)."

**Correction to mass conservation in the mechanism**

Several reactions in the mechanism were discovered not to conserve mass, specifically the reactions forming the C15d species, several reactions of O3RO2 with the RO2_b pool and a few HOM photolysis reactions. These reactions have been corrected in the updated mechanism and the effect of these changes were thoroughly assessed. None of these corrections caused significant change to the results or altered our

conclusions. Full detail is given in the response to comment pertaining to line 321 from the second reviewer.

**Review 1**

> *The article demonstrates the implementation of a reduced HOM mechanism into a re- duced mechanism, CRI, which was derived from MCM3.3.1. HOM were discovered a few years ago. They are supposedly formed by a fast process called autoxidation. Since from their structure HOM and accretion products have low to extreme low vapor pressure, they are important candidates for SOA formation and persistence. This links HOM to relevant issues related to aerosol effects, e.g. for climate warming. The pur- pose of the presented work was to provide a simple enough version of HOM formation mechanism within a CRI for implementation in larger models.*

> *The authors describe quite clearly the steps of the implementation with respect to the changes in the CRI. Their attention turned to the implementation of autoxidation in competition to bimolecular termination reactions. In addition they put efforts also in the implementation of HOM accretion products, as those may play a role in atmospheric nu- cleation. HOM accretion products are supposedly formed by recombination reactions of (HOM) peroxy radicals. By splitting the sum of peroxy radicals in three classes and considering class interactions they enabled a treatment of the recently described inter- action of HOM peroxy radicals with low molecular weight peroxy radicals, specifically the interaction of a-pinene and isoprene. They further implemented the temperature dependence of HOM formation as recently observed. Aspects of their improvement of the CRI – HOM mechanism were tested against flow tube experiments by Berndt et al., with satisfying results. These parts of the work are very interesting and important.*

We are glad that the reviewer found the section describing the implementation of the mechanism into the existing CRI framework clear and appreciate the reviewer's comments that the work to incorporate accretion product formation and temperature dependence as well as the tests against flow tube experiments are interesting and important.

> *The new CRI-HOM was then used to calculate vertical profiles over two stations in Hyytiälä and Manaus. Here the descriptions of the results are not fully congruent with the graphical representations. Overall, the vertical profiles of HOM and HOM accretion products are however in a reasonable range. That means within general experience it could be possible. Any verification of the height profiles by observation is missing. Things become even more speculative when the authors try to predict PI and PD nucleation and the role of biogenics and HOM. I am not sure if that part is really helpful. They authors admit that any validation is currently impossible (in the first sentence of the according paragraph). Despite of the latter, overall the article is timely, well written and can be published in ACP after addressing the comments below.*

We appreciate the reviewer's comments on the timeliness and quality of the writing. We respectfully disagree with the reviewer's suggestion that the description in the text is not in agreement with the vertical profile and believe that the text presents a fair representation of the figures. With regards to the

reviewer's comments on height profiles, the input data for several key chemical species in both Hyytiala and the Amazon were scaled by observational data to account for model biases and produce a more realistic picture for the present day profiles. In response to the reviewer's comments we also compared the modelled profiles of isoprene and α-pinene to observations from the ATTO tower and saw reasonable agreement (see comment pertaining to line 467 and response). Regarding nucleation rates, which were calculated using unaltered model data, we believe such work is useful as it provides an approximation of the nucleation rates that would be expected should CRI-HOM be incorporated in its current format into the UKCA global chemistry-aerosol-climate model. The large uncertainty surrounding the nucleation rates is acknowledged, but this reflects current scientific understanding of nucleation processes and the complexity in their simulation. However, despite these uncertainties nucleation is an essential process for composition-climate modelling and a major contributor to overall uncertainty in climate projections. Work is therefore urgently needed to improve it's representation in models, and we believe our work is an important contribution to this topic. While Kirkby et al (2016), whose work provides the basis for the nucleation rate equations used here, observed new particle formation from 10-carbon HOMs, the recent study by Heinritzi et al (2020) suggests that 20-carbon accretion product may be the key species in nucleating new particles resulting in a possible reduction to total nucleation rate. This is already acknowledged in the supplement (line 154-156) and to this end, we have added the following to line 620:

"All HOMs were treated as being equally efficient at nucleating new particles, in agreement with approach and nucleation rates used by Kirkby et al (2016) and Gordon et al (2017). Recent work by Heinritzi et al (2020) suggests that 20-carbon accretion products may be better at nucleating new particles and therefore the results presented are likely to be an upper bound although nevertheless informative. Representing the different nucleation efficiencies of different HOM species will be investigated in future work ."

**Specific Comments from Reviewer 1:**

*line 53: Bianchi et al., is a review; here you reference to the original papers in order to give the authors the credits*

Additional references have been added to include relevant work cited in Bianchi's review on line 53:

"...formation of "highly oxygenated organic molecules" (HOMs) (Mentel et al., 2014, Ehn et al., 2014, Kurtén et al., 2016, Bianchi et al., 2019)"

*line 199f: I think it must be RTN24O2, instead of RTN24BO2. Or it must be "RTN26BO2 - RTN23BO2". Or something else in the nomenclature is not consistent in this paragraph.*

This was a typographical error. RTN24O2 is correct and this change has been made to line 199.

*line 204 and line 236: If you want to make the mechanism efficient, why do you start lumping O3-HOM from the 5th generation, while you starting for OH HOM with the 4th generation?*

The lumping of the 5th gen and higher O3RO2 was done as there existed experimental data for 1st-4th gen O3RO2 while only existed experimental data for the 1st-3rd gen OHRO2. However, it is acknowledged that further mechanism reduction is possible and this will be an aim for updated versions of CRI-HOM.

> line 219: *"Alkoxy radicals are not represented explicitly due to their rapid reactions which, typically for larger peroxy radicals, are decomposition or isomerisation. " The statement doesn't make sense to me; alkoxy instead of peroxy?*

The intended message of this sentence was that for the alkoxy radicals formed from large peroxy radicals isomerisation and decomposition would be more important than reaction with $O_2$ and carbonyl formation. To clarify this, the sentence in question has been amended to the following:

"Alkoxy radicals are not represented explicitly due to their rapid reactions which, typically for alkoxy radicals formed from larger peroxy radicals, are decomposition or isomerisation."

> line 339: *How much effort would it be to implement and to test a range lifetimes typical for sticky molecules applied to your HOM species. Isn't deposition always a weak point in atmospheric models? If deposition is faster than upward transport, your vertical profiles would be obsolete.*

The HOM species themselves are not advected vertically as stated on line 611 and advection and deposition of the sources gases (α-pinene, isoprene etc) is already factored in by the parent model. We are implicitly assuming that the HOMs are short lived enough that advection is a minor contribution. Thus any change to HOM deposition or loss to the CS at one altitude will not affect HOM at other altitudes. Rather the profiles show the concentration of [HOM] predicted by the box model when supplied with input data (temperature, pressure, concentration of various parent species) from the UKCA climate model data with loss to the CS implemented by scaling the measured surface CS by relative aerosol surface area density at the level of interest.

The sensitivity for loss to the CS, predicted to be the major loss process for HOM (Bianchi et al (2017), Dal Maso et al (2002), Petäjä et al (2009), Tan et al (2018), Wu et al (2018)), is explored in the paper by scaling the CS by factors of 10 and 0.1. The results are shown to be significant (Fig S.24) and identified as an area for future work (line 676 - 677).

> line 403-409 and Table 4: *If I understand the autoxidation process correctly, the autoxidation rate must slow down at the end as suited H atoms are already consumed in previous autoxidation steps. This is the case for OH, but not for ozone? The overprediction of the highest generations compared to Berndt et al. is not necessarily only due to missing loss processes, it can be also due to overestimated source strength, by your last step autoxidation rate coefficients.*

The availability of H atoms will play a part in the autoxidation rate although, as there are more than 5 hydrogens which could be removed, exhausting the supply of hydrogens is unlikely to be a dominant issue given the number of autoxidation steps this mechanism considers. In addition, the extent of functionalisation of peroxy radical is also believed to be important; with enhanced functionalisation increasing autoxidation rate (e.g. Schervish et al (2019), Jenkin et al (2019a), Bianchi et al (2018), Otkjær et al (2018)). The shape of the molecule also plays a part, with restrictions from ring structures (of different sizes in the $O_3$ and OH pathways) also affecting the rate of H shift reactions (Rissanen et al.,

2015). All these factors result in an overall rate coefficient which is very challenging to predict and, as some of the factors enhance the rate coefficient and others reduce it, the fact that autoxidation coefficients for the $O_3$ and OH pathways do not follow quite the same pattern is not surprising. The complexity of this issue means fitting to experimental data such as that from Berndt et al (2018b) is the best method.

Regarding the issue of source strength, it is appreciated that the concentration of any $RO_2$ species is determined by its source and sink. The approach taken here involved constraining the sink (dominated by the autoxidation coefficient in the case of Berndt's experiment) for the 1st generation $RO_2$ and then, using this as source of the 2nd generation $RO_2$, constraining the sink of the 2nd generation $RO_2$ and so on. We believe this is the best approach as it allows for the maximum possible level of constraint but we do acknowledge that if there were additional loss processes for any $RO_2$, their inclusion would necessitate a reduction in the autoxidation coefficient of the corresponding $RO_2$ and thus source strength for the next generation $RO_2$. This remains a key challenge for future work and has been clarified for the reader with the following amendment to line 406 :

 Additional loss process would likely reduce the fitted autoxidation coefficients because they would provide an additional sink for the $RO_2$ species which does not lead to the production of the next generation $RO_2$. Therefore, the autoxidation coefficients determined in this work are likely to be upper limits but further insight into this is not possible with the data currently available. This is a key area for further study.

> *line 426-429: Overpredicting the first generation of OHRO2 has nothing to do with the HOM-mechanism, correct? From this point of view it may be a severe principal failure of your CRI scheme. What could be reasons for that? This should be discussed a little more extensively.*

JMW thoughts: In terms of the remaining discrepancy, Finally, it is worth conceding that there may be an error in the CRI mechanism (although the overall rate of APINENE + OH is consistent between CRI and the MCM has been used by others, e.g. Pye et al (2018) but argue that it is not as large (i.e. not a "severe principal failure") as reviewer suggests the CRI scheme.

The modelled 1st generation OHRO2 was dominated by RTN28AO2 which had a concentration around 10 times greater than RTN28BO2. RTN28AO2 does not autoxidise to form later generation $RO_2$ and therefore cannot contribute to HOM. Thus, the overprediction of the 1st generation OHRO2 is unlikely to have a significant impact on HOM concentrations. The concentration of the 1st generation OHRO2 was believed to be underestimated in the experimental work (Berndt et al., 2018b) by a factor of 5. The most likely reason for this is the lower binding energy of the 1st generation OHRO2 with the reagent ions used in the mass spectrometry which arises from the lower number of heteroatoms in the peroxy radical. Since the CRI-HOM model simulations are 10 times greater than the experimental data for 1st gen OHRO2, the model is likely to be overpredicting the 1st generation OHRO2 by only a factor of 2. The rate coefficient for the production of the 1st generation OHRO2 has undergone extensive evaluation and the same coefficient is used in the CRI v2.2 parent mechanism which has been optimised against the Master Chemical Jenkin (Jenkin et al., 1997, Saunders et al., 2003, Jenkin et al., 2015, Jenkin et al., 2019).

Therefore we respectfully disagree that this is a "severe principal failure" of the CRI. A more likely explanation is the presence of additional, as yet unknown loss processes not currently included in the model, but in the absence of additional data, no further insights can be made at this time. To clarify this, the following adjustment has been made to line 428:

"The experimental measurements of 1st generation OHRO2 concentration from Berndt et al (2018b) were believed to be underestimated by about a factor of 5  (Fig. S3). This suggests the model overprediction of the concentration of OHRO2 may be about a factor of 2. The cause of the discrepancy between modelled and measured 1st generation OHRO2 remains unclear. The rate coefficient for the production of the 1st generation OHRO2 has undergone extensive evaluation and the same coefficient is used in the CRI v2.2 parent mechanism which has been optimised against the Master Chemical Jenkin (Jenkin et al., 1997, Sanders et al., 2003, Jenkin et al., 2015, Jenkin et al., 2019a). Sensitivity tests perturbing the branching ratio between RTN28AO2 and RTN28BO2 revealed that even doubling the fraction of RTN28BO2, a significant deviation from literature (Berndt et al., 2016, Pye et al., 2018), had negligible effect as did changing initial [OH] by +100 % / -90%. Another explanation is the presence of additional, as yet unknown loss processes not currently included in the model, but in the absence of additional data, no further insights can be made at this time. More importantly, the 1st generation OHRO2 does not form HOM itself and so it is unlikely to have a significant impact on HOM concentration. Furthermore, the modelled 1st generation OHRO2 was dominated by RTN28AO2, the species which does not autoxidise to form later generation $RO_2$. Nevertheless, this remains an important area for future work but one where more data is needed for additional constraints to be put in place." [**EDIT**]

> *line 467: In section 4, I do not understand the selections of sites for comparison. Why compare Alabama with Southern Finland at the ground, but calculating vertical profiles over Southern Finland and Amazonia. You should compare with the Manus ground data, too. Moreover, there were big campaigns over Amazonia and Finland, also with airplanes. Can't you use data to validate at least parts of your vertical profiles, e.g. OH, O3, NOx, a-pinene, isoprene, selected OVOC?*

The sites in Alabama and Finland were chosen because there exists data of HOM (or related species) concentrations at these sites along with the concentrations of several other important species (isoprene, $O_3$, OH etc) which were used as model inputs to facilitate more faithful modelling of the surface conditions. Surface conditions for the Amazon were not considered in detail as no studies exist of surface HOM measurements in the Amazon (Bianchi et al., 2019) and Zhu et al (2019) notes that nucleation at ground level is almost never observed in the Amazon, a finding in agreement with the results from simulations in this work.

The Amazon and Finland sites were chosen to explore the importance of the Isoprene/MT ratio since these locations are markedly different in this context. In the Amazon, the vertical profiles of isoprene and α-pinene concentration, which were used as input data for each box model run, were adjusted based on multiple vertical measurements (Kuhn et al., 2007) to account for biases in UKCA model data (l. 583-

584) and improve the validity of the simulated profiles. The same approach was taken for the runs over Hyytiala with concentrations for α-pinene scaled to agree with data from Roldin et al (2019) and there was good agreement between the modelled OH and $O_3$ used by Roldin et al (2019) and the OH and $O_3$ UKCA data used as CRI-HOM model input. In addition, we have now also compared the lower parts of the isoprene and α-pinene profiles to those measured at the ATTO tower (Yáñez-Serrano et al., 2015) measured over the lowest 80 m. We find that the modelled isoprene showed good agreement with the observed isoprene column, falling within the observed data's standard deviation, while the α-pinene fell just outside the upper limit of the observed data's standard deviation. To highlight this, a new figure (Fig S22) has been added to the SI and the following text has been added in line 584:

"The scaled values of isoprene and α-pinene showed reasonable agreement with observations taken up to 80 m in altitude at the ATTO tower (Yáñez-Serrano et al., 2015). Modelled isoprene fell within 0.5 ppb of observation taken at 2 pm in June while modelled monoterpene were within 0.1 ppb of observation, well within the observational standard deviation in both cases (Fig. S22)."

> line 589: Do really mean "semi-qualitative". That would implicate not even qualitatitve. . .? The profiles look quite reasonable, overall. And you highlight features of the profiles. . .

The confusion surrounding "semi-qualitative" is acknowledged and it has been replaced with "illustrative" on line 588 which better conveys the intended meaning that the shape of the profiles are useful but there is uncertainty in the absolute values:

"Therefore, we can suggest that our simulated vertical profiles be regarded as illustrative as more work is required to identify if the condensation sink should be species dependent."

> line 592-594: What is the sense of the comparison then (see comment above)? Can't you split off from the observations the compounds which are in your model?

The comparison to the observations from Alabama was performed to assess whether the CRI-HOM mechanism was producing reasonable values in a second location rather than serve as an opportunity for deep scrutiny. Furthermore, separating out the C10 and C9 compounds is unlikely to provide much greater clarity as some of the observed C10 compounds will probably come from other species such as $\beta$-pinene, whose concentration is not stated, and therefore direct comparison is not possible. We believe that the conclusion we have drawn, that the CRI-HOM model produces a "reasonable value", is fair and does not exceed the level of confidence we can have in the model given the observed data..

> line 596-602 and Figure 7: In contrast to your statement, the OH data decrease with height and O3-HOM and OH-HOM do not have the same share there anymore. Why do the OH-HOM decrease in upper troposphere? It would be helpful to show vertical profiles for OH, O3 and possibly NO, too. Moreover, in legend of Figure 7 you describe features (arrows) which I cannot see. And the color code for O3-HOM is brown, while the line is orange.

Figure 7 has been adjusted to include plots of $O_3$, OH and NO as requested. The decline at high altitude in HOM from OH relative to HOM from $O_3$ arises from the greater sensitivity to temperature of HOM formed from OH-oxidation than ozonolysis. This comes from the fact that 2nd generation and higher O3RO2 can form HOMs via reaction with $HO_2$, meaning only 1 autoxidation step is required. However,

only 3rd generation and higher OHRO2 can form HOMs via the same mechanism (due to the lower number of oxygen atoms in the initial α-pinene oxidation product and discussed in Section 2 of the manuscript), necessitating two autoxidation steps. As the autoxidation coefficients are highly temperature dependent, the need for two steps confers a greater temperature sensitivity to HOM from OH. This is less noticeable at high temperatures where autoxidation can compete effectively with bimolecular reactions but this ceases to be the case at ~250 K (based on typical NO and $HO_2$ concentrations) and it is around this temperature that significant divergence starts to occur between HOM from OH and HOM from $O_3$. This occurs at ~5-6 km in Hyytiala but at 8-9 km in the Amazon due to its elevated temperature profile. This is also illustrated by the the significant difference in HOM from OH profiles resulting from autoxidation activation energies of 12077 K and 6000 K. The divergence occurs at a much higher temperature (lower altitude) in the 12077 K case because autoxidation is outcompeted by bimolecular reactions more easily. To clarify this point in the manuscript, the following addition has been made to line 599:

"HOM from OH showed a significantly greater sensitivity to temperature, diverging from the HOM from $O_3$ at around 5 km in Hyytiala and 8 km in the Amazon due to the elevated temperature profile. This was attributed to the requirement for 1$^{st}$ generation OHRO2 to undergo two autoxidation steps before HOMs can be formed (Section 2.2) while 1$^{st}$ generation O3RO2 only need to undergo one autoxidation step and thus have a weaker temperature dependence. This effect only becomes noticeable at temperatures below ~250 K when autoxidation ceases to compete effectively with bimolecular reactions."

The arrows in Figure 7 were previously removed and analysis transferred to Table 5. The caption has been corrected by the following removal.

**[DONE]**

The colours used for O3-HOM have been checked and the author confirms that the same colour, red, is used in the legend and for lines.

*line 672: There were no vertical profiles over Alabama, right?*

Yes, there were no vertical profiles over Alabama.

*Captions, Figure 1 and 2: I suggest to introduce the meaning of $RO_2$s, m, b also in the captions.*

The following text has been added to the captions of Figures 1 and 2.

The $RO_2$ pool is split into subsections covering big ($RO_2$b), medium ($RO_2$m) and small ($RO_2$s) peroxy radicals to facilitate addition of accretion product formation.

*Typos, errors:*

*line 70: reference Sindelarova et al., 2014, is missing in the reference list*

Sindelrova et al (2014) has been added to the reference list.**[DONE]**

*line 141: reference Kiendler-Scharr, instead of Kiendler-Scherr*

This has been corrected. **[DONE]**

*line 150-160: in the reaction equation 3 and 4: it should be C5RO2 instead of C10RO2*

This has been corrected. Equations (3) and (4) are now:

$$C10RO_2 \; + \; C5RO_2 \; \rightarrow C10 \; + \; C5 \; + \; O_2 \, (3)$$

$$C10RO_2 \; + \; C5RO_2 \; \rightarrow C10RO \; + \; C5RO \; + \; O_2 \, (4)$$

*line 245: in equ. (9) 10x instead of 10z*

This has been corrected, equation (9) now reads:

$RTNaBOyO2 \; + \; HO_2 \; \rightarrow RTN28OOH \; or \; C10x \; (9)$**[DONE]**

*line 274: Jenkin et al. 2019, a or b missing*

This has been corrected.**[DONE]**

*line 287: m instead of me*

This has been corrected.**[DONE]**

*line 306: than instead of that*

This has been corrected.**[DONE]**

*line 625: you call the underestimation of H2SO4 slightly, I see 1-2 orders of magnitude*

This point is acknowledged and both uses of the word "slightly" have been removed from this paragraph to yield the following from line 623:

"Modelled concentrations in the Amazon ($3x10^4$ $cm^{-3}$) were  lower than observation ($10^5$-$10^6$ $cm^{-3}$ (Wimmer et al., 2018)) although the observations were taken in a pasture site downwind of Manaus surrounded by the rainforest not in the rainforest itself and are therefore likely to be higher than in-situ rainforest values. Thus, the nucleation rates we have calculated for $SA_{act}$ are likely to be a reasonable estimate in Hyytiala and  low biased in the Amazon."

Given that the Amazon measurements are from a region which is likely to have higher $SO_2$ than the more remote jungle (as it is influenced by the Manaus plume), we believe that the nucleation profiles are still informative.

*There are frequently passages in the text using a different font size. e.g. lines 57/58, line 378/379, line 421, line 422, line 458/459, line 464/465, line 561*

These amendments have been made.

*Table 1: "gen."*

This has been corrected.

*Table 3: needs reformatting of the text fields. . .*

This has been corrected.

*In general, some page formatting issues with Figures and Tables. Figures S17, S18: The subscripts are too small.*

Figures S17 and S18 (now Figures S19 and S20) have been enlarged to remedy the issue relating to the subscripts.

**Review 2**

*General comments:*

*The manuscript represents a valuable addition to the recent surge in literature reports regarding HOM formation from monoterpenes, as it synthesizes inputs from various sources into a modular chemical mechanism that can be applied to further research into the role of HOMs in the atmosphere. However, with an eye toward those future projects to which this mechanism will surely make an important contribution, it **is imperative that the authors make clear the remaining uncertainties, sensitivities, and assumptions inherent to the key parameters and output of this CRI-HOM mechanism**. The mechanism is an important first step, but in order for future fine-tuning to be conducted, it will be necessary to not just acknowledge but to actively advertise the aspects of the mechanism that remain most uncertain. This will give the critical dialog between models, observations, and laboratory studies room to improve this mechanism as new constraints become available. To that end, I think a number of efforts could be made in this manuscript to clarify the sources and magnitudes of uncertainty, the origins of certain specific assumptions, the sensitivity of mechanism parameters (e.g. branching ratios, rate coefficients) to assumptions made, and the ranges of parameter values that would be consistent with the limited and/or highly uncertain HOM and RO2 observations.*

*As has already been alluded to, the mechanism relies on a number of assumptions and extrapolations between species for many branching ratios and reactions rates. That in itself isn't bad and doesn't invalidate any of this, but requires careful attention to the sensitivity to those assumptions and the resulting uncertainty in the mechanism's parameterizations and output. I would hate to suggest running more models; instead I think*

*(a) **some of this sensitivity analysis already exists in the SI (e.g. L 364-366) and should be given a more prominent billing;***

*(b) **some parameters and model output could benefit from bootstrap back-of-envelope calculations (or comparisons to previous literature, as I suspect exists for e.g. the first-***

*generation pinene ozonolysis yields) as to their sensitivities to certain inputs or ranges that would be consistent with observations, and*

*(c) descriptions of the mechanism would benefit from more careful attention to what can be stated with certainty and what results of the simulations are so sensitive to highly uncertain numbers that they can't be considered conclusive.*

*Along these lines, see especially the comments to L222-224, 304-314, 341-347, 507-508 below.*

We thank the reviewer for their comments and are pleased that they assess the work as a valuable contribution to the field. We acknowledge the suggestions made by the reviewer and briefly summarise our response to points (a) - (c) below with more detailed responses provided to the reviewer's specific comments.

In response to comment (a) more detail of the sensitivity analysis has been transferred from the SI to the main text and this is detailed in response to the reviewer's comment on lines 416-418.

In response to (b), more detailed comparison to literature has been added to the manuscript regarding multiple mechanistic parameters and the associated uncertainty (see response to comment beginning "*Finally, it's somewhat unclear without reading...*"). We have also performed multiple sensitivity studies to assess the impact of the uncertainty in the branching ratio of alkoxy radicals (see response to the comment pertaining to line 222), magnitude of rate coefficients k14 and k15 (see response to the comment pertaining to line 304) and their branching ratio, accretion product formation rate coefficient (see response to the comment pertaining to Figure 5) and HOM loss to OH (see response to the comment pertaining to lines 341-347.)

In response to (c), in addition to details of the sensitivity studies added in the main text, additional information has been added to the manuscript regarding multiple mechanistic parameters and our confidence in them. Furthermore, Table S6 has been repurposed to summarise the impact of uncertainty in different mechanistic parameters discussed in the main text.

*The large uncertainty bounds on measurements to which the mechanism was compared (as noted in L 428-429) suggests a need for reporting a range of mechanistic parameterizations consistent with the measurements, rather than single values. The kind of analysis you do on L 364-366 of the SI is hugely useful for these purposes, and should be incorporated into the main text (and, as mentioned below, the ranges and sources reported in Table S6 would also be useful in the main text).*

The sensitivity studies have been added into the main text and more detail is provided in the response to the comment pertaining to lines 416-418. The contents on original Table S6 has been added to Table 4 in the revised version.

*However, these sensitivity studies could benefit from more detailed descriptions. **It sounds as though the uncertainties were only estimated by changing one rate at a time and comparing the***

*resulting changes in the concentration of the peroxy radical in question with the upper and lower bounds of the experimental uncertainty. This would neglect any compounding effects from simultaneous changes in multiple autoxidation coefficients, or in both autoxidation coefficients and alkoxy decomposition: isomerization branching ratios, correct? This should be acknowledged (or, if possible without too much additional work, the cumulative uncertainties could be estimated and reported)..*

Uncertainties were estimated as the reviewer describes. The limitations of this method are acknowledged but with this addition, as recommended by the reviewer, we feel the method used is now sufficiently clear. We also note that an alternative approach would be a Monte Carlo simulation but feel this is beyond the scope of this paper. To clarify this in the manuscript the following amendments were made on line 401:

". An estimation of the uncertainty in the autoxidation is also provided in Table 4. These values were calculated by adjusting the autoxidation rate coefficients one at a time to determine the maximum and minimum values of an autoxidation rate coefficient for which the corresponding peroxy radical would fall within the experimental uncertainty region. This approach neglects any cross-sensitivities through the joint uncertainty in several rate coefficients. A full Monte Carlo uncertainty analysis addressing this issue is beyond the scope of this manuscript but would make a valuable follow up for future work in this field. Therefore, the autoxidation rate coefficient uncertainties are large as the experimental error uncertainties are large."

*Also, if similar sensitivities exist to show the range of HOM yields from RO2-RO2 chemistry that is consistent with the large uncertainty bounds shown in Figure 5, those would be useful to see as well.*

The accretion products were observed to make a negligible contribution to HOM yield given their much lower concentration compared to the HOM monomers (C10x and C10z). Accordingly, the uncertainty in total HOM yield arising from the experimental uncertainty in accretion product formation is negligible. However, we do address the issue of uncertainty in the rate coefficients for C20d and C15d formation arising from the experimental uncertainty and this is described in detail in response to the comment pertaining to Figure 5.

*Finally, it's somewhat unclear without reading the whole paper and SI very carefully what branching ratios / rates are fit, which are plugged in from measured values or extrapolated from similar species, and which are educated guesses. Someone wanting to use this mechanism or adapt it for their own uses might want to know which coefficients are flexible and which are most tightly constrained (or measured). Can a quick representation of that be provided? Either as an expanded Table 4 / Table S6 (with, say, a superscript character on each rate or branching ratio to denote which come from what sources) or additional annotations to figures 1 and 2 that make*

*it clear (e.g. colour-coded arrows corresponding to which rates and branching ratios come from which sources).*

We agree that a key aim of this work is for this mechanism to be used by others and, following the reviewer's suggestion, we have repurposed Table S6 to provide more information about the source and confidence in model parameters. In addition to Table S6, we have made further additions to the text which address this issue in response to specific comments below.

To make it easier for others to use the mechanism, we have also made available the KPP documentation files used for the mechanism and deposited them in the University of Cambridge repository. We have added the following to line 701:

*"Data Availability.* All modelled data is available upon request from James Weber and all experimental data from Torsten Berndt. The KPP files for the CRI-HOM mechanism have been deposited in the University of Cambridge data repository and can be viewed at doi.org/10.17863/CAM.54546."

**Specific Comments from Reviewer 2:**

> *L 154-155: For mass conservation and to fit with the explanation in L 145-147, I assume reactions 3-4 are supposed to have C10RO2 + C5RO2 as the reactants?*

Yes, this error has been corrected as discussed in the response to 1st reviewer's comments pertaining to lines 150-160.

> *L 194-195: Figure 1 implies that these TNCARB26 and RCOOH25 co-products, along with the major products of RN26BO2 and RTN24O2, are formed in fixed yields from pinene ozonolysis. However, this sentence (and my understanding of Criegee intermediates) would imply that the branching ratios to these products depends on the relative abundance of the Criegee intermediates' reaction partners, such as water. Could you clarify here and/or in the caption to Figure 1 whether/how this Criegee chemistry is represented, and whether it matters?*

Criegee intermediates are not considered explicitly in this mechanism. The 17.5% and 2.5% static branching ratios for TNCARB26 and RCOOH25 respectively, used to parameterize some of the effects of Criegee intermediate chemistry, remain unchanged from the Common Representatives Intermediates v2.2 mechanism and are supported by numerous studies (IUPAC Task Group on Atmospheric Chemical Kinetic Data Evaluation (http://iupac.pole-ether.fr , last accessed 17th May 2020), Atkinson and Arey (2003), Johnson and Marston (2008)). While the relative abundance of the Criegee reaction partners is likely to affect the product distribution to some extent, these branching ratios are believed to be acceptable in most ambient conditions. To clarify the matter, the following addition has been made to lines 194-195:

"In addition, TNCARB26 (closed shell carbonyl species) and RCOOH25 (pinonic acid) arise from the reaction of Criegee intermediates with water. The yields of these species, 17.5% and 2.5% respectively, remain unchanged from the CRI v2.2 mechanism and are well supported in the literature (IUPAC Task

Group on Atmospheric Chemical Kinetic Data Evaluation (http://iupac.pole-ether.fr , last accessed 17th May 2020),  Atkinson, Arey et al (2003), Johnson, Marston et al (2008))."

> *L 199: This sentence refers to "RTN24BO2", but the co-product in Figure 1 is "RTN24O2".*

This typographical error has been resolved with RTN24BO2 corrected to RTN24O2 as discussed in response to the 1st reviewer's comments. **[DONE]**

> *L 222-224: Are there any constraints or uncertainty bounds on this 50:50 ratio of de-composition and isomerization pathways used here? It seems that in environments where reaction with NO is competitive with the other reaction pathways (and even in very low-NO conditions, since the alkoxy radicals are formed in RO2-RO2 reactions as well), the resulting HOM yield could be highly sensitive to this branching ratio. In the absence of concrete evidence for these specific alkoxy intermediates' branching ratios, can you provide some estimate of the sensitivity of your mechanism's output to the chosen ratio?* `This would be useful either here or later, when you describe the important of this NO-derived HOM in the context of the model output (~L 505-511)`

To investigate the sensitivity of the mechanism to this branching ratio, we ran two sensitivity tests with decomposition : isomerisation ratios of 75:25 and 25:75 respectively. These changes did not affect the ability of the model to reproduce data from Berndt et al (2018b) (Simulations A and B) which is unsurprising given the conditions of Berntd's experiment which promoted autoxidation as the dominant loss mechanism for $RO_2$. Therefore, the uncertainty in this branching ratio does not affect the values of the rate coefficients determined for autoxidation and accretion product formation.

We also performed these sensitivity tests with Simulation C to look at the effect on HOM yield. In this case, the difference in HOM yield for low NOx (up to 200 ppt) was negligible. At 2ppb of NOx a greater discrepancy of around ±0.7 percentage points (at 290 K) (~ 20%) was observed with the test with lower isomerisation producing a lower HOM yield as expected. Above 2 ppb NOx, the difference between perturbation increased but this concincided with a significant drop in HOM yield as autoxidation was outcompeted by reaction of $RO_2$ with NO. To illustrate this further, we have included a new figure in SI (Fig. S4(b)) and refer to it in the added text. Importantly, the range of HOM yields arising from the uncertainty in autoxidation temperature dependence was larger than the range of HOM yields spanned by the isomerisation-decomposition ratio perturbation (for NOx ≤ 2 ppb), and this was even more the case at 270 K, suggesting that, while important, the isomerisation-decomposition ratio is less important than the refining our understanding of the autoxidation temperature dependence.

To clarify this in the manuscript, the following additions have been made:

**Line 224:** "Sensitivity tests perturbing the branching ratio between 75:25 and 25:75 were performed to probe the consequences of this uncertainty. These tests suggested the precise values of this branching ratio within this range did not affect the fitting of rate coefficients for autoxidation and accretion product formation (Section 3.1). These branching ratio perturbations did lead to changes in HOM yield (Fig. S4(b)) and are discussed in more detail in Section 3.2."

**Line 520:** "The HOM yield showed negligible sensitivity to the alkoxy radical decomposition-isomerisation branching ratio below 200 ppt of $NO_x$ and around ±0.7 percentage points (~20%) at 2 ppb $NO_x$. However, this range was easily encompassed by the range arising from autoxidation temperature dependence uncertainty. Above 2 ppb $NO_x$, this ratio had greater influence as NO reactions with $RO_2$ started to compete more efficiently with autoxidation but this coincided with the sharp drop in HOM yield (Fig. S4(b)). Therefore, while further work is needed to develop the isomerisation-decomposition branching ratio description, it is unlikely to have a significant influence in the low-$NO_x$ conditions where HOM are predicted to be most prevalent and in these conditions the uncertainty in temperature dependence of autoxidation is predicted to have a larger effect."

Table S6 has also been adjusted to include an entry summarising these conclusions.

> *244-246: Should equation 9 (and the line preceding it) read C10x or C10z? I realize the difference is described in Table 1, but it would be helpful to spell it out in the text as well so the reader doesn't get confused on this point.*

The correct HOM species is C10x. The equation is has been corrected to read:

$$RTNaBObO2\ +\ HO_2\ \rightarrow RTN28OOH\ or\ C10x\ (9)$$

The complexity is acknowledged and to this end the sentence beginning on line 243 has been amended to as follows:

 The HOM produced by all later generation OHRO2 is termed C10x (Eq. 9).

> *L 283-294: Either this semicolon is meant to be a colon, or something's missing in the description of the UCARB10/UCARB12 products that would turn it into a standalone clause.*

This should be a colon and the sentence has been corrected.

> *L 287: Does this "me" mean medium? If so, doesn't this contradict the statement (pre- vious page, L272-3) that medium peroxy radicals react individually with each peroxy radical pool too?*

"me" should read medium - this error has been corrected. Yes there is a contradiction. When considering the reaction of a particular medium peroxy radical, we model it to react only with the entire pool. However, when considering a large peroxy radical, this reacts not with the whole pool but with the big, medium and small pools individually. The phrase "*Each big and medium peroxy radical reacts separately with each peroxy radical pool while, to minimise the total number of reactions, all small peroxy radicals react with the total pool as accretion product formation is much less favourable (Jenkin et al., 2019)*" was not updated when the decision to make medium peroxy radicals react with whole peroxy radical pool (lines 288-289) was taken. The above phrase (lines 272-3) has been amended to:

Each big  radical reacts separately with each peroxy radical pool while, to minimise the total number of reactions, all small peroxy radicals react with the total pool as accretion product formation is much less favourable (Jenkin et al., 2019). Medium peroxy radicals are discussed below.

*L 304-314: The citation of MCM implies that k14 and k15 were derived from Jenkin et al. 2019a, but the citation two lines later implies they are from Molteni et al. 2018. Which were they?*

k14 and k15 were based on Molteni et al (2019) for O3RO2 and Roldin et al (2019) for OHRO2 and the text has been amended to reflect this as shown below.

"...resulted in R13 being more important that R14 and R15 which had rate coefficients based on literature (MCM, Jenkin et al., 2019a) (Molteni et al., 2019, Roldin et al., 2019) up to an order of magnitude lower."

*Also, this presents another fixed input in the mechanism to which the model output and parameter fitting might be highly sensitive. Are there uncertainty bounds on the rate coefficients from Molteni et al. or Roldin et al. 2019 that can be used to estimate this sensitivity?*

Roldin et al (2019) does not provide an estimate of uncertainty in the rate coefficient. Molteni et al (2019) provides a range of experimentally-determined rate coefficients from different combinations of reacting O3RO2 species. The individual rate coefficients disclosed by Molteni spanned 2 orders of magnitude, illustrating the influence that the reacting $RO_2$ has on the rate coefficient. The mean of all available rate coefficients was taken. The uncertainty in rate coefficients k14 and k15 is acknowledged and the challenge of deriving a lumped value for a parameter which varies between $RO_2$ is evident. To this end, we first ran several sensitivity tests scaling all $RO_2$-$RO_2$ rate coefficients of the large peroxy radicals (except for the accretion product formation rate coefficients) were scaled values by factors of 10 and 0.1 to explore the likely outer bounds of parameter space. Applying these scalings did not affect the model output when simulating runs from Berndt et al (2018b) (Simulations A and B). Therefore we can conclude that the uncertainty in k14 and k15 did not affect the fitting of the rate coefficients for autoxidation or accretion product formation which is unsurprising as Berndt's experiments were designed so that losses via $RO_2$-$RO_2$ reactions were small.

To explore the impact of the uncertainty further, the simulation run to examine HOM yield (Simulation C) was rerun with the aforementioned scalings. In this case, HOM yield proved more sensitive to this perturbation. More modest scalings were also considered with scalings of +100% and -50% produced changes in HOM yield of +65% and -40% respectively, suggesting significant sensitivity to these rate coefficients. To illustrate this further we have included a new figure in the SI (Fig. S4(a)) showing this dependence and refer to it in the text. We acknowledge that this approach will not capture the fact that HOM yield may be more sensitive to changes in the values for k14 and k15 for some $RO_2$ than others. To fully investigate this would require a Monte Carlo simulation which is beyond the scope of this work. To clarify the matter and emphasise the sensitivity of HOM yield to this parameter, the following additions have been made:

**Line 311** "Sensitivity tests where all values of k14 and k15 were scaled by the same factor revealed that the uncertainty in the value of these rate coefficients did not affect the fitting of rate coefficients for autoxidation and accretion product formation (Section 3.1). These branching ratio perturbations did lead to changes in HOM yield (Fig. S4(a)) and are discussed in more detail in Section 3.2."**[DONE]**

**Line 520 (already introduced a very start of response)** "The slight low bias may be in part due the values of k14 and k15 which were shown to influence the HOM yield relatively strongly. Sensitivity tests involving universal doubling and halving of the rate coefficients produced HOM yield changes of around +65% and -40% (Fig. S4(a)) respectively while preserving the general dependencies on $NO_x$ and temperature (Table S6). This area of uncertainty will be the focus of future work.".

> *Are there constraints on the 50:50 branching between R14 and R15, or any particular reason to have chosen that branching? Can any estimates be made of the mechanism's sensitivity to this branching?*

Jenkin et al (2019a) suggests a branching ratio of 60% alkoxy radical (k14) to 40% closed shell species (k15) for primary and secondary $RO_2$ species, extending this to 80:20 for tertiary $RO_2$ species. Thus the 50:50 value used by CRI-HOM encompases the range of similar literature values. To investigate this further, we ran two sensitivity tests with alkoxy : closed shell branching ratios of 60:40 and 80:20 respectively. These changes did not affect the ability of the model to reproduce data from Berndt et al (2018b) and thus uncertainty in this branching does not affect the values of the rate coefficients determined for autoxidation and accretion product formation. The effect on HOM yield (Simulation C) was also negligible and much smaller than the range arising from the uncertainty in the temperature dependence of the autoxidation rate coefficient.

To clarify this in the manuscript, the following sentence has been added to line 311:

"This value is close to the value of closed shell : alkoxy radical of 40:60 ratio suggested for primary and secondary peroxy radicals by Jenkin et al (2019a) but further from the 20:80 suggested for tertiary peroxy radicals. However, sensitivity tests where the mechanism was run with branching ratios of 40:60 and 20:80 revealed that the precise values of this branching ratio within this range did not affect the fitting of rate coefficients for autoxidation and accretion product formation (Section 3.1). These branching ratio perturbations led to changes in negligible HOM yield (Section 3.2, Simulation C) which were much smaller than the range in the HOM yield simulated to arise from the uncertainty in the autoxidation temperature dependence and are therefore considered to be of minor importance."

An entry has also been made to Table S6 summarising these points.

> *L 325 & L309: You report a range of chosen fitted values for k13 and k16. Is this range of different values for the different peroxy radicals within each group, or does it repre- sent some sort of uncertainty?*

For both the 20 carbon and 15 carbon accretion products, different rate coefficients were used for different reacting $RO_2$ species to reflect the enhanced propensity to form accretion products when more functionalised. Full detail of this is provided in the SI, specifically reactions 21, 23, 25, 27, 29, 46, 63, 64, 65 and 66 for the 20 carbon accretion product and reactions 36, 37, 38, 39, 40, 47, 71, 72, 73 and 74. To provide additional clarity for the reader, the relevant reactions in the SI list have been provided in full (see additions made below in response to the next comment). [Add ref to KPP files]

> *Referring forward to section 3.1.3 as suggested here does not clarify the fitting procedure. The repeated use of "chosen" and "assigned" sounds more like the values were user-selected out of a*

*pre-defined range rather than fit to data, and the model output then compared favourably to the highly uncertain aggregate HOM measurements (which, as far as I can tell, don't distinguish between the generation in which the HOM was formed). How was the initial range over which to "fit" chosen?*

It is agreed that "assigned" and "chosen" could be misinterpreted and the rate coefficients were indeed derived from fitting against data via numerous rounds of optimisation. The initial range of values were chosen from literature (Berndt et al., 2018b) but this data only covered a small subset of the possible reactions, hence the need for optimisation. To clarify the issue the following additions have been made.

[revised manuscript text omitted]

*L 321: Why doesn't R16 conserve mass the same way R13 does?*

The reviewer is correct that there is an error in the original reaction R16 which caused a loss of mass, artificially, for these peroxy radicals. This has been amended to the following reaction which does conserve mass:

C10RO2 + RO2$_m$ → 0.667C15d + O2

To investigate the consequence of this correction, the updated mechanism was run in Simulation B and the rate coefficients for R16 were adjusted so that the model could reproduce observed C15d concentrations. Scaling the rate coefficient in nearly all cases by a factor of 1.5 allowed the model to reproduce the experimental data. This simple adjustment was suitable because the formation of C15d was a minor sink for O3RO2 and OHRO2 and so increasing the rate coefficient had a negligible effect on O3RO2 and OHRO2 while maintaining the original production flux of C15d.

RN26BO2 = C15d : 3E-12 RO2$_m$ ;
RN26BO2 = 0.667C15d : 3.9E-12 RO2$_m$ ;

RN25BO2O2 = C15d : 4E-12 RO2$_m$ ;
RN25BO2O2 = 0.667C15d : 5.2E-12 RO2$_m$ ;

RN24BO4O2 = 0.667C15d : 5E-12 RO2$_m$  ;
RN24BO4O2 = 0.667C15d : 6.5E-12 RO2$_m$  ;

RN23BO6O2 = C15d : 5E-12 RO2$_m$  ;

RN23BO6O2 = 0.667C15d : 6.5E-12 RO2$_m$  ;

RNxBOyO2 = C15d : 5E-12 RO2$_m$   ;
RNxBOyO2 = 0.667C15d : 7.5E-12 RO2$_m$   ;

RTN28AO2 = 0.667C15d : 1.2E-12*RO2_m ;
RTN28AO2 = 0.667C15d : 1.8E-12*RO2_m ;

RTN28BO2 = 0.667C15d : 1.2E-12 RO2$_m$ ;
RTN28BO2 = 0.667C15d : 1.8E-12 RO2$_m$ ;

RTN27BO2O2 = 0.667C15d: 2.5E-12 RO2$_m$ ;
RTN27BO2O2 = 0.667C15d: 3.75E-12 RO2$_m$ ;

RTN26BO4O2 = 0.667C15d: 2.5E-12 RO2$_m$ ;
RTN26BO4O2 = 0.667C15d: 3.75E-12 RO2$_m$ ;

RTNxBOyO2 = 0.667C15d : 2.5E-12 RO2$_m$ ;
RTNxBOyO2 = 0.667C15d : 3.75E-12 RO2$_m$ ;

In addition, after further inspection it was realised that several other reactions in the mechanism were not conserving mass and these were also corrected in the updated mechanism. The updated reactions involved the photolysis of HOM, a minor sink for HOM as discussed in the main text, and some of the reactions of O3RO2 with the RO2$_b$ pool. The original (blue) and updated (red) reactions are shown below and have also been amended in the reaction list in the SI. Note that separately some photolysis reactions had been labelled erroneously as having frequency J15 instead of J22, the frequency used. This correction has also been made.

C10z = RN25BO2O2 : J15;
C10z = RN25BO2O2 : J22  ;

C10x = RTN27BO2O2: J15;
C10x = RTN27BO2O2:  J22;

C20d = 0.5RN25BO2O2 + 0.5RTN27BO2O2 : J15;
C20d = RN25BO2O2 + RTN27BO2O2 : J22;

C15d = 0.5RN25BO2O2 + 0.5RTN27BO2O2: J15;
C15d = UCARB12 + 0.5RN25BO2O2 + 0.5RTN27BO2O2 : J22;

For the following reactions of O3RO2 with the RO2$_b$ pool, it was realised that two reactions had been lumped into one but the rate coefficient had not been adjusted properly (by a doubling of the rate coefficient and halving of each product's fractional coefficient). This has been corrected in the updated version:

RN26BO2 =  TNCARB26 + 0.5RN25BO2O2 + 0.5CARB16 + 0.5RN8O2  : 1.68E-12*RO2_b ;
RN26BO2 = 0.5TNCARB26 + 0.25RN25BO2O2 + 0.25CARB16 + 0.25RN8O2  : 3.36E-12*RO2_b ;

RN25BO2O2 = C10z + 0.5RN24BO4O2 + 0.5CARB16 + 0.5RN8O2 :  1.68E-12*RO2_b;
RN25BO2O2 = 0.5C10z + 0.25RN24BO4O2 + 0.25CARB16 + 0.25RN8O2 :  3.36E-12*RO2_b;

RN24BO4O2 = C10z + 0.5RN23BO6O2 + 0.5CARB16 + 0.5RN8O2 :  1.68E-12*RO2_b ;
RN24BO4O2 = 0.5C10z + 0.25RN23BO6O2 + 0.25CARB16 + 0.25RN8O2 :  3.36E-12*RO2_b ;

RN23BO6O2 = C10z + 0.5RNxBOyO2 + 0.5CARB16 + 0.5RN8O2 : 1.68E-12*RO2_b ;
RN23BO6O2 = 0.5C10z + 0.25RNxBOyO2 + 0.25CARB16 + 0.25RN8O2 : 3.36E-12*RO2_b ;

RNxBOyO2 = C10z + 0.5RNxBOyO2 + 0.5CARB16 + 0.5RN8O2 :  1.68E-12*RO2_b ;
RNxBOyO2 = 0.5C10z + 0.25RNxBOyO2 + 0.25CARB16 + 0.255RN8O2 :  3.36E-12*RO2_b ;

The effect of these corrections was thoroughly investigated by re-running simulations A-D and recalculating the altitude and nucleation profiles. In all the cases the effect was very small. In Simulations A and B, the change to the k16 values had no effect on the other fitted rate coefficients for accretion product formation (k13) or the autoxidation coefficients. This is unsurprising as the photolysis of HOM was turned off in these simulations and the losses of O3RO2 and OHRO2 to C15d formation and of O3RO2 to the reactions with the RO2$_b$ pool were small relative to autoxidation. In the yield calculation in Simulation C, the change was much smaller than the change arising from the correction to the yield calculation. In Simulation D the effect on all 4 HOM species was very small (<3% change) and the changes to other species such as O3 and OH indistinguishable. Fig 6(b) has been updated with the new HOM concentration values. In the altitude profiles, concentrations of HOM changed by < 2% in Hyytiala and the Amazon, leading to negligible changes in the profiles and nucleation rates. Figures 7 and 8 have been updated. Modelled surface concentrations in Hyytiala changed by 0.01 ppt and by 0.05 ppt in Alabama. Neither change affects the conclusion drawn in each location. Table 5 has been updated to reflect these changes.

> *L 341-347: What is the rate coefficient for the reaction of RTN28OOH + OH (and was it measured?), and why is this the one to stand in for the HOMs?*

This rate coefficient is $2.38 \times 10^{-11}$ molecules$^{-1}$ cm$^3$ s$^{-1}$ and is used by CRI v2.2 for the reaction of large 10-carbon species such as RTN28OOH, which is the closest existing species to a HOM, with OH. The rate coefficient between OH + HOM is not known. Bianchi et al (2019) speculated that it could be close to the collision limit of $1 \times 10^{-10}$ molecules$^{-1}$ cm$^3$ s$^{-1}$ noting that functionalisation of species typically increases the

rate coefficient for its reaction with OH (Atkinson (2000). Sensitivity tests were done with OH + HOM at $1\times10^{-10}$ molecules$^{-1}$ cm$^3$ s$^{-1}$ but this had negligible effect on simulated [HOM] or HOM yield which agrees with the suggestion in Bianchi et al (2019) that HOM loss is dominated by physical removal. This also makes sense when considering that at an OH concentration of $10^{-6}$ cm$^3$, even a rate coefficient at the collision limit would result in an first order rate coefficient of $10^{-4}$ s$^{-1}$, smaller than the condensation sinks from observation used in this paper by at least an order of magnitude. Faced with a lack of further evidence, the rate coefficient was kept at $2.38\times10^{-11}$ molecules$^{-1}$ cm$^3$ s$^{-1}$. However, we have added the following text to line 343 to clarify the rate coefficient used and the sensitivity of the model to this parameter.

"This rate coefficient was $2.38\times10^{-11}$ molecules$^{-1}$ cm$^3$ s$^{-1}$ and, in light of suggestion that the rate coefficient of OH with HOM could be higher (Bianchi et al., 2019), sensitivity tests increasing the rate coefficient to $1\times10^{-10}$ molecules$^{-1}$ cm$^3$ s$^{-1}$ were performed but no material effect was observed."

*Where did the MCM photolysis frequencies come from – were they measured or also extrapolated from other species?*

The MCM photolysis frequencies are documented in several papers (Saunders et al (2003), Jenkin et al (1997)) and more detailed regarding the sources are provided on the MCM website. Photolysis frequencies for larger molecules are extrapolated from those for smaller molecules where more extensive measurements have been made. For example, the photolysis of the peroxide linkage in RTN28OOH is based on the photolysis of the same functional group in $C_2H_5OOH$.

*Again, are there any estimates of the uncertainty in these photolysis/reactions rate coefficients or the sensitivity of the mechanism to them?*

The uncertainty in the photolysis frequency is not readily available and will be a topic for future work. Furthermore, HOM loss via photolysis in the CRI-HOM scheme was a minor loss mechanism; at least an order of magnitude smaller than loss to the condensation sink even at the lower end of CS values. This is supported by the conclusion of Bianchi et al (2019), compiling data from Dal Maso et al (2002), Petäjä et al (2009), Tan et al (2018) and Wu et al (2018), that the major loss mechanism for HOM is physical removal and the following has been added to line 348 to to emphasise this point:

"Physical loss is believed to be the major sink for HOM (Dal Maso et al., 2002, Petäjä et al., 2009, Tan et al., 2018, Wu et al., 2018, Bianchi et al., 2019)."

*While your reasoning that their gas phases losses are unlikely to affect OH or O3 seems reasonable, the loss rates should be very important to new particle formation in some circumstances.*

For each level in the vertical profiles, new particle formation (NPF) rates were calculated using established methodology (Kirkby et al (2016), Gordon et al (2017)) combined with the HOM concentration output by the box model. As discussed (line 584) and shown (Fig. S23), loss of HOM to the CS has a large impact on HOM concentration and thus on NPF rates, especially as new particle formation via the pure biogenic nucleation mechanism shows a non-linear dependence on [HOM]. We agree with

the reviewer that the loss rates are therefore important and feel we have made the sensitivity to this parameter clear to the reader and identified this as an important area for future work.

> *L 392-397: As mentioned in the comment above to L 194-195, couldn't these yields be variable instead of static, and dependent on the environmental conditions that affect the branching pathways of the Criegee intermediates? Numerous past experimental efforts have quantified product yields from a-pinene ozonolysis; are these branching ratios consistent with those past efforts? With what certainty is the 0.206 s-1 autoxidation rate coefficient known, and how sensitive are all the subsequent steps in this mechanism to it?*

As discussed in the response to the comment pertaining to lines 194-95, the branching of α-pinene products from ozonolysis into closed shell species (pinonic acid and pinonaldehyde) and peroxy radical is well understood and fixed values are used as standard (IUPAC Task Group on Atmospheric Chemical Kinetic Data Evaluation) for representative ambient conditions (i.e., mid latitude boundary layer) in the parent CRI v2.2 mechanism and this mechanism. Well established yields of TNCARB26 (16-17%) and RCOOH25 (2%) are combined with the remaining 80% which proceeds via peroxy radical intermediates and produces a range of different products (Atkinson and Arey (2003), Johnson and Marston (2008)). An addition is made to the text (discussed earlier) to highlight to the reader the confidence we can have in this approach. The mechanism in this paper conforms to the 80% yield and the breakdown of the peroxy radical pathway of 30% RTN24O2 and 50% RN26BO2 is similar to the 20:60 split in the MCM of single 9 carbon species C96O2 and C10 species.

The value of 0.206 s$^{-1}$ is based on the theoretical calculations of the relative energies of possible 1st generation O3RO2 and their autoxidation coefficients (Kurten et al., 2015). Kurten et al. (2015) do not provide any error estimates and so in order to understand the impacts of uncertainty in this rate coefficient we have followed the same process as outlined in the main text on line 401 (previously in Table S6, now moved to Table 4 as requested) as for other rate coefficients. This process, which constrains the uncertainty in the rate constants by the measured concentrations of radicals, yields an uncertainty for this reaction of +0.025/ -0.04 s$^{-1}$. Subsequent simulations probing the bounds of this uncertainty result in impacts on the down-stream chemistry and highlight that this is an important parameter for which further constraints in future work would be valuable.

> *L 404-406: This sentence seems grammatically incomplete. Perhaps the "and" isn't needed?*

This has been corrected with the addition of the following amendment.

"The autoxidation coefficients in Table 4 are higher than those considered in the theoretical study of Scherivish et al (2019) but closer to the values measured by Zhao et al (2018) and the values suggested by Roldin et al (2019)."

*L 406-408: This is a crucial point for the present study, and one that I don't think should be so quickly discarded. How (i.e. to what extent) would this change the autoxidation coefficients?*

We acknowledge the importance of this issue and extensive efforts were made to determine the cause. It should be noted that the fitted autoxidation coefficients in this work are not too dissimilar to those used by Roldin et al (2019) or Zhao et al (2018) and so we do not believe the effect to be overwhelming but we do not believe further progress can be made with the currently available data. The presence of additional loss processes would reduce the fitted autoxidation coefficients because they would provide additional sinks for each $RO_2$ without producing the next generation $RO_2$. Therefore, the autoxidation rate coefficients derived here are most probably upper limits. To clarify this, the following adjustment (already discussed in the response to the comments of the 1st reviewer) has been made to line 408:

" Additional loss process would likely reduce the fitted autoxidation coefficients because they would provide an additional sink for the $RO_2$ species which does not lead to the production of the next generation $RO_2$. Therefore, the autoxidation coefficients determined in this work are likely to be upper limits but further insight into this is not possible with the data currently available. This is a key area for further study but no further conclusions can be made with the data currently available**."**

*Line 416-418: This type of sensitivity study is very helpful for understanding the strength of the constraints on your mechanism parameters, but I am confused by the wording of "realistic deviations". Are these the maximum deviations consistent with the autoxidation coefficients, or the maximum deviations consistent with the experiments? If the former, do we have reason to believe that the experiments fell within this range? And if the latter, how were they deemed "realistic"?*

We believe these deviations are representative in the context of the experimental setup. NO and $NO_2$ concentrations were believed to be 4 ppt ($10^8$ cm$^{-3}$) given the purity of the gas used (personal communication with T. Berndt) and so increases of 250 % and decreases of 75% were considered to span the likely range of concentrations. In the absence of measurements, an OH concentration of $10^6$ cm$^{-3}$ was deemed reasonable while the common ratio approximation of 100:1 applied to yield $HO_2$ of 4 ppt ($10^8$ cm$^{-3}$). OH was further investigated with an 100 % increase and 90% decrease and $HO_2$ with a 250 % and decrease of 75%. To clarify this, the following adjustment has been made to line 410-418.

"Unfortunately, the flow tube studies of Berndt et al (2018b) lack observations to constrain the full chemical space simulated by the box model. In particular there were no measurements of NO, $HO_2$ and OH. ~~Therefore, a rigorous series of sensitivity tests (described in the SI) were carried out to quantify the importance of uncertainty in the initial concentrations of OH, NO and HO2 on the results of the model. Initial OH was shown to have no effect on the measured parameters (O3RO2, OHO2 and accretion products) while NO and HO2 had some effect on OHRO2 (mainly through the change to OH) and C20d (via the change to radical termination rate). Initial conditions of 106 cm-3 for OH and 4ppt for HO2, NO and NO2 were used. Based on the sensitivity simulations it was concluded that realistic deviations in concentration of +10 ppt / -3 ppt for HO2, NO and NO2 and + 106 cm-3 / -5×105 cm-3 for OH from the initial values would not lead to deviations in the concentrations of RO2 or accretion products sufficient to warrant a change in the autoxidation coefficients.Likewise increasing and decreasing in OH of (100 %~~

 The effect of the uncertainty in the initial experimental concentrations of NO, HO$_2$ and OH on the modelled concentrations of O3RO2, OHRO2 and accretion products and thus fitted the autoxidation coefficients and accretion production formation rates coefficients was investigated with a series of sensitivity tests. Initial conditions of $10^6$ cm$^{-3}$ for OH and 4 ppt for HO$_2$ were used. NO and NO$_2$ were initialised at 4 ppt, based on the purity of the flow gas (personal communication with T. Berndt). For HO$_2$, NO and NO$_2$ sensitivity simulations indicated that increases of 10 ppt (250% increase) and decreases of 3 ppt (75% decrease) did not lead to deviations in the concentrations of RO$_2$ or accretion products sufficient to warrant a change in the rate coefficients for autoxidation or accretion product formation. Initial OH concentration had negligible effect (<5% change) on O3RO2, OHRO2 and C20d when varied over $1\times10^5 - 2\times10^6$ cm$^{-3}$ (90 % decrease, 100 % increase)."

> *L 426: Should this refer to Fig. 3b instead of a?*

Yes, this correction has been made.

> *L 428-429: Why are these measurement believed to be so drastically underestimated? Whether or not they are, the results suggest that some element of the first-generation OH-derived peroxy radical chemistry is substantially biased in the mechanism; do you have any indication of what this might be?*

This question has been answered in the response to the first reviewer's comments (line 426).

> *L 490: This sentence mentions 4 mechanism versions, but the rest of the paragraph only seems to describe 3.*

The following amendment has been made.

"Given the lack of additional literature in this area, 3 new versions of the new mechanism were created to probe the effects of temperature and activation energy on HOM yield and subsequent evolution."

> *L 507-508: See comment above on lines 222-224: this NOx-dependent behaviour derives from a highly uncertain 50:50 alkoxy radical decomposition : isomerization branching ratio, but is described here as an important consequence of the mechanism. Would this behaviour hold true for a range of reasonable estimates of the branching ratio? How sensitive is it to the chosen branching?*

This response to this comment is included in the response to the comment pertaining to line 222.

> *L 516: Are the model uncertainty ranges reported here for the different temperature dependences? It's very important to distinguish this from some sort of total uncertainty estimate.*

The model uncertainty ranges reported in this section arise from the different temperature dependences. To acknowledge this, the following amendment has been made to the sentence beginning on line 516.

The model predicted total HOM yields at 290 K of 1.9±0.2% (0.01 ppb NO) to 3.9±0.5% (1 ppb NO) with the quoted range resulting from the range of temperature dependencies considered.

*L 518: Missing a word in "comparing favourably to yield measured by" ?*

The error has been corrected in the amendment to the previous comment.

*L 527: What explains the drastic decrease in the simulated acetone production?*

In CRI v2.2, acetone is produced at a fixed 80% yield from α-pinene ozonolysis in a step which is a simplification of the multiple chemical steps which occur in reality. This approach is suited to the CRI's original purpose, simulating air quality in Western Europe with relatively high NOx, but is less accurate at lower NOx. It is in these lower NOx conditions where CRI-HOM produces less acetone than CRI v2.2 and this is because the autoxidation pathways become more important and funnel material away from pathways which produce acetone.

*L 571: "We also at the effect" ... something's wrong here.*

This error has been corrected with the following amendment to line 571.

"We also look at the effect of the simulated HOMs on nucleation rates in the lower troposphere."

*L 588: What does it mean for a result to be semi-qualitative? That even some non- quantitative qualities of the simulated vertical profiles are expected to be erroneous? What ones?*

The term "semi-qualitative" has been amended to "illustrative" as discussed in response to the first reviewer's comment pertaining to line 589.

*Table 4: The uncertainties and sources of these rates are important enough that I think Table S6 should be combined into Table 4.*

This has been done and a further explanation as to how the errors were calculated has been added.

*Figure 3: Why are there no error bars on the 3rd and 4th generation O3RO2 observations? Where do the error bars come from (e.g. are they instrument uncertainty on the measurements or something else)?*

The error bars were omitted for the 3rd and 4th generation as they would be almost exactly the same size as those of the 2nd generation species and make the graph harder to read. The error bars are from experimental uncertainty. To clarify this uncertainty, the caption of Figure 3 was adjusted to read.

"Figure 3 - Comparison of the HOM-precursors (a) O3RO2 and (b) OHRO2 produced by the model and from Berndt et al (2018b) for experiments performed with different initial concentrations of α-pinene (Simulation A). The model reproduces the increase in O3RO2 and 2nd and 3rd generation OHRO2 with initial α-pinene well. The model struggled to reproduce concentrations of the 1st generation OHRO2 (not shown). Note that the error shown is the experimental error from Berndt et al (2018b) and the error bars for the 3rd and 4th generation O3RO2 species are of very similar size to the error bars of the 2nd generation species but have been omitted for clarity."

To clarify this, the following text has been added to the Figure 4 as well:

"The error shown is the experimental error from Berndt et al (2018b)."

> *L 979-980: While the note that first-generation OHRO2 are poorly reproduced is appreciated, it seems misleading not to put them on the graph, and deprives the reader of a visual representation of this important element of the mechanism. Are the concentrations too high to fit on this graph?*

We acknowledge the issue with the 1st generation OHRO2 but feel that the additional detail which has been added in response to comments from both reviewers means that this issue, and the plausible suggestions for its occurrence, has been made sufficiently clear to readers. In addition, the large estimated underprediction in the 1st generation OHRO2 in the work of Berndt et al (2018b) would complicate plotting of the 1st generation OHRO2 alongside the plots of the 2nd and 3rd generation OHRO2 where the experimental error in the observed concentration was much lower. To this end, an additional plot has been added to the SI (Fig. S3) showing the 1st generation OHRO2 experimental and modelled concentrations for Simulation A. Attention is drawn to this in the main text with the following adjustment on line:

"The experimental measurements of 1$^{st}$ generation OHRO2 concentration from Berndt et al (2018b) were believed to be underestimated by about a factor of 5  (Fig. S3)."

> *Figure 5: Can the y-axes be adjusted to show the reader the extent of the measurement uncertainty? The large uncertanties suggest that a wide range of HOM yields would be consistent with the observations, including some yield parameterizations that wouldn't display the much-heralded decrease in C20 (and total) accretion products and increase in C15 products. Were any sensitivity estimates made regarding HOM yields in the mechanism that could be shown here?*

The y-axes of Figure 5 have been adjusted as requested. We acknowledge the issue presented by the large experimental uncertainty and note that HOM yield is not parameterised directly but is influenced by mechanistic parameters like autoxidation coefficients and dimerisation rate coefficients and as such, we feel these are the parameters which should be probed further. In addition, the accretion products contributed negligibly to the HOM yield, given their much lower concentration, and therefore the uncertainty in their concentration will not affect HOM yield appreciably. To investigate the impact of experimental uncertainty on the confidence we can have in the rate coefficient for accretion product formation, we performed some additional sensitivity tests in response to the reviewer's comments where all rate coefficients for reactions forming C20d were scaled by the same factor. Scalings of 0.66 and 2 spanned the region of experimental uncertainty. The same approach was applied separately for the rate coefficients for the formation of the C15d species and scalings of 0.5 and 1.5 spanned the range of experimental uncertainty. To provide the reader with a better idea of the certainty in these mechanistic parameters, as requested by the reviewer, these uncertainty limits have been included in Table S6 and discussed in the main text (see the response to comments pertaining to lines 325 & L309). We acknowledge that this approach will not capture the fact that C20d or C15d concentrations may be more sensitive to changes in the formation rate coefficients for some RO$_2$ than others. To fully investigate this would require a Monte Carlo simulation which is beyond the scope of this work and we feel that the

additional information given in the manuscript and Table S6 from this sensitivity study provides the reader with a fair idea of the impact of the uncertainty in the rate coefficients for accretion product formation.

§

[revised manuscript text omitted]

\* The rate coefficient for the production of the closed shell and alkoxy radical from reaction of the first generation O3RO2 species, RN26BO2, with RO2$_m$ and RO2$_s$ was taken to be the average of the rate coefficients of the three actual species (C107O2, C109O2 and C10BO2 using the notation of Molteni et al (2019)), weighted by the branching ratio of their production. The rate coefficients for C107O2, C109O2 and C10BO2 were calculated using the methodology of Jenkin et al (2019a).

[Figure]

**Figure S1 – Effect of C20d formation rate coefficient on model performance compared to observations from Berndt et al (2018b) under varying initial conditions of α-pinene (Simulation A, Table 3). The model was able to reproduce observed concentrations within experimental error (shaded region) here and in Fig 5 when the rate coefficients were increased with increasing peroxy radical functionalisation (line marked "Vary"). The lines with k= 1×10⁻¹⁰, 1×10⁻¹¹, 1×10⁻¹² and 1×10⁻¹³ show model performance when the specified rate coefficient (in units of cm³ molecules⁻¹ s⁻¹) was used for all O3RO2 and OHRO2. The simulations with accretion formation rate coefficients suggested by Roldin et al (2019) (~ 10⁻¹³-10⁻¹² cm³ molecules⁻¹ s⁻¹) produced significantly lower C20d concentrations while using rate coefficients suggested by Molteni et al (2019) (≥ 10⁻¹⁰ cm³ molecules⁻¹ s⁻¹) overpredicted C20d concentrations.**

[Figure]

**Figure S2 – Effect of C15d formation rate coefficient on model performance compared to observations from Berndt et al (2018b) under conditions of fixed initial α-pinene concentration and varying initial isoprene concentration (Simulation B, Table 3). The model was able to replicate the general trend of increasing C15d with isoprene when the rate coefficients were increased with increasing peroxy radical functionalisation (line marked "Vary"), reproducing observation within experimental error (shaded region). The lines with k=1×10⁻¹¹, 1×10⁻¹² and 1×10⁻¹³ show model performance when the specified rate coefficient (in units of cm³ molecules⁻¹ s⁻¹) was used for all O3RO2 and OHRO2.**

[Figure]

**Figure S3 – 1ˢᵗ generation OHRO2 observation (Berndt et al., 2018b) and modelled with estimated experimental underprediction of factor of 5. Reasons for observation-model discrepancy are discussed in the main text.**

**Table S2 - Summary of HOM mechanisms and autoxidation activation energies**

| Mechanism | Autoxidation Activation Energy / K | Comments |
|---|---|---|
| HOM$_{TI}$ | N/A - temperature independent | Autoxidation coefficients based on fitting from data from Berndt et al (2018b) at 297K |
| HOM$_{6000}$ | 6000 | Representing possible lower bound of activation energy |
| HOM$_{9000}$ | 9000 | Representing possible middle value of activation energy |
| HOM$_{12077}$ | 12077 | Value suggested by Roldin et al (2019) |

**HOM Yield Equations**

The yields for 10-carbon HOMs from ozonolysis ($\gamma_{C10z}$), OH oxidation ($\gamma_{C10x}$) and the total HOM yield ($\gamma_{total}$) are given by Eq. 1, Eq. 2 and Eq. 3 respectively.

$$\gamma_{C10x} = \frac{[C10x](k_{OH+HOM}+CS+J)}{k_{OH}[OH][AP]} \quad (1)$$

$$\gamma_{C10z} = \frac{[C10z](k_{OH+HOM}+CS+J)}{k_{O_3}[O_3][AP]} \quad (2)$$

$$\gamma_{total} = \frac{([C10z]+[C10x])(k_{OH+HOM}+CS+J)}{(k_{O_3}[O_3]+k_{OH}[OH])[AP]} \quad (3)$$

where $[O_3]$, $[OH]$, $[C10z]$ and $[C10x]$ are the concentrations of O$_3$, OH and the 10-carbon HOMs formed from ozonolysis and OH oxidation respectively, $k_{OH+HOM}$ is the rate coefficient for the reactions of HOMs with OH, $CS$ is the HOM condensation sink, $J$ is the HOM photolysis frequency and $k_{O_3}$ and $k_{OH}$ are the reaction rate coefficients of α-pinene with O$_3$ and OH respectively.

[Figure]

**Figure S4 – Modelled HOM yield at 290 K showing with (a) scaling of k14, k15 rate coefficients and (b) perturbations of the alkoxy decomposition-isomerisation branching ratio (originally 50:50).**

65 **Comparison to CRI v2.2**

The new mechanism and the CRI v2.2 were run in a box model (Simulation D, Table 3) for 8 days with varying temperature (298 K average, amplitude of 4 K) and emissions of isoprene and α-pinene varying sinusoidally (Fig S3). Time-independent base NO emissions of $4.7 \times 10^9$ molecules $m^{-2}$ $s^{-1}$ were used with scaling factors of 1, 3, 10, 30, 100 and 200 employed in a manner consistent with Jenkin et al (2015). Time dependent isoprene emissions reached a maximum of $1.1 \times 10^{12}$ molecules

70 $m^{-2}$ $s^{-1}$ at 13:00 local time and had an average of $7.1 \times 10^{11}$ molecules $m^{-2}$ $s^{-1}$ over the period 06:00 to 18:00, similar to emissions used in Jenkin et al (2015) and Bates et al (2019). Time dependent base α-pinene emissions with a mean of $3.23 \times 10^9$ molecules $m^{-2}$ $s^{-1}$ and maximum of $5.30 \times 10^9$ molecules $m^{-2}$ $s^{-1}$ at 1500 hours were applied. Further runs were performed with α-pinene emissions scaled by factors of $10^{-3}$, $10^{-2}$, 0.1, 0.2, 0.5, 1, 2, 3 and 5 to investigate the model's performance. Initial conditions of $CH_4$ (1.8 ppm), CO (100 ppb), $O_3$ (20 ppb) and HCHO (300 ppt) were applied.

Photolysis frequencies simulating conditions at the equator also varied in the diurnal cycle. The box model simulated an instantaneously well-mixed planetary boundary with mixing with the free troposphere (with same composition of initial conditions) represented by the box height increasing from 250 m at night to 1500 m at midday before collapsing back to 250 m at 2100 hours.

The "concentration" of a species was taken to be the mean daytime concentration on the $8^{th}$ day, the metric used by Jenkin et al (2015) and Bates et al (2019). The performance of all the HOM mechanisms ($HOM_{TI}$, $HOM_{6000}$, $HOM_{9000}$ and $HOM_{12077}$) was compared to the CRI v2.2.

85 The HOM mechanisms matched the CRI extremely well for OH, $O_3$, NO, $NO_2$, $HO_2$, α-pinene and isoprene as well as the hydroperoxides and nitrates derived from isoprene, methyl vinyl ketone and methacrolein, and the important SOA precursor isoprene epoxy diol (IEPOX)).

[Figure]

**Figure S5 - Diurnal cycle of emissions of α-pinene and isoprene for 8-day comparison of CRI v2.2 R5 with HOM mechanism versions.**

[Figure]

95 **Figure S64 - Absolute and percentage difference in 8th day daylight mean O₃ between the CRI v2.2 R5 and the HOM₉₀₀₀ mechanism. The difference between mechanisms is less than ±0.05 ppb.**

[Figure]

**Figure S75 - Absolute and percentage difference in 8th day daylight mean OH between the CRI v2.2 R5 and the**
100 **HOM₉₀₀₀ mechanism. The difference between mechanisms is less than ±0.3% for the vast majority of the emissions space with the difference exceeding this only under very high emissions of α-pinene.**

[Figure]

**Figure S86 - Absolute difference in 8th day daylight mean NO between the CRI v2.2 R5 and the HOM₉₀₀₀ mechanism. The**
105 **difference between mechanisms is less than ±2.5 ppt for the vast majority of the emissions space with the difference exceeding this only under very high emissions of NO and α-pinene.**

[Figure]

110 **Figure S9 - 8th day daylight mean O$_3$ in CRI v2.2 R5 and HOM$_{9000}$ model**

[Figure]

**Figure S10 - 8th day daylight mean isoprene in CRI v2.2 R5 and HOM$_{9000}$ model**

[Figure]

**Figure S11 - 8th day daylight mean α-pinene in CRI v2.2 R5 and HOM$_{9000}$ model**

[Figure]

115

**Figure S12 - 8th day daylight mean OH in CRI v2.2 R5 and HOM$_{9000}$ model**

[Figure]

**Figure S13 - 8th day daylight mean 1st generation isoprene peroxy radical in CRI v2.2 R5 and HOM$_{9000}$ model**

[Figure]

120  **Figure S14 - 8th day daylight mean 1st generation isoprene hydroperoxide in CRI v2.2 R5 and HOM$_{9000}$ model**

[Figure]

**Figure S15 - 8th day daylight mean 1st generation isoprene nitrate in CRI v2.2 R5 and HOM$_{9000}$ model**

[Figure]

**Figure S16 - 8th day daylight mean isoprene epoxydiol in CRI v2.2 R5 and HOM$_{9000}$ model**

[Figure]

125

**Figure S17 - 8th day daylight mean combined methyl vinyl ketone and methacrolein in CRI v2.2 R5 and HOM$_{9000}$ model**

[Figure]

**Figure S16 - 8th day daylight mean combined acetone in CRI v2.2 R5 and HOM$_{9000}$ model. The difference was attributed to the added competition supplied by the autoxidation pathways, diverting the degradation of α-pinene away from the traditional pathways which form acetone. However, this discrepancy between mechanisms did not led to significant disagreement between the HOM mechanism and CRI v2.2 R5 for O$_3$ and OH concentrations.**

[Figure]

**Figure S19 – Peroxy radicals from ozonolysis (O3RO2) exhibiting a decrease with NO$_x$ and the clear dominance of the highest generation peroxy radical. Negligible difference is observed between the 4 HOM mechanisms for each peroxy radical.**

[Figure]

**Figure S20** – Peroxy radicals from OH oxidation (OHRO2) exhibiting a decrease with $NO_x$ and the clear dominance of the highest generation peroxy radical. Negligible difference is observed between the 4 HOM mechanisms for each peroxy radical.

140

[Figure]

**Figure S21** – Closed shell species in base mechanism compared to $HOM_{TI}$ mechanism. The lower concentrations of TNCARB26, CARB16 and RTN28NO3 were attributed to the increased competition from the autoxidation pathways in the HOM mechanism. RN18NO3 was significantly lower in the HOM mechanisms (not shown) as discussed in the main text.

145

150

[Figure]

155 Figure S22 – Observed concentrations of isoprene (left) and monoterpene (right) at the ATTO tower at 1:30-2:30 pm in June 2013 (Yáñez-Serrano et al., 2015) (shading shows standard deviation of observational data) and modelled concentrations of species.

[Figure]

160

**Figure S230 – Total HOM concentrations in Amazon and Hyytiala with shaded region showing the effect of increasing/decreasing CS by a factor of 10. The value of the CS has a significant influence on HOM concentrations.**

165 **Table S3 – Species and physical parameters used in the HOM altitude profile modelling. Note that for nucleation calculations, the same input species and parameters were used but all data were monthly means.**

| Data from UKCA run (2pm 16th June, averaged over 2010-2104) | Data from UKESM Historical |
|---|---|
| Temperature, pressure, $O_3$, OH, isoprene, monoterpene*, NO, $NO_2$, $NO_3$, $N_2O_5$, CO, $HO_2$, $H_2O$ | $CH_4$, CO, HCHO, $CH_3O_2$, $C_2H_5O_2$, isoprene nitrate and hydroperoxides, $H_2O_2$, $CH_3OOH$, HONO, $C_2H_6$, $C_2H_5OOH$, $CH_3CHO$, PAN, $C_3H_8$, $C_3H_7OOH$, $C_2H_5CHO$, $CH_3NO_3$, Methacrolein, Methylglyoxal, HCOOH, $CH_3CO_3$, $C_3H_7O_2$, $C_2H_5CO_3$, $CH_3OH$ |

* The modelled monoterpene concentration was halved to approximate the α-pinene concentration (Rinne et al., 2002)

**Table S4 - Values of surface level CS and local time of run used for HOM altitude profiles (Lee et al., 2016)**

| | Location | | |
|---|---|---|---|
| | Hyytiala | Manaus | Brent, Alabama |
| CS / s$^{-1}$ | 0.004 | 0.9 | $0.012 \pm 0.006$ |
| Local time | 14:00 | 14:00 | 12:00 |

170

**Nucleation Parameterisations**

The rates of neutral and ion-induced pure biogenic nucleation ($J_n$ and $J_{iin}$ respectively) are described by the parameterisations (Kirkby et al (2016)) in Eq. 4 and Eq. 5:

$$J_n = a_1[HOM]^{a_2 + \frac{a_5}{[HOM]}} \text{ (4)}$$

175 $$J_{iin} = a_3[HOM]^{a_4 + \frac{a_5}{[HOM]}}[n_\pm] \text{ (5)}$$

Where $a_i$ are fitted parameters and $[n_\pm]$ the concentration of ions calculated by method described Kirkby et al (2016). In this work, no distinction was made between the different HOM species; the [HOM] term was taken as the sum of all HOM species. In reality, the larger accretion products are likely to be better at nucleating due to their lower volatility and even among 10-carbon HOMs, more oxidised species will also be more proficient at new particle formation. The condensation 180 sink for ions was calculated by summing over aerosol modes and (Eq. 6).

$$CS = \frac{2kT\mu}{\varepsilon}\sum(wd \times c) \times 10^6 \text{ (6)}$$

Where $k$ is the Boltzmann constant, $T$ temperature (in Kelvin), $\mu$=1.2 x 10$^{-4}$ m$^2$ V$^{-1}$ s$^{-1}$, $\varepsilon$ =1.6022 x 10$^{-19}$ C, $wd$ is the wet diameter (in m) of the aerosol mode and $c$ the mode's particle concentration (per cm$^3$) (wd and c were taken from UKCA run).

185 The ion loss rate, $X$, was then calculated as the sum of the condensation and nucleation sinks (Eq. 7).

$$X = CS + a_3[HOM]^{a_4 + \frac{a_5}{[HOM]}} \text{ (7)}$$

The recombination coefficient, $\alpha$, is given by Eq. 8:

$$\alpha = 6 \times 10^{-8}\sqrt{\frac{300}{T}} + 6 \times 10^{-26}c_{air}\left(\frac{300}{T}\right)^4 \text{ (8)}$$

Where $c_{air}$ is the concentration of air in molecules per cm$^3$.

190 $$[n_\pm] = \frac{\sqrt{(X^2 - 4\alpha q)} - X}{2\alpha} \text{ (9)}$$

Where $q$ is the rate of ion-pair production in cm$^{-3}$ s$^{-1}$.

The sulphuric acid activation parameterisation used was that developed by Kulmala et al (2006) with coefficient from Sihto et al (2006) as used by Scott et al (2014) (Eq. 10).

$$J_{act} = A[H_2SO_4]$$

195    (10)

Where $A=2\times10^{-6}$ s$^{-1}$

[Figure]

**Figure S21 – Percentage contribution to total nucleation rate (PBN + SA$_{act}$) of PBN. Significant increase is predicted for the PI**
200    **Hyytiala case in particular, indicating the important implications of including PBN in climate models.**

**Changes to CRI v2.2 R5 mechanism**

Simple rate coefficients (e.g. kRO2NO) and photolysis frequencies (e.g. J41) were taken from CRI (Jenkin et al., 2008,
Jenkin et al., 2019b). Unless otherwise stated, unimolecular rate coefficients have units of s$^{-1}$. The peroxy radical pools
205   (RO2$_b$, RO2$_m$, RO2$_s$ and RO2) represent the total concentration of peroxy radicals falling within the respective pool. In the
mechanism used in modelling, certain reactions were lumped together with product fractions weighted by relative rate
coefficients to reduce the total number of reactions. For clarity, reactions have been decomposed below. The autoxidation
coefficients provided are those fitted at 297 K. Table S5 shows the expressions for the autoxidation coefficients in the 3
temperature dependent mechanisms.

210    The standard reactions rate coefficients used by the CRI are as follows:

KRO2NO = 2.7D-12*EXP(390/TEMP)

KRO2HO2 = 2.91D-13*EXP(1300/TEMP)

KRO2NO3 = 2.3D-12

**Reactions removed from CRI v2.2 R5 mechanism**

215    **Ozonolysis of alpha pinene and treatment of resulting peroxy radical RN18AO2**

APINENE + O3=OH+CH3COCH3+RN18AO2 : 8.05D-16*EXP(-640/TEMP)*0.80  ;

APINENE + O3 = TNCARB26 + H2O2 : 8.05D-16*EXP(-640/TEMP)*0.175  ;

APINENE + O3 = RCOOH25 : 8.05D-16*EXP(-640/TEMP)*0.025 ;

RN18AO2 + NO = CARB16 + HO2 + NO2 :  KRO2NO*0.946 ;

220    RN18AO2 + NO = RN18NO3 :  KRO2NO*0.054 ;

RN18AO2 + NO3 = CARB16 + HO2 + NO2 :  KRO2NO3 ;

RN18AO2 + HO2 = RN18OOH :  KRO2HO2*0.770 ;

RN18AO2 = CARB16 + HO2 :  8.80D-13*RO2 ;

**OH oxidation of alpha pinene and treatment of resulting peroxy radical RTN28O2**

225    APINENE + OH = RTN28O2 :  1.20D-11*EXP(444/TEMP) ;

RTN28O2 + NO = TNCARB26 + HO2 + NO2 :  KRO2NO*0.767*0.915 ;

RTN28O2 + NO = CH3COCH3 + RN19O2 + NO2 :  KRO2NO*0.767*0.085 ;

RTN28O2 + NO = RTN28NO3 :  KRO2NO*0.233 ;

RTN28O2 + NO3 = TNCARB26 + HO2 + NO2 :  KRO2NO3 ;

230    RTN28O2 + HO2 = RTN28OOH :  KRO2HO2*0.914 ;

RTN28O2 = TNCARB26 + HO2 :  2.85D-13*RO2 ;

**Reactions added**

**Ozonolysis of α-pinene producing 1st generation O3RO2, RN26BO2 - branching ratio set to 50% based on experimental observations of Berndt et al (2018b)**

235    1. APINENE + O3 = 0.14375TNCARB26 + 0.0625RCOOH25 + 0.85OH + 0.5RN26BO2 + 0.3RTN2402: 8.05E-16*EXP(-640/TEMP);

**Reactions of RN26BO2**

Reaction with HO2 forms hydroperoxide species already in CRI, not a HOM due to insufficient oxygens.

2. RN26BO2 + HO2 = RTN26OOH : KRO2HO2*0.9;

Reaction with NO, NO₃ forms next generation O3RO2 via alkoxy radical isomerisation and fragmentation products (smaller RO2, RN9O2, and closed shell species, CARB16) at 50:50 branching ratio). NO also forms small yield of RN18NO3, estimated from original CRI v2.2 R5.

3. RN26BO2 + NO = 0.025RTN28NO3 + 0.487RN25BO2O2 + 0.487CARB16 + 0.487RN9O2 + 0.975NO2: KRO2NO;

4. RN26BO2 + NO3 = 0.5RN25BO2O2 + 0.5CARB16 + 0.5RN9O2 + NO2 : KRO2NO3 ;

**Autoxidation of RN26BO2 to 2nd generation O3RO2, RN25BO2O2**

5. RN26BO2 = RN25BO2O2 : 0.206 ;

**Reactions of RN25BO2O2**

**Reaction with HO2 forms HOM monomer C10z as product has sufficient oxygens. Reaction with NO, NO₃ follows the same principle as RN26BO2.**

6. RN25BO2O2 + HO2 = C10z : KRO2HO2*0.914;

7. RN25BO2O2 + NO = 0.5RN24BO4O2+ 0.5CARB16 + 0.5RN8O2 + NO2: KRO2NO

8. RN25BO2O2 + NO3 = 0.5RN24BO4O2+ 0.5CARB16 + 0.5RN8O2 + NO2 : KRO2NO3 ;

**Autoxidation of RN25BO2O2 to 3rd generation O3RO2, RN24BO4O2**

9. RN25BO2O2 = RN24BO4O2 : 1.7;

**Reactions of RN24BO4O2**

10. RN24BO4O2 + HO2 = C10z : KRO2HO2*0.914;

11. RN24BO4O2 + NO = 0.5RN23BO6O2+ 0.5CARB16 + 0.5RN8O2 + NO2: KRO2NO

12. RN24BO4O2 + NO3 = 0.5RN23BO6O2+ 0.5CARB16 + 0.5RN8O2 + NO2 : KRO2NO3 ;

**Autoxidation of RN24BO4O2 to 4th generation O3RO2, RN23BO6O2**

13. RN24BO4O2 = RN23BO6O2 : 1.7 ;

**Reactions of RN23BO6O2**

14. RN23BO6O2 + HO2 = C10z : KRO2HO2*0.914;

15. RN23BO6O2 + NO = 0.5RNxBOyO2 + 0.5CARB16 +0.5RN8O2 + NO2: KRO2NO ;

16. RN23BO6O2 + NO3 = 0.5RNxBOyO2 + 0.5CARB16 +0.5RN8O2 + NO2: KRO2NO3 ;

**Autoxidation of RN23BO6O2 to lumped "5th generation and higher" O3RO2, RNxBOyO2**

17. RN23BO6O2 = RNxBOyO2 : 1.6;

**Reactions of RNxBOyO2 - no further autoxidation**

18. RNxBOyO2 + HO2 = C10z : KRO2HO2*0.914;

19. RNxBOyO2 + NO = 0.5RNxBOyO2 + 0.5CARB16 + 0.5RN8O2 + NO2: KRO2NO ;

20. RNxBOyO2 + NO3 = 0.5RNxBOyO2 + 0.5CARB16 + 0.5RN8O2 + NO2: KRO2NO3 ;

**Reactions of O3RO2 with big peroxy radical pool (RO2_b)**

All reactions with RO2$_b$ produce a 20-carbon accretion product at a rate coefficient from fitting to experimental data (Berndt et al., 2018b). Reactions also produce, with equal rate coefficients (from Molteni et al., 2019), closed shell species which are classified as HOMs for all cases (except for the reaction of RN26BO2 which is not sufficiently oxidised) and alkoxy radicals which go on to react as previously described in this work.

21. RN26BO2 = 0.5 C20d : 0.97E-11 RO2$_b$ ;

22. RN26BO2 = TNCARB26 + 0.5 CARB16 + 0.5 RN25BO2O2 : 1.68E-12 RO2$_b$ ;

22. RN26BO2 = 0.5TNCARB26 + 0.25RN25BO2O2 + 0.25CARB16 + 0.25RN8O2 : 3.36E-12*RO2_b ;

23. RN25BO2O2 = 0.5 C20d : 2.5E-11 RO2$_b$ ;

24. RN25BO2O2 = C10z + 0.5 RN24BO4O2 + 0.5 CARB16 + 0.5 RN8O2 : 1.68E-12 RO2$_b$ ;

24. RN25BO2O2 = 0.5C10z + 0.25RN24BO4O2 + 0.25CARB16 + 0.25RN8O2 : 3.36E-12*RO2_b;

25. RN24BO4O2 = 0.5 C20d : 3.4E-11 RO2$_b$ ;

26. RN24BO4O2 = C10z + 0.5 RN23BO6O2 + 0.5 CARB16 + 0.5 RN8O2 : 1.68E-12 RO2$_b$ ;

26. RN24BO4O2 = 0.5C10z + 0.25RN23BO6O2 + 0.25CARB16 + 0.25RN8O2 : 3.36E-12*RO2_b ;

27. RN23BO6O2 = 0.5 C20d : 3.6E-11 RO2$_b$;

28. RN23BO6O2 = C10z + 0.5 RNxBOyO2 + 0.5 CARB16 + 0.5 RN8O2 : 1.68E-12 RO2$_b$ ;

28. RN23BO6O2 = 0.5C10z + 0.25RNxBOyO2 + 0.25CARB16 + 0.25RN8O2 : 3.36E-12*RO2_b ;

29. RNxBOyO2= 0.5 C20d : 3.6E-11 RO2$_b$ ;

30. RNxBOyO2 = C10z + 0.5 RNxBOyO2 + 0.5 CARB16 + 0.5 RN8O2 : 1.68E-12 RO2$_b$ ;

30. RNxBOyO2 = 0.5C10z + 0.25RNxBOyO2 + 0.25CARB16 + 0.255RN8O2 : 3.36E-12*RO2_b ;

**Reactions of O3RO2 with medium and small peroxy radical pools (RO2$_m$ and RO2$_s$)**

Reaction of RN26BO2 is based on corresponding species in MCM.

31. RN26BO2 = 0.5RN25BO2O2 + 0.5CARB16 + 0.5RN9O2 : 8.13E-13 (RO2$_s$+RO2$_m$) ;

Rate coefficient and branching ratios of later generation O3RO2 with medium and small peroxy radical pools taken from Roldin et al (2019). The alkoxy radical produced goes on to react as described earlier in this work.

32. RN25BO2O2 = 0.3RN24BO4O2 + 0.3CARB16 + 0.3RN8O2 + 0.4C10z: 5E-12 (RO2$_s$+RO2$_m$) ;

33. RN24BO4O2 = 0.2RN23BO6O2 + 0.2CARB16 + 0.2RN8O2 + 0.6C10z: 7E-12 (RO2$_s$+RO2$_m$);

34. RN23BO6O2 = 0.1RNxBOyO2 +0.1CARB16 +0.1RN8O2 + 0.8C10z : 9E-12 (RO2$_s$+RO2$_m$) ;

35. RNxBOyO2 = 0.1RNxBOyO2+0.1CARB16+0.1RN8O2+0.8C10z : 1E-11 (RO2$_s$+RO2$_m$) ;

Rate coefficient of O3RO2 with isoprene-derived peroxy radical from fitting of model to experimental data (Berndt et al, 2018b).

36. RN26BO2 = 0.667C15d : 3.9E-12 RO2$_m$ ;

37. RN25BO2O2 = 0.667C15d : 5.24E-12 RO2$_m$ ;

38. RN24BO4O2 = 0.667C15d : 6.5E-12 RO2$_m$ ;

39. RN23BO6O2 = 0.667C15d : 6.5E-12 RO2$_m$ ;

40. RNxBOyO2 = 0.667C15d : 7.55E-12 RO2$_m$ ;

**OH oxidation of alpha pinene producing two OHRO2 - RTN28AO2 + RTN28BO2**

41. APINENE + OH = 0.78 RTN28AO2 + 0.22 RTN28BO2: 1.20E-11*EXP(440/TEMP);

**Reactions of RTN28AO2 are the same as for RTN28O2 in original CRI v2.2 R5 except for accretion product formation. RTN28AO2 does not undergo autoxidation.**

42. RTN28AO2 + NO = 0.23RTN28NO3 + 0.77TNCARB26 + 0.77NO2: 2.7D-12*EXP(360/TEMP)*0.767 ;

43. RTN28AO2 + HO2 = RTN28OOH : 2.91D-13*EXP(1300/TEMP)*0.914 ;

44. RTN28AO2 + NO3 = TNCARB26 + HO2 + NO2 : 2.3D-12 ;

45. RTN28AO2 = TNCARB26: 6.65E-13*RO2 ;

46. RTN28AO2 = 0.5 C20d : 0.4E-11*RO2_b ;

47. RTN28AO2 = 0.667C15d : 1.82E-12*RO2_m ;

**Reactions of RTN28BO2**

Reaction with HO2 forms hydroperoxide species already in CRI, not a HOM due to insufficient oxygens.

48. RTN28BO2 + HO2 = RTN28OOH : KRO2HO2*0.914 ;

Reaction with NO, NO$_3$ forms next generation OHRO2 via alkoxy radical isomerisation and fragmentation products (smaller RO2, RN9O2, and closed shell species, CARB16) at 50:50 branching ratio). NO also forms small yield of RN18NO3, estimated from original CRI v2.2 R5.

49. RTN28BO2 + NO = 0.125*RTN28NO3 + 0.875CH3COCH3 + 0.875RN19O2 + 0.875NO2: KRO2NO ;

50. RTN28BO2 + NO3 = CH3COCH3 + RN17O2 + NO2 : KRO2NO3 ;

**Autoxidation of RTN28BO2 to produce 2nd generation OHRO2, RTN27BO2O2**

51. RTN28BO2 = RTN27BO2O2 : 2.1 ;

**Reactions of RTN27BO2O2**

Reaction with HO2 forms hydroperoxide species already in CRI, not a HOM due to insufficient oxygens.

52. RTN27BO2O2 + HO2 = RTN28OOH : KRO2HO2*0.914 ;

53. RTN27BO2O2 + NO = 0.5RTN26BO4O2+0.5CARB16 + 0.5RN9O2 +NO2: KRO2NO ;

54. RTN27BO2O2 + NO3 = 0.5RTN26BO4O2+0.5CARB16 + 0.5RN9O2 +NO2: KRO2NO3 ;

**Autoxidation of RTN27BO2O2 to produce 3rd generation OHRO2, RTN26BO4O2**

55. RTN27BO2O2 = RTN26BO4O2 : 2.1 ;

**Reactions of RTN26BO4O2**

Hydroperoxide from RTN26BO4O2 has sufficient oxygens to be classified as a HOM.

56. RTN26BO4O2 + HO2 = C10x : KRO2HO2*0.914 ;

57. RTN26BO4O2 + NO = 0.5RTNxBOyO2 + 0.5CARB16 + 0.5RN8O2 +NO2: KRO2NO ;

58. RTN26BO4O2 + NO3 = 0.5RTNxBOyO2+0.5CARB16 + 0.5RN8O2 +NO2: KRO2NO3 ;

**Autoxidation of RTN26BO4O2 to produce "4th generation and higher" OHRO2, RTNxBOyO2**

59. RTN26BO4O2 = RTNxBOyO2 : 0.25 ;

**Reactions of RTNxBOyO2 - no further autoxidation occurs**

60. RTNxBOyO2 + HO2 = C10x : KRO2HO2*0.914 ;

61. RTNxBOyO2 + NO = 0.5RTNxBOyO2 + 0.5CARB16 + 0.5RN8O2 +NO2: KRO2NO ;

62. RTNxBOyO2 + NO3 = 0.5RTNxBOyO2 + 0.5CARB16 + 0.5RN8O2 +NO2: KRO2NO3 ;

**Reactions of OHRO2 with big peroxy radical pool (RO2$_b$)**

350  Rate coefficient from fitting of model to experimental data (Berndt et al, 2018b).

63. RTN28BO2 = 0.5 C20d : 0.4E-11 RO2$_b$ ;

64. RTN27BO2O2 = 0.5 C20d : 2.5E-11 RO2$_b$ ;

65. RTN26BO4O2 = 0.5 C20d : 5.5E-11 RO2$_b$ ;

66. RTNxBOyO2 = 0.5 C20d : 3.5E-11 RO2$_b$ ;

355  **Reactions of OHRO2 with medium, small and total peroxy radical pools (RO2$_m$, RO2$_s$ and RO2)**

67. RTN28BO2 = 0.7TNCARB26 + 0.3CH3COCH3 + 0.3RN17O2 : 6.7E-15*RO2

68. RTN27BO2O2 = 0.4TNCARB26 + 0.3RTN26BO4O2 + 0.3CARB16 + 0.3RN10O2 : 5E-12*RO2 ;

69. RTN26BO4O2 = 0.4C10x + 0.3RTNxBOyO2 + 0.3CARB16 + 0.3RN9O2: 8E-12*RO2 ;

70. RTNxBOyO2 = 0.8C10x + 0.1RTNxBOyO2 + 0.1CARB16 + 0.1RN8O2 : 1E-11*RO2 ;

360  Rate coefficient of OHRO2 with isoprene-derived peroxy radical from fitting of model to experimental data (Berndt et al, 2018b).

71. RTN28BO2 = 0.667C15d : 1.8E-12 RO2$_m$ ;

72. RTN27BO2O2 = 0.667C15d: 3.75E-12 RO2$_m$ ;

73. RTN26BO4O2 = 0.667C15d: 3.75E-12 RO2$_m$ ;

365  74. RTNxBOyO2 = 0.667C15d : 3.75E-12 RO2$_m$ ;

**Photolysis of HOMs**

Photolysis of peroxide linkage and carbonyl linkages were considered using MCM frequencies J41 and J22 respectively. The KPP parameter "SUN" was used  in experiments where the photolysis frequency was varied.

Photolysis of peroxide linkage in HOM monomer produces one OH and one alkoxy radical which behaves as previously
370  discussed (50% decomposition, 50% isomerisation). As the extent of oxidation of the HOM is unknown, isomerisation produces second generation peroxy radical by default.

75. C10z = 0.5CARB16 + 0.5RN9O2 + 0.5RN25BO2O2 + OH : J41;

76. C10x = 0.5CARB16 + 0.5RN9O2 +  0.5RTN27BO2O2 + OH: J41;

Photolysis of C20d produces two alkoxy radicals. The isomerisation products are 2nd gen OHRO2 and 2nd gen O3RO2.

375  77. C20d = 0.5RN25BO2O2 + 0.5RTN27BO2O2 + RN9O2 + CARB16 : J41;

Photolysis of C15d produces two alkoxy radicals. The isoprene-derived alkoxy radical produces UCARB12 (as inCRI v2.2 R5) while the alkoxy radical from alpha pinene forms next generation peroxy radicals via isomerisation (50 % OHRO2 and 50% O3RO2) and fragmentation products.

78. C15d = UCARB12 + 0.25RN25BO2O2 + 0.25RTN27BO2O2 + 0.5RN9O2 + 0.5CARB16: J41;

380  Photolysis of carbonyl linkage produces an acyl radical and an alkyl radical which will form peroxy radicals. It is assumed that one of these peroxy radical is big enough to be considered (2nd generation) O3RO2 or OHRO2

79. C10z = RN25BO2O2 : J22;

80. C10x = RTN27BO2O2:  J22;

81. C20d = RN25BO2O2 + RTN27BO2O2 : J22;

385  For C15d, one of the two peroxy radicals formed is assumed to be of medium size and produce UCARB12 which isomerisation (as occurs for isoprene-derived peroxy radicals).

82. C15d = 0.5RN25BO2O2 + 0.5RTN27BO2O2 + UCARB12: J22;

**HOM loss to OH**

All HOM species are lost to OH with same rate coefficient as that for large hydroperoxide RTN28OOH in CRI v2.2 R5. The
390  products, closed shell CRI species CARB10 and CARB15, were chosen under the assumption that the HOM fragments and the sum of CRI indices of the product is close to the CRI index of the peroxy radical which formed the HOM (23-27). The reaction of C15d also produces a product featured in the oxidation pathway of isoprene, UCARB10.

83. C10z + OH = CARB10 + CARB15 + OH :  2.38E-11 ;

84. C10x + OH = CARB10 + CARB15 + OH :  2.38E-11 ;

395  85. C15d + OH = CARB10 + CARB15 + UCARB10+ OH :  2.38E-11 ;

86. C20d + OH = 2CARB10 + 2CARB15 + OH :  2.38E-11 ;

**Table S5 - Temperature dependencies used for 3 temperature dependent mechanism versions.**

| Species | $HOM_{6000}$ | $HOM_{9000}$ | $HOM_{12077}$ |
|---|---|---|---|
| RN26BO2 | 1.223E8*EXP(-6000/T) | 2.981E12*EXP(-9000/T) | 9.413E16*EXP(-12077/T) |
| RN25BO2O2 | 1.009E9*EXP(-6000/T) | 2.460E13*EXP(-9000/T) | 7.768E17*EXP(-12077/T) |

| | | | |
|---|---|---|---|
| RN24BO4O2 | 1.009E9*EXP(-6000/T) | 2.460E13*EXP(-9000/T) | 7.768 E17*EXP(-12077/T) |
| RN23BO6O2 | 9.500E8*EXP(-6000/T) | 2.315E13*EXP(-9000/T) | 7.311E17*EXP(-12077/T) |
| RTN28BO2 | 1.247E9*EXP(-6000/T) | 3.038E13*EXP(-9000/T) | 9.595E17*EXP(-12077/T) |
| RTN27BO2O2 | 1.247E9*EXP(-6000/T) | 3.038E13*EXP(-9000/T) | 9.595E17*EXP(-12077/T) |
| RTN26BO4O2 | 1.484E8*EXP(-6000/T) | 3.617E12*EXP(-9000/T) | 1.142E17*EXP(-12077/T) |

400  The uncertainty in the autoxidation coefficients was estimated by further box models simulations where an autoxidation coefficient was adjusted so that the corresponding species was simulated at the upper and lower concentrations values given the experimental uncertainty.

**Table S6 – Estimated Uncertainty in Autoxidation Coefficients (at 297 K)**

| Species | Coefficient / s$^{-1}$ |
|---|---|
| RN26BO2 | 0.206 (+0.025/ -0.04) |
| RN25BO2O2 | 1.7 (+1.1/-0.4) |
| RN24BO4O2 | 1.7(+1.1 / -0.4) |
| RN23BO6O2 | 1.6 (+0.8/ -0.5) |
| RTN28BO2 | 2.1 Taken directly from Xu et al., 2018 |
| RTN27BO2O2 | 2.1 (+1.6 / -0.2 ) |
| RTN26BO4O2 | 0.25 (+0.3 / -0.1) |

405  **Table S6 – References for model parameters and confidence**

| Parameter | Value(s) | Source | Confidence |
|---|---|---|---|
| Autoxidation Coefficients | Detailed in Table 4 | Derived in this work* | Estimated uncertainty in Table 4 |
| Rate coefficients for C20d formation (k13) | Section 2.3.1 and SI reaction list: 21, 23, 25, 27, 29, 46, 63, 64, 65, 66 | Derived in this work | Sensitivity tests suggested uncertainty range of +100% / - 35% (scalings of 0.65-2) |
| Rate coefficients for C15d formation (k16) | Section 2.3.2 and SI reaction list: 36, 37, 38, 39, 40, 47, 71, 72, 73, 74 | Derived in this work | Sensitivity tests suggested uncertainty range of ±50% (scalings of 0.5-1.5) |

| | | | |
|---|---|---|---|
| Closed Shell / Alkoxy radical from a specific big RO$_2$ reacting with RO$_2$b pool (k14, k15) | Section 2.3.1 and SI reaction list: 22, 24, 26, 28, 30, 32-35 | Molteni et al (2019), Roldin et al (2019) | Scaling by factors of 10 and 0.1 did not affect rate coefficients fitted for autoxidation or accretion product formation. HOM yield greater sensitivity with universal scalings of +100% and –50% resulting in a doubling and halving of HOM yield respectively. |
| k14/k15 branching ratio | 50:50 | Ratio similar to Jenkin et al (2019a) values of 40:60 (1°, 2° RO$_2$), 20:80 (3° RO$_2$) | Sensitivity tests with ratios of 40:60 and 20:80 did not affect rate coefficients fitted for autoxidation or accretion product formation and had minor effects on HOM yield. |
| HOM + OH rate coefficient | $2.38 \times 10^{-11}$ molecules$^{-1}$ cm$^3$ s$^{-1}$ | Based on comparable species in CRI v2.2, RTN28OOH | Increasing rate coefficient to collision limit (as suggested by Bianchi et al., 2019) had negligible affect |
| Alkoxy radical decomposition-isomerisation branching ratio | 50:50 | Estimate | Sensitivity tests with ratios of 75:25 and 25:75 did not affect rate coefficients fitted for autoxidation or accretion product formation. HOM yield below 200 ppt NOx was unaffected and at 2 ppb NOx, uncertainty in autoxidation temperature dependence dwarfed this uncertainty. Ratio more important at higher NOx but this coincided with drastically reduced HOM yield. |

*Note that the autoxidation rate coefficient for the 1$^{st}$ generation species RTN28BO2 was taken from Xu et al (2018)

**Breakdown of Peroxy Radical Pools In CRI-HOM**

**Large Peroxy Radical Pool (8 or more carbons)**

410 RTN28AO2, RTN28BO2, RTN27BO2O2, RTN26BO4O2, RTNxBOyO2, RN26BO2, RTN24O2, RN25BO2O2, RN24BO4O2, RN23BO6O2, RNxBOyO2, NRTN28O2, RA19CO2, RTX28O2, NRTX28O2, RTN26O2, RTN25O2, RTX22O2, RTN24O2, RTN23O2

**Medium Peroxy Radical Pool (4-7 carbons)**

RU12O2, NRU12O2, RN13O2, RN12O2, NRN12O2, RA13O2, DHPR12O2, RN11O2, RA16O2, RU10O2, RU10AO2, 415 MACO3, RN13AO2, RU12O2, NRU12O2, RTN14O2, RN16AO2, RN14O2, RTN10O2, RN17O2, RN15AO2, RN15O2, RN18AO2, RN16O2, RN18O2, RN19O2

**Small Peroxy Radical Pool (3 or fewer carbons)**

CH3O2, C2H5O2, HOCH2CH2O2, CH3CO3, C2H5CO3, ICH3H7O2, RN10O2, HOCH2CO3, NRN6O2, RN9O2, NRN9O2, RN8O2

420

**Sensitivity Tests**

Initial concentrations of 4 ppt of NO, NO$_2$ and HO$_2$ (1x10$^8$ cm$^{-3}$) and 1x10$^6$ cm$^{-3}$ OH were used in the modelling of flow cell data from Berndt et al (2018b). The uncertainty in the experimental concentrations of NO, HO$_2$ and OH has an effect on the modelled concentrations of O3RO2, OHRO2 and accretion products and thus fitted the autoxidation coefficients and

425 accretion production formation rates coefficients. To assess the effect of this uncertainty, multiple model runs were carried out with different initial conditions of NO, HO and HO$_2$.

**NO**

NO concentrations were believed to be below 1x10$^8$ cm$^{-3}$ (~4 ppt) and initial conditions from 1x10$^7$ cm$^{-3}$ (0.4 ppt) to 1x10$^{10}$ cm$^{-3}$ (0.4 ppb) were considered with particular attention paid to the range 5x10$^7$ – 5x10$^8$ cm$^{-3}$. The O3RO2 exhibited

430 negligible dependence on initial NO while OHRO2 displayed a noticeable but small dependence. Relative to the assumed NO concentration of 1x10$^8$ cm$^{-3}$, NO of 5x10$^8$ cm$^{-3}$ increased OHRO2 concentrations by <10% (slightly larger than experimental uncertainty) while NO of 5x10$^7$ cm$^{-3}$ led to a decrease of <5%. C20d also exhibited negligible dependence on NO (<2%). Given that NO was likely to be less than 1x10$^8$ cm$^{-3}$ and the effect of lowering the concentration further was observed to be considerably smaller than experimental error, the uncertainty in NO was considered of minor importance.

435 ### HO$_2$

The initial concentration HO$_2$ was varied from 1x10$^7$ cm$^{-3}$ (0.4 ppt) to 1x10$^{10}$ cm$^{-3}$ (0.4 ppb) (initial NO of 1x10$^8$ cm$^{-3}$). O3RO2 species showed little dependence to initial HO$_2$ between 0.4 ppt and 80 ppt while OHRO2 exhibited greater dependence with 40 ppt increasing OHRO2 by up to 35% relative to 4 ppt and 0.4 ppt decreasing OHRO2 by < 10% and C20d varied by <±10% from 0.4 ppt to 40 ppt initial HO2 (all within experimental uncertainty).

440 ### OH

Initial OH concentration had negligible effect on O3RO2, C20d and OHRO2 even when it was varied over two orders of magnitude (10$^5$ – 10$^7$ cm$^{-3}$).